# Towards complete and error-free genome assemblies of all vertebrate species

High-quality and complete reference genome assemblies are fundamental for the application of genomics to biology, disease, and biodiversity conservation. However, such assemblies are available for only a few non-microbial species[1–4]. To address this issue, the international Genome 10K (G10K) consortium[5,6] has worked over a five-year period to evaluate and develop cost-effective methods for assembling highly accurate and nearly complete reference genomes. Here we present lessons learned from generating assemblies for 16 species that represent six major vertebrate lineages. We confirm that long-read sequencing technologies are essential for maximizing genome quality, and that unresolved complex repeats and haplotype heterozygosity are major sources of assembly error when not handled correctly. Our assemblies correct substantial errors, add missing sequence in some of the best historical reference genomes, and reveal biological discoveries. These include the identification of many false gene duplications, increases in gene sizes, chromosome rearrangements that are specific to lineages, a repeated independent chromosome breakpoint in bat genomes, and a canonical GC-rich pattern in protein-coding genes and their regulatory regions. Adopting these lessons, we have embarked on the Vertebrate Genomes Project (VGP), an international effort to generate high-quality, complete reference genomes for all of the roughly 70,000 extant vertebrate species and to help to enable a new era of discovery across the life sciences.

Chromosome-level reference genomes underpin the study of functional, comparative, and population genomics within and across species. The first high-quality genome assemblies of human[1] and other model species (for example, *Caenorhabditis elegans*[2], mouse[3], and zebrafish[4]) were put together using 500–1,000-base pair (bp) Sanger sequencing reads of thousands of hierarchically organized clones with 200–300-kilobase (kb) inserts, and chromosome genetic maps. This approach required tremendous manual effort, software engineering, and cost, in decade-long projects. Whole-genome shotgun approaches simplified the logistics (for example, in human[7] and *Drosophila*[8]), and later next-generation sequencing with shorter (30–150-bp) sequencing reads and short insert sizes (for example, 1 kb) ushered in more affordable and scalable genome sequencing[9]. However, the shorter reads resulted in lower-quality assemblies, fragmented into thousands of pieces, where many genes were missing, truncated, or incorrectly assembled, resulting in annotation and other errors[10]. Such errors can require months of manual effort to correct individual genes and years to correct an entire assembly. Genomic heterozygosity posed additional problems, because homologous haplotypes in a diploid or polyploid genome are forced together into a single consensus by standard assemblers, sometimes creating false gene duplications[11–14].

To address these problems, the G10K consortium[5,6] initiated the Vertebrate Genomes Project (VGP; https://vertebrategenomesproject.org) with the ultimate aim of producing at least one high-quality, near error-free and gapless, chromosome-level, haplotype-phased, and annotated reference genome assembly for each of the 71,657 extant named vertebrate species and using these genomes to address fundamental questions in biology, disease, and biodiversity conservation.

Towards this end, having learned the lessons of having too many variables that make conclusions more difficult to reach in the G10K from the G10K Assemblathon 2 effort[15], we first evaluated multiple genome sequencing and assembly approaches extensively on one species, the Anna's hummingbird (*Calypte anna*). We then deployed the best-performing method across sixteen species representing six major vertebrate classes, with a wide diversity of genomic characteristics. Drawing on the principles learned, we improved these methods further, discovered parameters and approaches that work better for species with different genomic characteristics, and made biological discoveries that had not been possible with the previous assemblies.

## Complete, accurate assemblies require long reads

We chose a female Anna's hummingbird because it has a relatively small genome (about 1 Gb), is heterogametic (has both Z and W sex chromosomes), and has an annotated reference of the same individual built from short reads[16]. We obtained 12 new sequencing data types, including both short and long reads (80 bp to 100 kb), and long-range linking information (40 kb to more than 100 Mb), generated using eight technologies (Supplementary Table 1). We benchmarked all technologies and assembly algorithms (Supplementary Table 2) in isolation and in many combinations (Supplementary Table 3). To our knowledge, this was the first systematic analysis of many sequence technologies, assembly algorithms, and assembly parameters applied on the same individual. We found that primary contiguous sequences (contigs) (pseudo-haplotype; Supplementary Note 1) assembled from Pacific Biosciences continuous long reads (CLR) or Oxford Nanopore long

reads (ONT) were approximately 30- to 300-fold longer than those assembled from Illumina short reads (SR), regardless of data type combination or assembly algorithm used (Fig. 1a, Supplementary Table 3). The highest contig NG50s for short-read-only assemblies were about 0.025 to 0.169 Mb, whereas for long reads they were about 4.6 to 7.66 Mb (Fig. 1a); contig NG50 is an assembly metric based on a weighted median of the lengths of its gapless sequences relative to the estimated genome size. After fixing a function in the PacBio FALCON software[17] that caused artificial breaks in contigs between stretches of highly homozygous and heterozygous haplotype sequences (Supplementary Note 1, Supplementary Table 2), contig NG50 nearly tripled to 12.77 Mb (Fig. 1a). These findings are consistent with theoretical predictions[18] and demonstrate that, given current sequencing technology and assembly algorithms, it is not possible to achieve high contig continuity with short reads alone, as it is typically impossible to bridge through repeats that are longer than the read length.

## Iterative assembly pipeline

Scaffolds generated with all three scaffolding technologies (that is, 10X Genomics linked reads (10XG), Bionano optical maps (Opt.), and Arima Genomics, Dovetail Genomics, or Phase Genomics Hi-C) were approximately 50% to 150% longer than those generated using one or two technologies, regardless of whether we started with short- or long-read-based contigs (Fig. 1b, Extended Data Fig. 1a, Supplementary Table 3). These findings include improvements we made to each approach (Supplementary Note 1, Supplementary Tables 4, 5, Supplementary Fig. 1). Despite similar scaffold continuity, the short-read-only assemblies had from about 18,000 to about 70,000 gaps, whereas the long-read assemblies had substantially fewer (about 400 to about 4,000) gaps (Fig. 1c). Many gaps in the short-read assemblies were in repeat or GC-rich regions. Considering the curated version of this assembly to be more accurate, we also identified roughly 5,000 to 8,000 mis-joins in short-read-based assemblies, whereas long-read-based assemblies had only from 20 to around 700 mis-joins (Fig. 1d). These mis-joins included chimeric joins and inversions. After we curated this assembly for contamination, assembly errors, and Hi-C-based chromosome assignments (Fig. 1e, f), the final hummingbird assembly had 33 scaffolds that closely matched the chromosome karyotype in number (33 of 36 autosomes plus sex chromosomes) and estimated sizes (approximately 2 to 200 Mb; Fig. 1g, h), with only 1 to 30 gaps per autosome (bCalAnn1 in Supplementary Table 6). Of the five autosomes with only one gap each, three (chromosomes 14, 15, and 19) had complete spanning support by at least two technologies (reliable blocks, Extended Data Fig. 1c; bCalAnn1 in Supplementary Table 6), indicating that the chromosome contigs were nearly complete. However, they were missing long arrays of vertebrate telomere repeats within 1 kb of their ends (Extended Data Fig. 1c; bCalAnn1 in Supplementary Tables 6, 7).

## Assembly pipeline across vertebrate diversity

Using the formula that gave the highest-quality hummingbird genome, we built an iterative VGP assembly pipeline (v1.0) with haplotype-separated CLR contigs, followed by scaffolding with linked reads, optical maps, and Hi-C, and then gap filling, base call polishing, and finally manual curation (Extended Data Figs. 2a, 3a). We systematically tested our pipeline on 15 additional species spanning all major vertebrate classes: mammals, birds, non-avian reptiles, amphibians, teleost fishes, and a cartilaginous fish (Supplementary Tables 8, 9, Supplementary Note 2). For the zebra finch, we used DNA from the same male as was used to generate the previous reference genome[19], and included a female trio for benchmarking haplotype completeness, where sequenced reads from the parents were used to bin parental haplotype reads from the offspring before assembly[20] (Extended Data

Figs. 2a, 3b). We set initial minimum assembly metric goals of: 1 Mb contig NG50; 10 Mb scaffold NG50; assigning 90% of the sequence to chromosomes, structurally validated by at least two independent lines of evidence; Q40 average base quality; and haplotypes assembled as completely and correctly as possible. When these metrics were achieved, most genes were assembled with gapless exon and intron structures[11], and fewer than 3% had frame-shift base errors identified in annotation. Q40 is the mathematical inflection point at which genes go from usually containing an error to usually not[21]. Of the curated assemblies (Supplementary Table 10, Supplementary Note 2), 16 of 17 achieved the desired continuity metrics (Extended Data Table 1). Scaffold NG50 was significantly correlated with genome size (Fig. 2a), suggesting that larger genomes tend to have larger chromosomes. On average, 98.3% of the assembled bases had reliable block NG50s ranging from 2.3 to 40.2 Mb; collapsed repeat bases[22] with abnormally high CLR read coverage (more than 3 s.d.) ranged from 0.7 to 31.4 Mb per Gb; and the completeness of the genome assemblies ranged from 87.2 to 98.1%, with less than 4.9% falsely duplicated regions, consistent with the false duplication rate we found for the conserved BUSCO vertebrate gene set (Extended Data Table 1, Supplementary Tables 11, 12).

## Repeats markedly affect continuity

For assemblies generated using our automated pipeline (Extended Data Fig. 3a) before manual curation, all but 2 (the thorny skate and channel bull blenny) of the 17 assemblies exceeded the desired continuity metrics (Supplementary Table 13). In searching for an explanation of these results, we found that contig NG50 decreased exponentially with increasing repeat content, with the thorny skate having the highest repeat content (Fig. 2b, Supplementary Table 13). Consequently, after scaffolding and gap filling, we observed a significant positive correlation between repeat content and number of gaps (Fig. 2c). The kākāpō parrot, which had 15% repeat content, had about 325 gaps per Gb, including 2 of 26 chromosomes with no gaps (chromosomes 16 and 18) and no evidence of collapses or low support, suggesting that the chromosomal contigs were complete (bStrHab1 in Supplementary Table 6). By contrast, the thorny skate, with 54% repeat content, had about 1,400 gaps per Gb (Extended Data Table 1); none of its 49 chromosomal-level scaffolds contained fewer than eight gaps, and all had some regions that contained collapses or low support (sAmbRad1 in Supplementary Table 6). Even after curation and other modifications to increase assembly quality (Supplementary Note 2), the number of collapses, their total size, missing bases, and the number of genes in the collapses all correlated with repeat content (Extended Data Fig. 4a–d). The average collapsed length, however, correlated with average CLR read lengths (10–35 kb; Extended Data Fig. 4e). There were no correlations between the number of collapsed bases and heterozygosity or genome size (Extended Data Fig. 4f, g). Depending on species, 77.4 to 99.2% of the collapsed regions consisted of unresolved segmental duplications (Extended Data Fig. 4h). The remainder were high-copy repeats, mostly of previously unknown types (Extended Data Fig. 4i), and of known types such as satellite arrays, simple repeats, long terminal repeats (LTRs), and short and long interspersed nuclear elements (SINEs and LINEs), depending on species (Extended Data Fig. 4j). We found that repeat masking before generating contigs prevented some repeats from making it into the final assembly (Supplementary Note 3). All of the above findings quantitatively demonstrate the effect that repeat content has on the ability to produce highly continuous and complete assemblies.

## Detection and removal of false duplications

During curation, we discovered that one of the most common assembly errors was the introduction of false duplications, which can be misinterpreted as exon, whole-gene, or large segmental duplications.

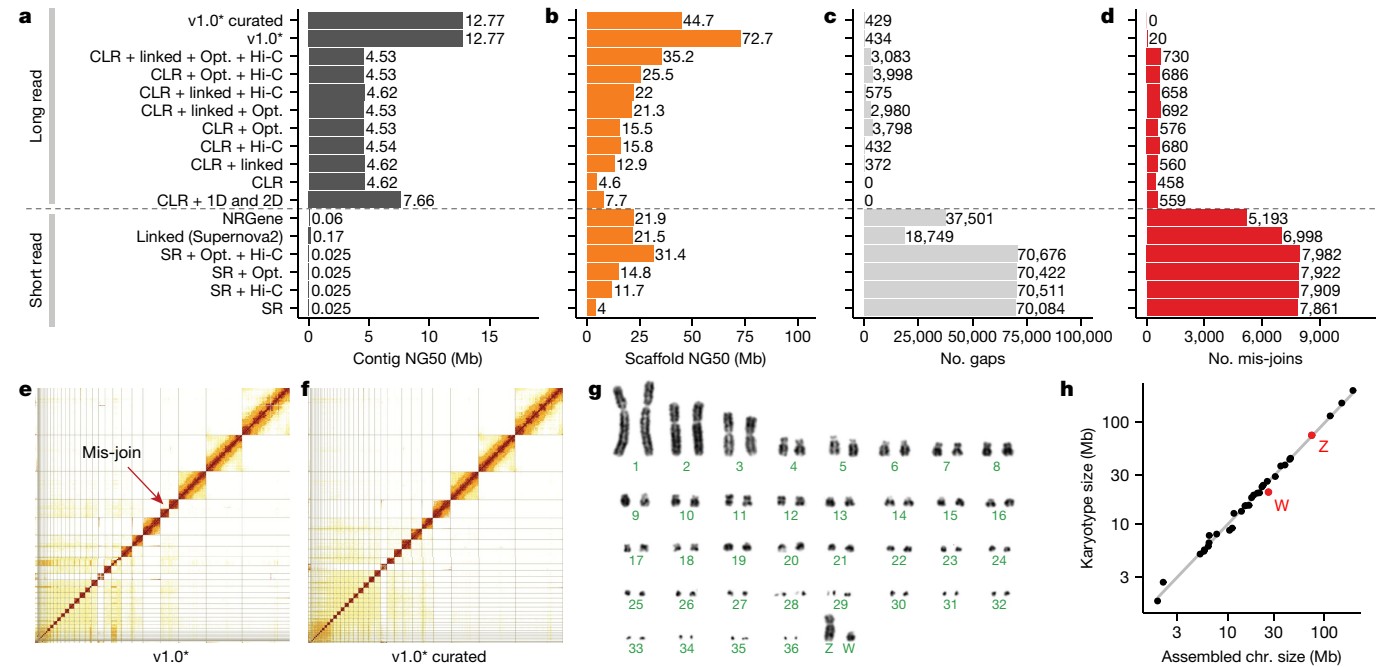

**Fig. 1 | Comparative analyses of Anna's hummingbird genome assemblies with various data types. a**, Contig NG50 values of the primary pseudo-haplotype. **b**, Scaffold NG50 values. **c**, Number of joins (gaps). **d**, Number of mis-join errors compared with the curated assembly. The curated assembly has no remaining conflicts with the raw data and thus no known mis-joins. *Same as CLR + linked + Opt. + Hi-C, but with contigs generated with an updated FALCON[17] version and earlier Hi-C Salsa version (v2.0 versus v2.2; Supplementary Table 2) for less aggressive contig joining. **e, f**, Hi-C interaction heat maps before and after manual curation, which identified

34 chromosomes. Grid lines indicate scaffold boundaries. Red arrow, example mis-join that was corrected during curation. **g**, Karyotype of the identified chromosomes ($n = 36 + ZW$), consistent with previous findings[70]. **h**, Correlation between estimated chromosome sizes (in Mb) based on karyotype images in **g** and assembled scaffolds in Supplementary Table 4 (bCalAna1) on a log–log scale. v1.0, VGP assembly v1.0 pipeline; linked, 10X Genomics linked reads; Hi-C, Hi-C proximity ligation; 1D, 2D, Oxford Nanopore long reads; NRGene, NRGene paired-end Illumina reads; SR, paired-end Illumina short reads.

We observed two types of false duplication: 1) heterotype duplications, which occurred in regions of increased sequence divergence between paternal and maternal haplotypes, where separate haplotype contigs were incorrectly placed in the primary assembly (Extended Data Fig. 5a); and 2) homotype duplications, which occurred near contig boundaries or under-collapsed sequences caused by sequencing errors (Extended Data Fig. 5b). False heterotype duplications appeared to occur with higher heterozygosity. For example, during curation of the female zebra finch genome, we found an approximately 1-Mb falsely duplicated heterozygous sequence (Extended Data Fig. 6a). This zebra finch individual had the highest heterozygosity (1.6%) relative to all other genomes (0.1–1.1%). Homotype duplications often occurred at contig boundaries, and were approximately the same length as the sequence reads (Extended Data Fig. 6b, c). We identified and removed false duplications during curation using read coverage, self-, transcript-, optical map- and Hi-C-alignments, and $k$-mer profiles (Extended Data Fig. 6, Supplementary Fig. 2).

Before we purged false duplications, the primary assembly genome size correlated positively with estimated percentage heterozygosity; more heterozygous genomes tended to have assembly sizes bigger than the estimated haploid genome size (Fig. 2d). Similarly, the extra duplication rate in the primary assembly, measured using $k$-mers[23] or conserved vertebrate BUSCO genes[24], varied from 0.3% to 30% and trended towards correlation with heterozygosity (Fig. 2e, f, Supplementary Table 13). Apparent false gene duplication rates correlated more strongly with the overall repeat rate in the assemblies (Fig. 2g, h). To remove these false duplications automatically, we initially used Purge_Haplotigs[13], which removed retained falsely duplicated contigs that were not scaffolded (Extended Data Fig. 5; VGP v1.0–1.5). Later, we developed Purge_Dups[14] to remove both falsely retained contigs and end-to-end duplicated contigs within scaffolds (Extended Data Fig. 5; VGP v1.6), which reduced the amount of manual curation. After

we applied these tools, the primary assembly sizes and the $k$-mer and BUSCO gene duplication rates were all reduced, and their correlations with heterozygosity and repeat content were also reduced or eliminated (Fig. 2d–h). These findings indicate that it is essential to properly phase haplotypes and to obtain high consensus sequence accuracy in order to prevent false duplications and associated biologically false conclusions.

## Curation is needed for a high-quality reference

Each automated scaffolding method introduced tens to thousands of unique joins and breaks in contigs or scaffolds (Supplementary Table 14). Depending on species, the first scaffolding step with linked reads introduced about 50–900 joins between CLR-generated contigs. Optical maps introduced a further roughly 30–3,500 joins, followed by Hi-C with about 30–700 more joins, and each identified up to several dozen joins that were inconsistent with the previous scaffolding step. Manual curation resulted in an additional 7,262 total interventions for 19 genome assemblies or 236 interventions per Gb of sequence (Supplementary Table 15). When a genome assembly was available for the same or a closely related species, it was used to confirm putative chromosomal breakpoints or rearrangements (Supplementary Table 15). These interventions indicate that even with current state-of-the-art assembly algorithms, curation is essential for completing high-quality reference assemblies and for providing iterative feedback to improve assembly algorithms. A further description of our curation approach and analyses of VGP genomes are presented elsewhere[25].

## Hi-C scaffolding and cytological mapping

Most large assembled scaffolds of each species spanned entire chromosomes, as shown by the relatively clean Hi-C heat map plots across each

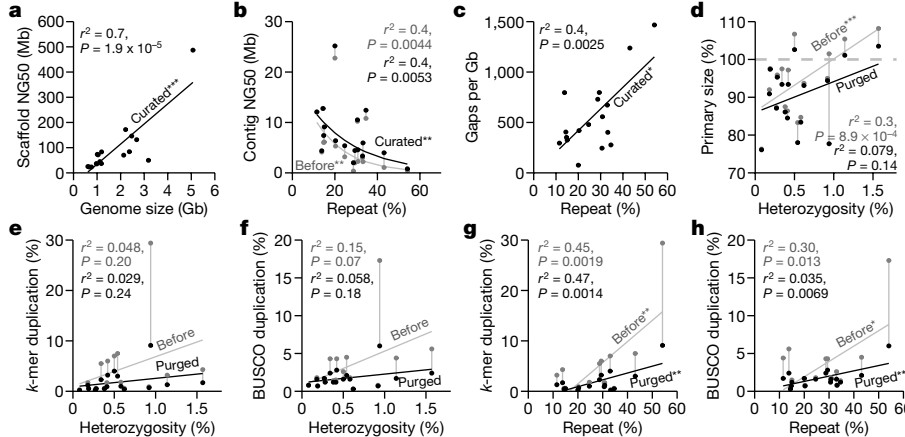

**Fig. 2 | Impact of repeats and heterozygosity on assembly quality.**
**a**, Correlation between scaffold NG50 and genome size of the curated assemblies. **b**, Nonlinear correlation between contig NG50 and repeat content, before and after curation. **c**, Correlation between number of gaps per Gb assembled and repeat content. **d**, Correlation between primary assembly size relative to estimated genome size (*y* axis) and genome heterozygosity (*x* axis), before and after purging of false duplications. Assembly sizes above 100% indicate the presence of false duplications and those below 100% indicate collapsed repeats. **e**, **f**, Correlations between genome duplication rate using *k*-mers[23] (**e**) and conserved BUSCO vertebrate gene set (**f**), and genome heterozygosity before and after purging of false duplications. **g**, **h**, As in **e**, **f**, but with whole-genome repeat content before and after purging of false duplications. Genome size, heterozygosity, and repeat content were estimated from 31-mer counts using GenomeScope[71], except for the channel bull blenny, as the estimates were unreliable (see Methods). Repeat content was measured by modelling the *k*-mer multiplicity from sequencing reads. Sequence duplication rates were estimated with Merqury[23] using 21-mers. *$P < 0.05$; **$P < 0.01$; ***$P < 0.001$, of the correlation coefficient: *P* values and adjusted $r^2$ from *F*-statistics. *n* = 17 assemblies of 16 species.

scaffold after curation (Extended Data Fig. 7), near perfect correlation between chromosomal scaffold length and karyotypically determined chromosome length (Fig. 1h), and the presence of telomeric repeat motifs on some scaffold ends (Supplementary Table 7). In our VGP zebra finch assembly, all inferred chromosomes were consistent with previously identified linkage groups in the Sanger-based reference, except for chromosomes 1 and 1B (Extended Data Fig. 8a). Their join in the VGP assembly was supported by both single CLR reads and optical maps through the junction. We also corrected nine inversion errors and filled in large gaps at some chromosome ends. In the platypus, we identified 18 structural differences in 13 scaffolds between the VGP assembly and the previous Sanger-based reference anchored to chromosomes using fluorescence in situ hybridization (FISH) physical mapping (Extended Data Fig. 8b, Supplementary Table 16). Of these 18, all were supported with Hi-C, and seven were also supported by both CLR and optical maps in the VGP assembly. Our platypus assembly also filled in many large (approximately 1–30 Mb) gaps and corrected many inversion errors (Extended Data Fig. 8b). Furthermore, we identified seven additional chromosomes (chromosomes 30–36) in the zebra finch, and eight (chromosomes 8, 9, 14, 15, 17, 19, 21, and X4; Extended Data Fig. 8a, b) in the platypus[26,27]. Relative to the VGP assembly, the earlier short-read Anna's hummingbird assembly was highly fragmented (Extended Data Fig. 8c), despite being scaffolded with seven different Illumina libraries spanning a wide range of insert sizes (0.2–20 kb). The previous climbing perch assembled chromosomes were even more fragmented and also had large gaps of missing sequence (Extended Data Fig. 8d). On average, 97% ± 3% (s.d.) of the assembled bases were assigned to chromosomes (Extended Data Table 1), compared with 76% and 32% in the prior zebra finch and platypus references, respectively. We believe the comparable or higher accuracy of Hi-C relative to genetic linkage or FISH physical mapping is due to the higher sampling rate of Hi-C pairs across the genome. Nonetheless, visual karyotyping is useful for complementary validation of chromosome count and structure[28].

## Trios help to resolve haplotypes

We were able to assemble the trio-based female zebra finch contigs into separate maternal and paternal chromosome-level scaffolds (Extended Data Fig. 9a) using our VGP trio pipeline (Extended Data Fig. 3b). Compared to the non-trio assembly of the same individual, the trio version had seven- to eightfold fewer false duplications (*k*-mer and BUSCO dups in Supplementary Tables 11, 12), well-preserved haplotype-specific variants (*k*-mer precision/recall 99.99/97.08%), and higher base call accuracy, exceeding Q43 for both haplotypes (Extended Data Table 1). The trio-based assembly was the only assembly with nearly perfect (99.99%) separation of maternal and paternal haplotypes, determined using *k*-mers specific to each[23]. We identified haplotype-specific structural variants, including inversions of 4.5 to 12.5 Mb on chromosomes 5, 11, and 13 that were not readily identifiable in the non-trio version (Extended Data Fig. 10a–e). Moving forward, the VGP is prioritising the collection of mother–father–offspring trios where possible, or single parent–offspring duos, to assist with diploid assembly and phasing, as well as the development of improved methods for the assembly of diploid genomes in the absence of parental genomic data, as described in another study[29].

## Effects of polishing on accuracy

Despite their increased continuity and structural accuracy, CLR-based assemblies required at least two rounds of short-read consensus polishing to reach 99.99% base-level accuracy (one error per 10 kb, Phred[30] Q40; Supplementary Table 5). Before polishing, the per-base accuracy was Q30–35 (calculated using *k*-mers). The most common errors were short indels from inaccurate consensus calling during CLR contig formation, which resulted in amino acid frameshift errors. Using our combined approach of long-read and short-read polishing applied on both primary and alternate haplotype sequences together, we polished from 82% to 99.7% of the primary and about 91.3% of the alternate assembly (Supplementary Table 17). Of the remaining unpolished sequence, one haplotype was sometimes reconstructed at substantially lower quality, because most reads aligned to the higher quality haplotype (Extended Data Fig. 11a). False duplications had similar effects, where the duplicated sequence acted as an attractor during the read mapping. Haplotypes in the more homozygous regions tended to be collapsed by FALCON-Unzip[17]. All such cases recruited reads from both haplotypes and thereby caused switch errors, which we confirmed in the trio-based assembly and fixed when excluding read pairs from the other haplotype

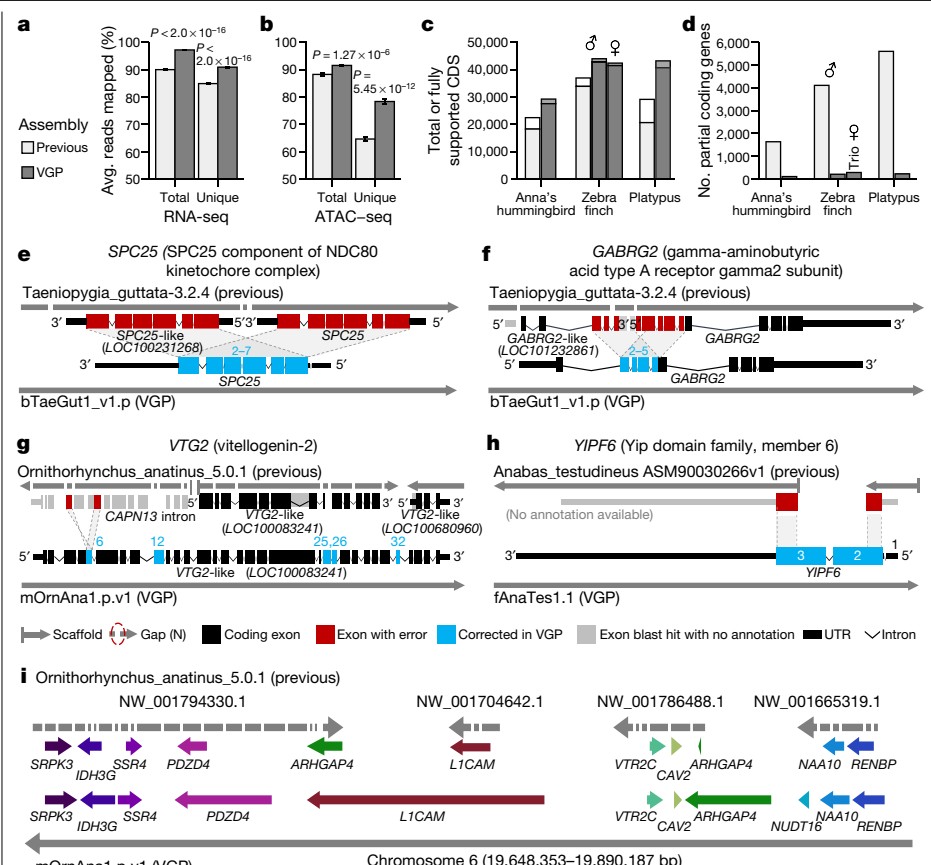

**Fig. 3 | Improvements to alignments and annotations in VGP assemblies relative to prior references. a**, **b**, Average percentage of RNA-seq transcriptome samples (**a**; *n* = 44, mean ± s.e.m.) and ATAC–seq genome reads (**b**; *n* = 12) that align to the previous and VGP zebra finch assemblies. Unique reads mapped to only one location in the assembly. Total is the sum of unique and multi-mapped reads. *P* values are from paired *t*-test. **c**, **d**, Total number of coding sequence (CDS) transcripts (full bar) and portion fully supported (inner bar) (**c**) and the number of RefSeq coding genes annotated as partial (**d**) in the previous and VGP assemblies using the same input data. **e–h**, Examples of assembly and associated annotation errors in previous reference assemblies corrected in the new VGP assemblies. See main text for descriptions. **i**, Gene synteny around the *VTR2C* receptor in the platypus shows completely missing genes (*NUDT16*), truncated and duplicated *ARHGAP4*, and many gaps in the earlier Sanger-based assembly compared with the filled in and expanded gene lengths in the new VGP assembly. Assembly accessions are in Supplementary Table 19.

during polishing (Extended Data Fig. 11b). These findings indicate that both sequence read accuracy and careful haplotype separation are important for producing accurate assemblies.

## Sex chromosomes and mitochondrial genomes

Sex chromosomes have been notoriously difficult to assemble, owing to their greater divergence relative to autosomes and high repeat content[31]. We successfully assembled both sex chromosomes (Z, W) for all three avian species, the first W chromosome (to our knowledge) for vocal learning birds (Extended Data Figs. 7, 9b), the X and/or Y chromosome in placental mammals (Canada lynx and two bat species), the X chromosome in the thorny skate, and for the first time, to our knowledge, all ten sex chromosomes (5X and 5Y) in the platypus[26] (Extended Data Fig. 9c). The completeness and continuity of the zebra finch Z and W chromosomes were further improved by the trio-based assembly (Extended Data Fig. 9b). However, the sex chromosome assemblies were still more fragmented than the autosomes, probably owing to their lower sequencing depth and high repeat content.

Mitochondrial (MT) genomes, which are expected to be 11–28 kb in size[32], were initially found in only six assemblies (Supplementary Table 18). The MT-derived raw reads were present, but they failed to assemble, in part because of minimum read-length cutoffs for the starting contig assembly. Furthermore, if the MT genome was not present during nuclear genome polishing, the raw MT reads were attracted to nuclear MT sequences (NuMTs), incorrectly converting them to the full organelle MT sequence (Extended Data Fig. 11c). To address these issues, we developed a reference-guided MT pipeline and included the MT genome during polishing[33] (Extended Data Fig. 3c; VGP v1.6). With these improvements, we reliably assembled 16 of 17 MT genomes (Supplementary Table 18) and discovered 2 kb of an 83-bp repeat expansion within the control region in the kākāpō (Extended Data Fig. 9d), and *Nad1* and *trnL2* gene duplications in the climbing perch (Extended

Data Fig. 9e). These duplications were verified using single-molecule CLR reads that spanned the duplication junctions or even the entire MT genome. Their absence in previous MT references[34,35] is likely to result from the inability of Sanger or short reads to correctly resolve large duplications. More details on the MT-VGP pipeline and new biological discoveries are reported elsewhere[33].

## Improvements to read alignment and annotation

Compared to previous Sanger (zebra finch and platypus) and Illumina (Anna's hummingbird and climbing perch) assemblies, we added about 42–176 Mb of missing sequence and placed 68.5 Mb (zebra finch) to 1.8 Gb (platypus) of previously unplaced sequence within chromosomes. We corrected about 7,800–64,000 mis-joins, and closed 55,177–193,137 gaps per genome (Supplementary Table 19). Consistent with these improvements, both transcriptome RNA sequencing (RNA-seq) data (Fig. 3a) and genome assay for transposase-accessible chromatin using sequencing (ATAC–seq) data (Fig. 3b) aligned with about 5 to 10% greater mapability to our new VGP assemblies compared with the previous assemblies. The NCBI RefSeq and EBI Ensembl annotations revealed: 5,434 to 14,073 more protein-coding transcripts per species, with 94.1 to 97.8% fully supported (Fig. 3c, Supplementary Table 20); only about 100 to 300 partially assembled coding genes, compared with about 1,600 to 5,600 (Fig. 3d); more orthologous coding genes shared with human; and fewer transcripts that required corrections to compensate for premature stop codons or frame-shift indel errors (Extended Data Table 2). The total number of genes annotated went down in the VGP assemblies (Extended Data Table 2), partly because there were fewer false duplications (Supplementary Table 19). Supporting these results, the VGP assemblies had 0 to 13% higher *k*-mer completeness (95% mean ± 3.5% s.d. versus 88 ± 4.3%; Extended Data Table 2, Supplementary Table 19; *P* = 0.0047, *n* = 4 prior and 17 VGP assemblies, unpaired *t*-test).

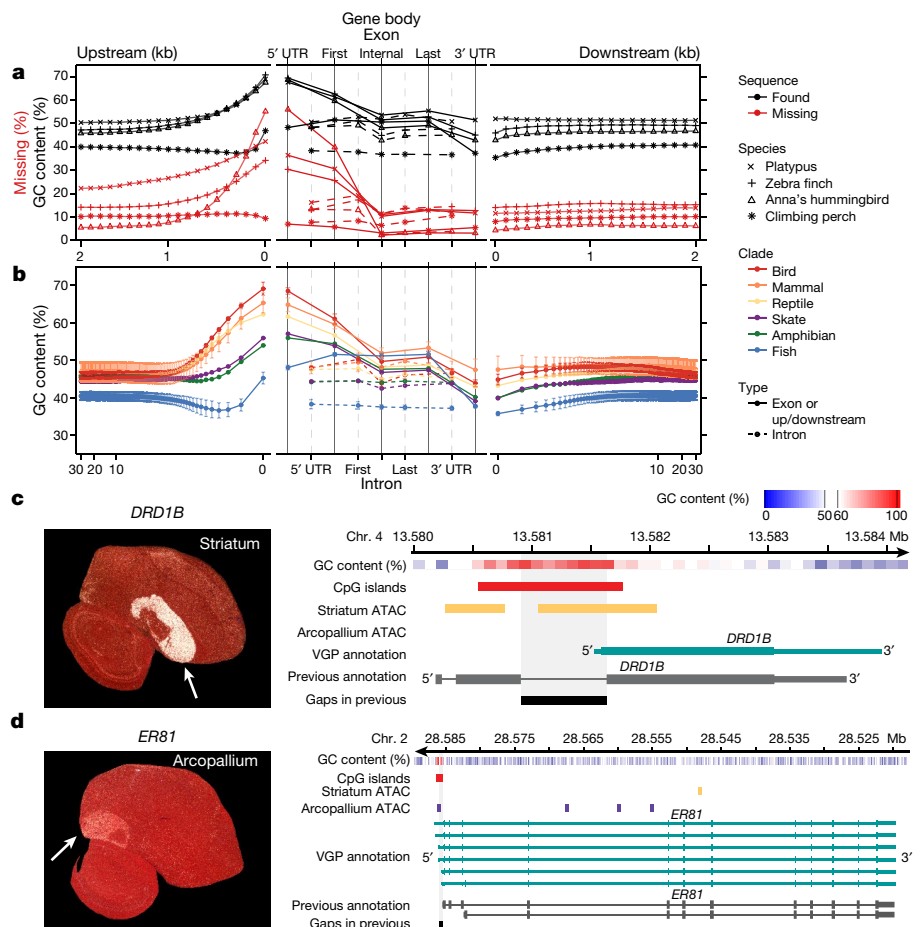

**Fig. 4 | VGP assemblies reveal GC content patterns in protein-coding genes. a**, Average GC content ($n = 14,000–18,000$ annotated coding genes; Extended Data Table 2) in VGP assemblies (black) and the percentage of genes with missing sequence in the earlier references (red) based on a Cactus alignment, in 100-bp blocks, 2 kb on either side of all protein-coding genes (left and right), and for UTRs, exons, and introns (middle). **b**, Average GC content (mean ± s.d. for lineages with more than one species) of the six major vertebrate lineages sequenced, for 30 kb upstream and downstream (in 100-bp blocks, log scale; left and right) and of the UTR, exons, and introns (middle). **c**, **d**, Left, specialized expression (arrows) shown by in situ hybridization of *DRD1B* in the zebra finch striatum (**c**) and *ER81* in the arcopallium (**d**), from Jarvis et al.[47]; the cerebellum was removed from the ER81 image. Right, ATAC–seq profiles in the GC-rich promoter regions of these genes, showing each gene's GC content (red is high), the ATAC–seq peaks in striatum (purple) or arcopallium (yellow) neurons, and portions of missing sequence (black) in the previous reference assembly (grey).

An example of a whole-gene heterotype false duplication in the RefSeq annotation of the previous zebra finch reference[19] is the BUSCO gene *SPC25*[36], for which each haplotype was correctly placed in the VGP primary and alternate assemblies (Fig. 3e). The *GABRG2* receptor, which shows specialized expression in vocal learning circuits[37], had a partial tandem duplication of four of its ten exons, resulting in annotated partial false tandem gene duplications (*GABRG2* and *GABRG2*-like; Fig. 3f). The vitellogenin-2 (*VTG2*) gene, a component of egg yolk in all egg-laying species[38], was distributed across 14 contigs in 3 different scaffolds in the previous platypus assembly (Fig. 3g). Two of these scaffolds received two corresponding *VTG2*-like gene annotations, and the third was included as false duplicated intron in *CAPN-13* (red), together causing false amino acid sequences in five exons (blue). The BUSCO *YIPF6* gene, which is associated with inflammatory bowel disease[39], was split between two different scaffolds and is thus presumed to be a gene loss in the earlier climbing perch assembly[40] (Fig. 3h). Each of these genes is now present on long VGP contigs, within validated blocks, with no gaps and no false gene gains or losses (Supplementary Table 21).

Going beyond individual genes, a ten-gene synteny window surrounding the vasotocin receptor 2C gene (*VTR2C*; also known as *AVPR2*), which is involved in blood pressure homeostasis and brain function[41,42], was split into 34 contigs on four scaffolds, one of which contained a false haplotype duplication of *ARHGAP4* in the previous platypus assembly[43] (Fig. 3i). In our VGP assembly, all eleven genes were in one 37-Mb-long contig within the approximately 50 Mb chromosome 6 scaffold. Furthermore, eight of the eleven genes were remarkably increased in size owing to the addition of previously unknown missing sequences. This chromosomal region was more GC-rich (54%) than the entire chromosome 6 (46%). Thousands of such false gains and losses in previous reference assemblies have been corrected in our VGP assemblies (more

details in refs. [27,44]), demonstrating that assembly quality has a critical effect on subsequent annotations and functional genomics.

## GC-rich regulatory regions of coding genes

We tested whether the higher-quality VGP assemblies enabled new biological discoveries. Notably, beginning about 1.5 kb upstream of protein-coding genes, in 100-bp blocks, there was a steady increase from about 6–20% to about 30–55% of genes having missing sequence in previous references (Fig. 4a); similarly high proportions of genes were missing their subsequent 5′ untranslated regions (UTRs) and first exons. This fluctuation in missing sequence was directly proportional to GC content (Fig. 4a). We therefore studied the GC content pattern across all protein-coding genes in all 16 new VGP assemblies and found a genome-wide signature: a rapid rise in GC content in the roughly 1.5 kb before the transcription start site, in the 5′ UTR, and in the first exon, followed by a steady decrease in subsequent exons and returning to near intergenic background levels in the 3′ UTR and about 1.5 kb after the transcription termination site (Fig. 4b). The introns had lower GC content, closer to the intergenic background. The intergenic GC content was stable within 30 kb on either side of each gene (Fig. 4b). Mammals, birds, and reptiles had the highest increase (around 20%) in GC content near the start site, followed by the amphibian and skate with medium levels (around 10%). Teleost fishes showed an initial decrease, followed by weaker increase (about 5%) from an already lower GC content (Fig. 4b). Given that the skate represents the sister branch to all other vertebrate lineages sequenced, these findings suggest that teleosts lost at least 5% GC content genome-wide, while maintaining most of the GC content pattern in protein-coding genes. Although it is known that promoter regions can be CpG rich, and GC content can vary between

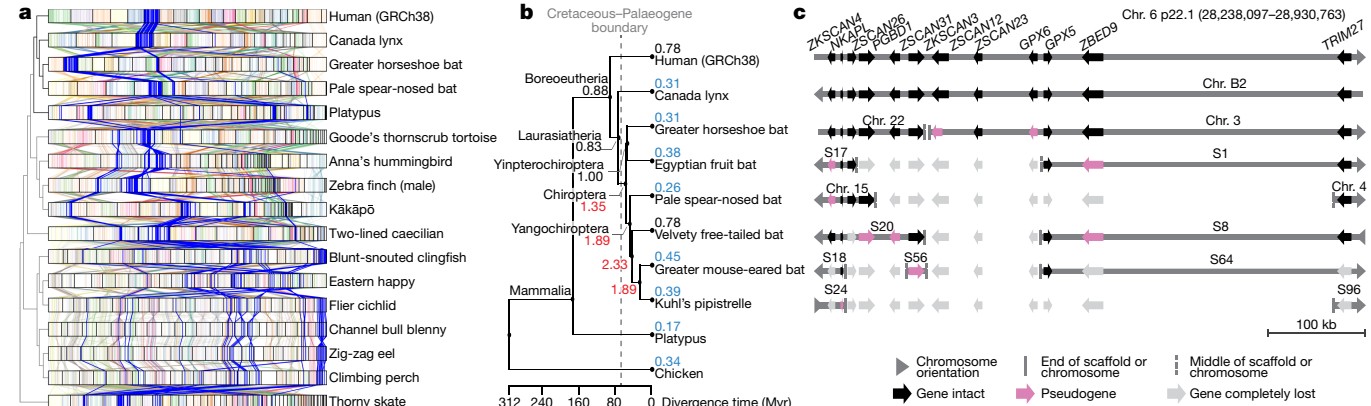

**Fig. 5 | Chromosome evolution among bats and other vertebrates.**
**a**, Chromosome synteny maps across the species sequenced based on BUSCO gene alignments. Chromosome sizes (bar lengths) are normalized to genome size, to make visualization easier. Genes (lines) are coloured according to the human chromosome to which they belong; those on human chromosome 6 are highlighted in blue and other chromosomes are in lighter shades. The cladogram is from the TimeTree database[72]. **b**, Phylogenetic relationship of the mammalian species sequenced and their inferred chromosome EBR rates

(breaks per Myr) on different branches. Red, higher rates than average (0.84); blue, lower than average. **c**, Summary of alignment, gene organization, and functional gene status surrounding a bat interchromosomal EBR involving the homologue of human chromosome 6. End of scaffold (S) or chromosome (Chr.) means that the breakpoint is located at a chromosome arm end; middle means that it is located within a scaffold or chromosome. Scale is relevant for human Chr. 6 only. Actual gene sizes in the non-human species may differ and were drawn to match the annotated human gene sizes for simplicity.

exons and introns[45,46], such a systematic pattern, the lineage-specific differences within vertebrates, and the magnitude of these differences had not been previously described, to our knowledge.

We tested whether the newly assembled GC-rich promoter regions contained novel regulatory sequences. Analysing the zebra finch brain, we found that genes with upregulated expression specific to the striatum (for example, *DRD1B*, which encodes a dopamine receptor) had ATAC-seq peaks in the GC-rich promoter and 5′ UTR region in striatal neurons, but not in arcopallium neurons (Fig. 4c); conversely, genes (for example, the *ER81* transcription factor) with upregulated expression in the arcopallium (mammalian cortex layer 5 equivalent[47]) had ATAC–seq peaks in the GC-rich region in arcopallium neurons but not striatal neurons (Fig. 4d). These GC-rich regions were missing in the earlier assembly. In addition, the missing region in *DRD1B* led to a false annotation as a two-exon gene[48], whereas the VGP assembly revealed a single-exon gene (Fig. 4c). These GC-rich promoter regions are candidates for driving cell-type-specific expression. These findings demonstrate the importance of using sequencing chemistry that reads through GC-rich regions, like the CLR method. The earlier hummingbird genome assembly was generated using Illumina TruSeq3 chemistry[16], which was designed to read through GC-rich regions, and yet about 55% of the genes were missing the 100-bp GC-rich region before the start site (Fig. 4a). Another paper contains additional findings on missing regions[27].

## Chromosomal evolution

We next investigated whether we could gain new insights into chromosome evolution among vertebrates. Given the more than 430 million years (Myr) of evolutionary divergence among the species sampled here, it was difficult to generate whole genome-to-genome alignments across all species. Thus, we focused our initial analyses on 1,147 highly conserved BUSCO vertebrate genes that are shared among our assemblies of all 16 species and the human reference (GRCh38). Human chromosomes mapped with greater orthology to 3.7 ± 1.3 (s.d.) chromosomes on average in other mammals, compared to 5.6 ± 2.2 in amphibians and 9.6 ± 3.3 in teleost fishes (Fig. 5a, Supplementary Table 22). The skate chromosome arrangement was more conserved with tetrapods, mapping to 2.9 ± 1.4 chromosomes on average, compared to 4.8 ± 2.5 in teleost fishes. These findings indicate that, along with a reduction in GC content, the teleost lineage has experienced

more massive chromosome rearrangements since divergence from their most recent common ancestor with tetrapods, consistent with a proposed higher rearrangement rate in teleosts[49].

To determine the precise locations of chromosome rearrangements between species, we focused on a shorter evolutionary distance of around 180 Myr among mammals, and added four additional bat species described in our Bat1K study[50], the human genome reference[51] (GRCh38.p12), and a recently upgraded long-read chicken reference[52] (galGal6a) as an outgroup. Pairwise whole-genome alignments to the human reference defined homologous synteny blocks and evolutionary breakpoint regions (EBRs) among the species. We found that breakpoint rates (EBRs per Myr) tripled among bats soon after the last mass extinction event (about 66 million years ago (Mya)), a time of rapid bat superfamily divergences[53] (about 60 Mya; Fig. 5b). Some rearrangements affected genes. For example, a 1.3-Mb inversion in greater horseshoe bat chromosome 28 (homologous to 29.5 Mb of human chromosome 15; Extended Data Fig. 12a) disrupted *STARD5*, a gene involved in cholesterol homeostasis in liver cells[54]. The rearrangement separated exons 1–5 from exon 6, and disrupted splicing of the transcripts (Extended Data Fig. 12b). Another example was an EBR that involved fission of an ancestral bat chromosome homologue of human chromosome 6 (boreoeutherian mammal chromosome 5[55]) and was later reused among the different bat lineages in rearrangements that involved the ancestral homologues of human chromosomes 1, 2 and 6 (Fig. 5c, Extended Data Fig. 12c). We also noted a fission in this region in the mouse, rat, and dog genomes[55]. On the basis of the conserved gene order in human and Canada lynx, we inferred that the boreoeutherian ancestral mammal locus corresponding to human 6p22.1 contained 12 genes, including four *ZSCAN* and two *ZKSCAN* transcription factors, and two *GPX* enzyme genes, all associated with sequentially increasing independent gene losses in bats (Fig. 5d). For example, the greater horseshoe bat lost only *ZSCAN12* and *GPX6* to pseudogenization, whereas Kuhl's pipistrelle lost all 12 genes. *ZSCAN* and *ZKSCAN* are involved in cell differentiation, migration and invasion, proliferation, apoptosis, and innate immunity[56]. We speculate that loss of *ZSCAN12* in all six bats could contribute to their immune tolerance to pathogens[50].

Other biological findings using these VGP assemblies are published elsewhere, and include: 1) more accurate synteny across species, leading to a better understanding of the evolution of and thus a universal nomenclature for the vasotocin (also known as vasopressin) and

**Table 1 | Proposed standards and metrics for defining genome assembly quality**

| Quality category | Metric | Finished | VGP-2020 | VGP-2016 | B10k-2014 | This study |
|---|---|---|---|---|---|---|
| Notation | *x.y.P.Q.C* | c.c.Pc.Q60.C100 | 7.c.P6.Q50.C95 | 6.7.P5.Q40.C90 | 4.5.Q30 | |
| Continuity | **Contig NG50 (*x*)** | = Chr. NG50 | >10 Mb | >1 Mb | >10 kb | 1–25 Mb |
| | **Scaffolds NG50 (*y*)** | = Chr. NG50 | = Chr. NG50 | >10 Mb | >100 kb | 23–480 Mb |
| | **Gaps per Gb** | No gaps | <200 | <1,000 | <10,000 | 75–1,500 |
| Structural accuracy | **Reliable blocks** | = Chr. NG50 | >10 Mb | >1 Mb | Not required | 2.3–40.2 Mb |
| | **False duplications** | 0% | <1% | <5% | <10% | 0.2–5.0% |
| | **Curation** | Conflicts resolved | Manual | Manual | Not required | Manual |
| Base accuracy | **Base pair QV (*Q*)** | >60 | >50 | >40 | >30 | 39–43 |
| | ***k*-mer completeness** | 100% complete | >95% | >90% | >80% | 87–98% |
| Haplotype phasing | **Phase block NG50 (*P*)** | = Chr. NG50 | >1 Mb | >100 kb | Not required | 1.6 Mb[a] |
| Functional completeness | **Genes** | >98% complete | >95% complete | >90% | >80% | 82–98% |
| | **Transcript mappability** | >98% | >90% | >80% | >70% | 96% |
| Chromosome status | **Assigned (*C*)** | >100% | >95% | >90% | Not required | 94.4–99.9% |
| | **Sex chromosomes** | Right order, no gaps | Localized homo pairs | At least one shared (for example, X or Z) | Fragmented | At least one shared |
| | **Organelles (for example, MT)** | One complete allele | One complete allele | Fragmented | Not required | One complete allele |

The six broad quality categories in the first column are split into sub-metrics in the second column. The recommendations for draft to finished qualities (columns 3–6) are based on those achieved in past studies[16,19,63], this study, and what we aspire to. In the *x.y.P.Q.C* notation, $x = \log_{10}$[contig NG50]; $y = \log_{10}$[scaffold NG50]; $P = \log_{10}$[haplotype phased NG50 block]; $Q$ = Phred base accuracy QV; and $C$ = percentage of the assembly assigned to chromosomes. c denotes 'complete' telomere-to-telomere continuity. The VGP assemblies (last column) satisfy the 6.7.6.Q40.C90 standard, but some come close to achieving a higher 7.c.7.Q50.C95 standard. These metrics apply to genomes about 1 Gb or bigger.
[a]Phase blocks calculated for the zebra finch non-trio assembly using haplotype specific *k*-mers from parental data[20]; the trio assemblies had NG50 phase blocks of 17.3 Mb (maternal) and 56.6 Mb (paternal).

oxytocin ligand and receptor gene families[57]; 2) greater understanding of the evolution of the carbohydrate 6-*O* sulfotransferase gene family, which encodes enzymes that modify secreted carbohydrates[58]; 3) the first Bat1K study[50], which generated a genome-scale phylogeny that better resolves the relationships between bats and other mammals, and which identified changes in bat genes that are involved in immunity and life span, including genes that are relevant to the COVID-19 pandemic[59]; 4) deleterious mutations that have been purged from the last surviving isolated and inbred population of the critically endangered kākāpō[60]; and 5) more complete resolution of the evolution of the complex sex chromosomes in platypus and echidna[26]. These discoveries were not possible with the previous reference assemblies, and we expect many future discoveries to follow.

## Proposed assembly quality metrics

Drawing on the lessons learned from this work, we propose that assembly quality should be summarized using 14 metrics under 6 categories (Table 1; full details in Supplementary Note 4). We summarize the most critical and commonly used metrics using the simple notation *x.y.P.Q.C*, where: $x = \log_{10}$[contig NG50], $y = \log_{10}$[scaffold NG50], $P = \log_{10}$[haplotype phase block NG50], $Q$ = QV base accuracy, and $C$ = percentage of the assembly assigned to chromosomes (Table 1). Our current minimum VGP standard, for example, is 6.7.P5.Q40.C90. This revises our prior notation[50,61,62], which reported log-scaled continuity measured in 'kilobases' rather than 'bases'. The thresholds we chose were based on empirical and quantitative observations between what is achievable currently and what is aspirational, and the question the assemblies are meant to answer. For example, the short-read paired-end library-based assemblies of the B10K Phase 1 genomes in 2014[16] and the 10XG linked-read assembly of the Anna's hummingbird presented here would be categorized as a 4.5.P7.Q50 assembly, with low continuity but high base accuracy (Table 1). Such a genome would be suitable for use in phylogenomics[63] and for population-scale SNP surveys[64]. If, instead, a genome is to be used to

study chromosomal evolution, then the VGP-2016 minimum metric 6.7.P5.Q40.C95, with high structural and base accuracies and more than 95% assigned to chromosomes (Table 1), would be necessary. If having GC-rich promoter regions and complete 5′ exons in most genes is essential, then long-read approaches that sequence through these regions are necessary. 'Finished' quality (Table 1) is obviously the ideal assembly result, but this level of quality is currently routine only for bacterial and non-vertebrate model organisms with smaller genome sizes that lack large centromeric satellite arrays[65–67] and for organelle genomes, as presented here[33]. The possibility of achieving complete, telomere-to-telomere assemblies of vertebrate and other eukaryotic species is foreseeable, given some assembled avian and bat chromosomes with zero gaps in this study, and the recent complete assembly of two human chromosomes[68,69].

## The Vertebrate Genomes Project

Building on this initial set of assembled genomes and the lessons learned, we propose to expand the VGP to deeper taxonomic phases, beginning with phase 1: representatives of approximately 260 vertebrate orders, defined here as lineages separated by 50 million or more years of divergence from each other. Phase 2 will encompass species that represent all approximately 1,000 vertebrate families; phase 3, all roughly 10,000 genera; and phase 4, nearly all 71,657 extant named vertebrate species (Supplementary Note 5, Supplementary Fig. 3). To accomplish such a project within 10 years, we will need to scale up to completing 125 genomes per week, without sacrificing quality. This includes sample permitting, high molecular weight DNA extractions, sequencing, meta-data tracking, and computational infrastructure. We will take advantage of continuing improvements in genome sequencing technology, assembly, and annotation, including advances in PacBio HiFi reads, Oxford Nanopore reads, and replacements for 10XG reads (Supplementary Note 6), while addressing specific scientific questions at increasing levels of phylogenetic refinement. Genomic technology advances quickly, but we believe the principles of our pipeline and

the lessons learned will be applicable to future efforts. Areas in which improvement is needed include more accurate and complete haplotype phasing, base-call accuracy, and resolution of long repetitive regions such as telomeres, centromeres, and sex chromosomes. The VGP is working towards these goals and making all data, protocols, and pipelines openly available (Supplementary Notes 5, 7).

Despite remaining imperfections, our reference genomes are the most complete and highest quality to date for each species sequenced, to our knowledge. When we began to generate genomes beyond the Anna's hummingbird in 2017, only eight vertebrate species in GenBank had genomes that met our target continuity metrics, and none were haplotype phased (Supplementary Table 23). The VGP pipeline introduced here has now been used to complete assemblies of more than 130 species of similar or higher quality (Supplementary Note 5; BioProject PRJNA489243). We encourage the scientific community to use and evaluate the assemblies and associated raw data, and to provide feedback towards improving all processes for complete and error-free assembled genomes of all species.

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

**Arang Rhie**[1,103], **Shane A. McCarthy**[2,3,103], **Olivier Fedrigo**[4,103], **Joana Damas**[5], **Giulio Formenti**[4,6], **Sergey Koren**[1], **Marcela Uliano-Silva**[7,8], **William Chow**[3], **Arkarachai Fungtammasan**[9], **Juwan Kim**[10], **Chul Lee**[10], **Byung June Ko**[11], **Mark Chaisson**[12], **Gregory L. Gedman**[6], **Lindsey J. Cantin**[6], **Francoise Thibaud-Nissen**[13], **Leanne Haggerty**[14], **Iliana Bista**[2,3], **Michelle Smith**[3], **Bettina Haase**[4], **Jacquelyn Mountcastle**[4], **Sylke Winkler**[15,16], **Sadye Paez**[4,6], **Jason Howard**[17], **Sonja C. Vernes**[18,19,20], **Tanya M. Lama**[21], **Frank Grutzner**[22], **Wesley C. Warren**[23], **Christopher N. Balakrishnan**[24], **Dave Burt**[25], **Julia M. George**[26], **Matthew T. Biegler**[6], **David Iorns**[27], **Andrew Digby**[28], **Daryl Eason**[28], **Bruce Robertson**[29], **Taylor Edwards**[30], **Mark Wilkinson**[31], **George Turner**[32], **Axel Meyer**[33], **Andreas F. Kautt**[33,34], **Paolo Franchini**[33], **H. William Detrich III**[35], **Hannes Svardal**[36,37], **Maximilian Wagner**[38], **Gavin J. P. Naylor**[39], **Martin Pippel**[15,40], **Milan Malinsky**[3,41], **Mark Mooney**[42], **Maria Simbirsky**[9], **Brett T. Hannigan**[9], **Trevor Pesout**[43], **Marlys Houck**[44], **Ann Misuraca**[44], **Sarah B. Kingan**[45], **Richard Hall**[45], **Zev Kronenberg**[45], **Ivan Sović**[45,46], **Christopher Dunn**[45], **Zemin Ning**[3], **Alex Hastie**[47], **Joyce Lee**[47], **Siddarth Selvaraj**[48], **Richard E. Green**[43,49], **Nicholas H. Putnam**[50], **Ivo Gut**[51,52], **Jay Ghurye**[49,53], **Erik Garrison**[43], **Ying Sims**[3], **Joanna Collins**[3], **Sarah Pelan**[3], **James Torrance**[3], **Alan Tracey**[3], **Jonathan Wood**[3], **Robel E. Dagnew**[12], **Dengfeng Guan**[2,54], **Sarah E. London**[55], **David F. Clayton**[56], **Claudio V. Mello**[57], **Samantha R. Friedrich**[57], **Peter V. Lovell**[57], **Ekaterina Osipova**[15,40,58], **Farooq O. Al-Ajli**[59,60,61], **Simona Secomandi**[62], **Heebal Kim**[10,11,63], **Constantina Theofanopoulou**[6], **Michael Hiller**[64,65,66], **Yang Zhou**[67], **Robert S. Harris**[68], **Kateryna D. Makova**[68,69,70], **Paul Medvedev**[69,70,71,72], **Jinna Hoffman**[13], **Patrick Masterson**[13], **Karen Clark**[13], **Fergal Martin**[14], **Kevin Howe**[14], **Paul Flicek**[14], **Brian P. Walenz**[1], **Woori Kwak**[63,73], **Hiram Clawson**[43], **Mark Diekhans**[43], **Luis Nassar**[43], **Benedict Paten**[43], **Robert H. S. Kraus**[33,74], **Andrew J. Crawford**[75], **M. Thomas P. Gilbert**[76,77], **Guojie Zhang**[78,79,80,81], **Byrappa Venkatesh**[82], **Robert W. Murphy**[83], **Klaus-Peter Koepfli**[84], **Beth Shapiro**[85,86], **Warren E. Johnson**[84,87,88], **Federica Di Palma**[89], **Tomas Marques-Bonet**[90,91,92,93], **Emma C. Teeling**[94], **Tandy Warnow**[95], **Jennifer Marshall Graves**[96], **Oliver A. Ryder**[44,97], **David Haussler**[43,85], **Stephen J. O'Brien**[98,99], **Jonas Korlach**[45], **Harris A. Lewin**[5,100,101], **Kerstin Howe**[3,104]✉, **Eugene W. Myers**[15,40,102,104]✉, **Richard Durbin**[2,3,104]✉, **Adam M. Phillippy**[1,104] & **Erich D. Jarvis**[4,6,86,104]✉

[1]Genome Informatics Section, Computational and Statistical Genomics Branch, National Human Genome Research Institute, National Institutes of Health, Bethesda, MD, USA. [2]Department of Genetics, University of Cambridge, Cambridge, UK. [3]Wellcome Sanger Institute, Cambridge, UK. [4]Vertebrate Genome Lab, The Rockefeller University, New York, NY, USA. [5]The Genome Center, University of California Davis, Davis, CA, USA. [6]Laboratory of Neurogenetics of Language, The Rockefeller University, New York, NY, USA. [7]Leibniz Institute for Zoo and Wildlife Research, Department of Evolutionary Genetics, Berlin, Germany. [8]Berlin Center for Genomics in Biodiversity Research, Berlin, Germany. [9]DNAnexus Inc., Mountain View, CA, USA. [10]Interdisciplinary Program in Bioinformatics, Seoul National University, Seoul, Republic of Korea. [11]Department of Agricultural Biotechnology and Research Institute of Agriculture and Life Sciences, Seoul National University, Seoul, Republic of Korea. [12]University of Southern California, Los Angeles, CA, USA. [13]National Center for Biotechnology Information, National Library of Medicine, NIH, Bethesda, MD, USA. [14]European Molecular Biology Laboratory, European Bioinformatics Institute, Wellcome Genome Campus, Hinxton, UK. [15]Max Planck Institute of Molecular Cell Biology and Genetics, Dresden, Germany. [16]DRESDEN-concept Genome Center, Dresden, Germany. [17]Novogene, Durham, NC, USA. [18]Neurogenetics of Vocal Communication Group, Max Planck Institute for Psycholinguistics, Nijmegen, The Netherlands. [19]Donders Institute for Brain, Cognition and Behaviour, Nijmegen, The Netherlands. [20]School of Biology, University of St Andrews, St Andrews, UK. [21]University of Massachusetts Cooperative Fish and Wildlife Research Unit, Amherst, MA, USA. [22]School of Biological Science, The Environment Institute, University of Adelaide, Adelaide, South Australia, Australia. [23]Bond Life Sciences Center, University of Missouri, Columbia, MO, USA. [24]Department of Biology, East Carolina University, Greenville, NC, USA. [25]UQ Genomics, University of Queensland, Brisbane, Queensland, Australia. [26]Department of Biological Sciences, Clemson University, Clemson, SC, USA. [27]The Genetic Rescue Foundation, Wellington, New Zealand. [28]Kākāpō Recovery, Department of Conservation, Invercargill, New Zealand. [29]Department of Zoology, University of Otago, Dunedin, New Zealand. [30]University of Arizona Genetics Core, Tucson, AZ, USA. [31]Department of Life Sciences, Natural History Museum, London, UK. [32]School of Natural Sciences, Bangor University, Gwynedd, UK. [33]Department of Biology, University of Konstanz, Konstanz, Germany. [34]Department of Organismic and Evolutionary Biology, Harvard University, Cambridge, MA, USA. [35]Department of Marine and Environmental Sciences, Northeastern University Marine Science Center, Nahant, MA, USA. [36]Department of Biology, University of Antwerp, Antwerp, Belgium. [37]Naturalis Biodiversity Center, Leiden, The Netherlands. [38]Institute of Biology, Karl-Franzens University of Graz, Graz, Austria. [39]Florida Museum of Natural History, University of Florida, Gainesville, FL, USA. [40]Center for Systems Biology, Dresden, Germany. [41]Zoological Institute, University of Basel, Basel, Switzerland. [42]Tag.bio, San Francisco, CA, USA. [43]UC Santa Cruz Genomics Institute, University of California, Santa Cruz, CA, USA. [44]San Diego Zoo Global, Escondido, CA, USA. [45]Pacific Biosciences, Menlo Park, CA, USA. [46]Digital BioLogic, Ivanić-Grad, Croatia. [47]Bionano Genomics, San Diego, CA, USA. [48]Arima Genomics, San Diego, CA, USA. [49]Dovetail Genomics, Santa Cruz, CA, USA. [50]Independent Researcher, Santa Cruz, CA, USA. [51]CNAG-CRG, Centre for Genomic Regulation, Barcelona Institute of Science and Technology, Barcelona, Spain. [52]Universitat Pompeu Fabra, Barcelona, Spain. [53]Department of Computer Science, University of Maryland College Park, College Park, MD, USA. [54]School of Computer Science and Technology, Center for Bioinformatics, Harbin Institute of Technology, Harbin, China. [55]Department of Psychology, Institute for Mind and Biology, University of Chicago, Chicago, IL, USA. [56]Department of Genetics and Biochemistry, Clemson University, Clemson, SC, USA. [57]Department of Behavioral Neuroscience, Oregon Health and Science University, Portland, OR, USA. [58]Max Planck Institute for the Physics of Complex Systems, Dresden, Germany. [59]Monash University Malaysia Genomics Facility, School of Science, Selangor Darul Ehsan, Malaysia. [60]Tropical Medicine and Biology Multidisciplinary Platform, Monash University Malaysia, Selangor Darul Ehsan, Malaysia. [61]Qatar Falcon Genome Project, Doha, Qatar. [62]Department of Biosciences, University of Milan, Milan, Italy. [63]eGnome, Inc., Seoul, Republic of Korea. [64]LOEWE Centre for Translational Biodiversity Genomics, Frankfurt, Germany. [65]Senckenberg Research Institute, Frankfurt, Germany. [66]Goethe-University, Faculty of Biosciences, Frankfurt, Germany. [67]BGI-Shenzhen, Shenzhen, China. [68]Department of Biology, Pennsylvania State University, University Park, PA, USA. [69]Center for Medical Genomics, Pennsylvania State University, University Park, PA, USA. [70]Center for Computational Biology and Bioinformatics, Pennsylvania State University, University Park, PA, USA. [71]Department of Computer Science and Engineering, Pennsylvania State University, University Park, PA, USA. [72]Department of Biochemistry and Molecular Biology, Pennsylvania State University, University Park, PA, USA. [73]Hoonygen, Seoul, Korea. [74]Department of Migration, Max Planck Institute of Animal Behavior, Radolfzell, Germany. [75]Department of Biological Sciences, Universidad de los Andes, Bogotá, Colombia. [76]Center for Evolutionary Hologenomics, The GLOBE Institute, University of Copenhagen, Copenhagen, Denmark. [77]University Museum, NTNU, Trondheim, Norway. [78]China National Genebank, BGI-Shenzhen, Shenzhen, China. [79]Villum Center for Biodiversity Genomics, Section for Ecology and Evolution, Department of Biology, University of Copenhagen, Copenhagen, Denmark. [80]State Key Laboratory of Genetic Resources and Evolution, Kunming Institute of Zoology, Chinese Academy of Sciences, Kunming, China. [81]Center for Excellence in Animal Evolution and Genetics, Chinese Academy of Sciences, Kunming, China. [82]Institute of Molecular and Cell Biology, A*STAR, Biopolis, Singapore, Singapore. [83]Centre for Biodiversity, Royal Ontario Museum, Toronto, Ontario, Canada. [84]Smithsonian Conservation Biology Institute, Center for Species Survival, National Zoological Park, Washington, DC, USA. [85]Department of Ecology and Evolutionary Biology, University of California Santa Cruz, Santa Cruz, CA, USA. [86]Howard Hughes Medical Institute, Chevy Chase, MD, USA. [87]The Walter Reed Biosystematics Unit, Museum Support Center MRC-534, Smithsonian Institution, Suitland, MD, USA. [88]Walter Reed Army Institute of Research, Silver Spring, MD, USA. [89]Department of Biological Sciences, Earlham Institute, University of East Anglia, Norwich, UK. [90]Institute of Evolutionary Biology (UPF-CSIC), PRBB, Barcelona, Spain. [91]Catalan Institution of Research and Advanced Studies (ICREA), Barcelona, Spain. [92]Centre for Genomic Regulation (CRG), Barcelona Institute of Science and Technology (BIST), Barcelona, Spain. [93]Institut Català de Paleontologia Miquel Crusafont, Universitat Autònoma de Barcelona, Barcelona, Spain. [94]School of Biology and Environmental Science, University College Dublin, Dublin, Ireland. [95]Department of Computer Science, The University of Illinois at Urbana-Champaign, Urbana, IL, USA. [96]School of Life Science, La Trobe University, Melbourne, Victoria, Australia. [97]Department of Evolution, Behavior, and Ecology, University of California San Diego, La Jolla, CA, USA. [98]Laboratory of Genomics Diversity-Center for Computer Technologies, ITMO University, St. Petersburg, Russian Federation. [99]Guy Harvey Oceanographic Center, Halmos College of Natural Sciences and Oceanography, Nova Southeastern University, Fort Lauderdale, FL, USA. [100]Department of Evolution and Ecology, University of California Davis, Davis, CA, USA. [101]John Muir Institute for the Environment, University of California Davis, Davis, CA, USA. [102]Faculty of Computer Science, Technical University Dresden, Dresden, Germany. [103]These authors contributed equally: Arang Rhie, Shane A. McCarthy, Olivier Fedrigo. [104]These authors jointly supervised this work: Kerstin Howe, Eugene W. Myers, Richard Durbin, Adam M. Phillippy, Erich D. Jarvis. ✉e-mail: kj2@sanger.ac.uk; gene@mpi-cbg.de; rd109@cam.ac.uk; adam.phillippy@nih.gov; ejarvis@rockefeller.edu

# Methods

## Genome assembly naming

For each completed assembly of an individual, we gave that assembly an abbreviated name with the following rules: Lineage/GenusSpecies/Individual#.Assembly#. The first letter, in lowercase, identifies the particular lineage: m, mammals; b, birds; r, reptiles; a, amphibians; f, teleost fish; and s, sharks and other cartilaginous fishes. The next three letters (first in caps) identify the species scientific genus name; the next three letters (first in caps) identifies the specific species name. In the last position is the genome identifier, where integers (1, 2, 3, …) represent different individuals of the same species, and decimals (1.1, 1.2, 1.3, …) represent different assemblies of the same individual. For example, the first submission of the curated Anna's hummingbird (*Calypte anna*) assembly is bCalAnn1.1, and an updated assembly for the same individual is bCalAnn1.2. When the abbreviated lineage or genus and species names for two or more species were identical, we replaced the subsequent letters (fourth, fifth and so on) of the genus or species name until they could be differentiated. We have created abbreviated names for all 71,657 vertebrate species (http://vgpdb.snu.ac.kr/splist/; https://id.tol.sanger.ac.uk/).

## Sample collection

The production of high-quality genome assemblies required us to obtain high-quality cells or tissue that would yield high-molecular-weight (HMW) DNA for long-read sequencing technologies (CLR and ONT) and optical mapping (Bionano). Therefore, we obtained fresh-frozen samples of various tissues (Supplementary Table 8). All samples were obtained according to approved protocols of the respective animal care and use committees or permits obtained by the respective persons and institutions listed in Supplementary Table 8. Additional details of the samples are on their respective BioSample pages (https://www.ncbi.nlm.nih.gov/biosample; accession numbers in Supplementary Table 8). All tissue types tested yielded a sufficient quantity and quality of DNA for sequencing and assembly, but we found that blood worked best for species that have nucleated red blood cells (that is, bird and reptiles), and spleen or cultured cells worked best for mammals, as of to date. Analysis of different tissue types will be presented elsewhere (in preparation).

## Isolation of high-molecular-weight DNA

**Agarose plug DNA isolation.** For tissue, HMW DNA was extracted using the Bionano animal tissue DNA isolation fibrous tissue protocol (cat no. RE-013-10; document number 30071), according to the manufacturer's guidelines. A total of 25–30 mg was fixed in 2% formaldehyde and homogenized using the Qiagen TissueRuptor or manual tissue disruption. For nucleated blood, 27–54 μl was used with an adapted protocol (Bionano, personal communication) of the Bionano Prep Blood and Cell Culture DNA Isolation Kit (cat no. RE-130-10). Lysates were embedded into agarose plugs and treated with Proteinase K and RNase A. Plugs were then purified by drop dialysis with 1× TE. DNA quality was assessed using pulse field gel electrophoresis (PFGE) (Pippin Pulse, SAGE Science, Beverly, MA) or the Femto Pulse instrument (Agilent). PFGE revealed that we isolated ultra-high-molecular-weight DNA between ~100 and ~500 kb long.

**Phenol–chloroform gDNA extraction.** For some samples, we performed phenol–chloroform extractions for HMW gDNA. Snap-frozen tissue was pulverized into a fine powder with a mortar and pestle in liquid nitrogen. The powdered tissue was lysed overnight at 55 °C in high-salt tissue lysis buffer (400 mM NaCl, 20 mM Tris base (pH 8.0), 30 mM EDTA (pH 8.0), 0.5% SDS, 100 μg/ml Proteinase K), and powdered lung tissue was lysed overnight in Qiagen G2 lysis buffer (cat no. 1014636, Qiagen, Hilden, Germany) containing 100 μg/ml Proteinase K at 55 °C. RNA was removed by incubation in 50 μg/ml RNase

A for 1 h at 37 °C. HMW gDNA was purified with two washes of phenol–chloroform-IAA equilibrated to pH 8.0, followed by two washes of chloroform-IAA, and precipitated in ice-cold 100% ethanol. Filamentous HMW gDNA was either spooled with shepherds hooks or collected by centrifugation. HMW gDNA was washed twice with 70% ethanol, dried for 20 min at room temperature and eluted in TE. For the flier cichlid muscle gDNA sample used for PacBio CLR and 10XG libraries, glycogen was precipitated by adding 1/10 (v/v) 0.3 M sodium acetate, pH 6.0 to the extracted genomic DNA, mixing carefully and spinning at room temperature at 10,000*g*. PFGE revealed that DNA molecule length was between 50 and 300 kb–often lower in size than that obtained with the agarose plug but sufficient for long-range sequencing of CLR and linked read data types.

**Others.** We also used the Qiagen MagAttract HMW DNA kit (cat no. 67563) and the KingFisher Cell and Tissue DNA kit (Thermo Scientific; cat no. 97030196), following the manufacturers' guidelines. These protocols yielded HMW DNA ranging from 30 to 50 kb. The Genomic Tip (Qiagen) kit was also used for tissue-based extraction of HMW DNA.

## Libraries and sequencing

**PacBio libraries and sequencing.** DNA obtained from agarose plugs was sheared down to ~40 kb fragment size with a MegaRuptor device (Diagenode, Belgium) and fragmented using Covaris g-tubes (520079) or by needle shearing. PacBio large insert libraries were prepared with either the SMRTbell Template Prep Kit 1.0-SPv3 (no.100-991-900) or the SMRTbell Express Template Prep Kit v1 (no. 101-357-000). Libraries were size-selected between 12 and 25 kb using Sage BluePippin (Sage Science, USA), depending on the DNA quality and extraction method. These libraries were sequenced on either RSII or Sequel I instruments, at least 60× coverage per species using Sequel Binding Kit and Sequencing Plate versions 2.0 and 2.1 with 10-h movie time (Supplementary Table 9).

**10X Chromium libraries and sequencing.** Unfragmented HMW DNA from the agarose plugs was used to generate linked read libraries on the 10X Genomics Chromium platform (Genome Library Kit & Gel Bead Kit v2 PN-120258, Genome Chip Kit v2 PN-120257, i7 Multiplex Kit PN-120262) following the manufacturer's guidelines. We sequenced the 10X libraries at ~60× coverage per species on an Illumina NovaSeq S4 150-bp PE lane.

**Bionano libraries and optical map imaging.** Unfragmented ultra-HMW DNA from the agarose plugs was labelled using either two different nicking enzymes (BspQI and BssSI) or a direct labelling enzyme (DLE1) following the Bionano Prep Labelling NLRS (document number 30024) and DLS protocols, respectively (document number 30206). Labelled samples were then imaged on a Bionano Irys or on a Bionano Saphyr instrument. For all species, we aimed for at least 100× coverage per label (Supplementary Table 9).

**Hi-C libraries and sequencing.** Chromatin interaction (Hi-C) libraries were generated using either Arima Genomics, Dovetail Genomics, or Phase libraries on muscle, blood, or other tissue with in vivo cross-linking (Supplementary Table 9) and sequenced on Illumina instruments. Arima-HiC preparations were performed by Arima Genomics (https://arimagenomics.com/) using the Arima-HiC kit that uses two enzymes (P/N: A510008). The resulting Arima-HiC proximally ligated DNA was then sheared, size-selected around 200–600 bp using SPRI beads, and enriched for biotin-labelled proximity-ligated DNA using streptavidin beads. From these fragments, Illumina-compatible libraries were generated using the KAPA Hyper Prep kit (P/N: KK8504). The resulting libraries were PCR amplified and purified with SPRI beads. The quality of the final libraries was checked with qPCR and Bioanalyzer, and then sequenced on Illumina HiSeq X at ~60× coverage following the manufacturer's protocols. Dovetail-HiC preparations were performed

by Dovetail using a single-enzyme (DpnII) proximity ligation approach. Phase-HiC libraries were made by Phase Genomics using a Proximo Hi-C Library single-enzyme reaction.

## Quality control

Before we performed any assembly, all genomic data of all data types from each sample were used to screen potential outlier libraries, outlier sequencing runs, or accidental species contamination with Mash[73] by measuring sequence similarity (Supplementary Fig. 4). When running Mash, we used 21-mers to generate sketches with sketch size of 10,000 and compared among each sequencing run, and then differences assessed between sequencing sets.

## Genome size, repeat content, and heterozygosity estimations

These estimations were made with $k$-mer-based methods applied to the Illumina short reads obtained from 10XG linked sequencing libraries. After trimming off barcodes during scaff10x[74] preprocessing, canonical 31-mer counts were collected using Meryl[23]. With the resulting 31-mer histogram, GenomeScope[71] was used to estimate the haploid genome length, repeat content, and heterozygosity. The thorny skate linked read data failed quality control, which we suspect was due to low complexity sequences from the high repeat content (54.1%) of the genome; so $k$-mers were collected later from Illumina whole-genome sequencing reads instead. The genome size and repeat content of the channel bull blenny were estimated from an alternative method that looks at the mode of long read overlap coverage and WindowMasker[75], as the estimated genome size from GenomeScope was almost doubling the known haploid genome size (1.29 Gb versus 0.6 Gb) and repeat content (28.0% versus 58.0%), for reasons related to either the quality of the 10X data or species differences.

## Benchmarking assembly steps with the Anna's hummingbird

To develop the VGP standard pipeline, we compared various scaffolding, gap filling, and polishing tools. Default options were used unless otherwise noted. Detailed software versions are listed in Supplementary Table 2.

**Contigging and scaffolding.** FALCON[76] and FALCON-Unzip[17] (smrtanalysis 3.0.0) were used to generate contigs that used CLR. Canu[77] 1.5+67 was used to generate the combined PacBio CLR and Oxford Nanopore ONT assembly. To benchmark scaffolding with linked reads, we used scaff10x[74] 2.0. For the linked read-only assembly, Supernova 2[78] was used. For the optical maps, two-enzyme hybrid scaffolding was used in the Bionano Solve v3.2.1 software, using BspQI and BssSI initially, as well as DLE1 later when the technology was developed. For benchmarking Hi-C in scaffolding, Salsa 2.2[79] was used for scaffolding results in Fig. 1a, with Hi-C reads generated from Arima Genomics. Additional comparisons for the Hi-C libraries were performed using assemblies provided by Dovetail Genomics and Phase Genomics (Supplementary Table 3). We used Hi-C from Arima Genomics as it had the smallest number of PCR duplicates and better coverage for short and long interactions at the time of comparison (Supplementary Fig. 1). Assembly statistics from HiRise, Proximo HiC, 3D-DNA[80] and Arima Hi-C are available in Supplementary Table 3. We concluded that all Hi-C scaffolding algorithms had similar performance. We decided to use Salsa, as HiRise and Proximo HiC were not open access, and 3D-DNA was computationally expensive on the DNAnexus platform. For short read assemblies, other than Supernova and the NRGene assembly, the assembly GCA_000699085.1[16] was used for benchmarking, which was generated with Illumina paired-end, multiple mate-pair libraries and the SoapDeNovo[81] assembler. The NRGene assembly was provided by the company with DeNovo Magic.

**Gap filling.** We ran PBJelly with support --capturedOnly --spanOnly parameters, to avoid greedy gap closures with no spanning read

support. For conservatively filling sequences, we compared different parameters in output stage with --minreads 1 and --minreads 4 in addition to no restrictions. We found that the number of gaps closed was similar to the gaps filled with Arrow[76] (Supplementary Table 4) and chose not to run PBJelly[82] for future assemblies.

**Short-read polishing.** Illumina polishing benchmarking was performed using Longranger[83] 2.1.3 and Pilon[84] 1.21 with --fix bases, local option (Supplementary Table 5). Later, for the VGP pipeline, we used FreeBayes[85] as Pilon[84] was not computationally scalable for large genomes with the updated Longranger 2.2.2.

**Base-level accuracy estimate.** Base-level accuracy was measured using a mapping-based approach and later using the $k$-mer-based approach[23]. To determine the number of rounds to polish, we used Illumina paired-end reads from the hummingbird[16].

**Mis-joins and missed-joins.** The curated hummingbird assembly was mapped to the target assemblies with MashMap2[86] with --filter_mode one-to-one --pi 95 using 5 kb segments (-s 5000) for CLR assemblies and 1 kb (-s 1000) for SR assemblies to compensate for the shorter contig sizes, as contigs smaller than a segment size will be excluded from the alignment. The number of mis-joins and missed joins were identified using the assembly_comparison.pl used in the 'Curation' section below (Supplementary Methods, Supplementary Fig. 5).

## VGP standard genome assembly pipeline 1.0 to 1.6

All 17 genomes were assembled with the VGP pipeline (Extended Data Fig. 2a) for benchmark purposes, with some uncurated. The pale spear-nosed bat, greater horseshoe bat, Canada lynx, platypus, male and female zebra finch, kākāpō, Anna's hummingbird, Goode's thornscrub tortoise, flier cichlid, and blunt-snouted clingfish assemblies were generated using the VGP pipeline 1.0 to 1.6 and curated for submission to NCBI and EBI public archives. The curated and submitted two-lined caecilian, zig-zag eel, climbing perch, channel bull blenny, eastern happy, and thorny skate assemblies were generated using a similar process developed in parallel (Supplementary Note 2). Two submitted curated versions of the female zebra finch were made, one using the standard VGP pipeline and the other using the VGP trio pipeline, so that comparative analyses could be performed by others.

**Contigging.** For PacBio data, contigs were generated from subreads using FALCON[76] and FALCON-Unzip[17], with one round of Arrow polishing (smrtanalysis 5.1.0.26412). A minimum read length of 2 kb or a cutoff at which reads longer than the cutoff include 50× coverage was used, whichever was longer. For calculating read coverage, we used estimated genome size from http://www.genomesize.com/ when available, or from the literature (Supplementary Table 11) while waiting for 10XG sequencing to estimate genome size using $k$-mers. FALCON and FALCON-Unzip were run with default parameters, except for computing the overlaps. Raw read overlaps were computed with DALIGNER parameters -k14 -e0.75 -s100 -l2500 -h240 -w8 to better reflect the higher error rate in early PacBio sequel I and II. Pread (preassembled read) overlaps were computed with DALIGNER parameters -k24 -e.90 -s100 -l1000 -h600 intending to collapse haplotypes for the FALCON step to better unzip genomes with high heterozygosity rate. FALCON-Unzip outputs both a pseudo-haplotype and a set of alternate haplotigs that represent the secondary alleles. We refer to these outputs as the primary contig set (c1) and alternate contig set (c2).

**Purging false duplications.** Heterotype false duplications occurred despite setting FALCON[76] parameters to resolve up to 10% haplotype divergence. FALCON-Unzip[17] also incorrectly retained some secondary alleles in the primary contig set, which appeared as false duplications. To reduce these false duplications, we ran Purge_Haplotigs[13], first during

curation (VGP v1.0 pipeline) and then later after contig formation (VGP v1.5 pipeline). To do the former, Purge_Haplotigs was run on the primary contigs (c1), and identified haplotigs were mapped to the scaffolded primary assembly with MashMap2[86] for removal. In the latter, identified haplotigs were moved from the primary contigs (c1) to the alternate haplotig set (p2). The remaining primary contigs were referred to as p1; p2 combined with c2 was referred to as q2. Later, in the VGP v1.6 pipeline, we replaced Purge_Haplotigs with Purge_Dups[14], a new program developed by several of the authors in response to Purge_Haplotigs not removing partial false duplication at contig boundaries. Purging also removes excessive low-coverage (junk) and high-coverage (repeats) contigs. To calculate the presence and overall success of purging false duplications, we used a *k*-mer approach (Supplementary Methods, Supplementary Fig. 6).

**Scaffolding with 10XG linked reads.** The 10X Genomics linked reads were aligned to the primary contigs (p1), and an adjacency matrix was computed from the barcodes using scaff10x[74] v2.0–2.1. Two rounds of scaffolding were performed. The first round was run with parameters -matrix 2000 -reads 12 -link 10, and the second round with parameters -matrix 2000 -reads 8 -link 10. A gap of 100 bp (represented with 'N's) was inserted between joined contigs. The resulting primary scaffold set was named s1.

**Scaffolding with Bionano optical maps.** Bionano cmaps were generated using the Bionano Pipeline in non-haplotype assembly mode and used to further scaffold the s1 assembly with Bionano Solve v3.2.1[87]. We began with a one-enzyme nick map (BspQI), followed by a two-enzyme nick map (BspQI and BssSI), and then with a DLE-1 one-enzyme non-nicking approach when the later data type became available (Supplementary Table 9). Scaffold gaps were sized according to the software estimate. The resulting scaffold set was named s2.

**Scaffolding with Hi-C reads.** Hi-C reads were aligned to the s2 scaffolds using the Arima Genomics mapping pipeline[88]. In brief, both ends of a read pair were mapped independently using BWA-MEM[89] with the parameter -B8, and filtered when mapping quality was <10. Chimeric reads containing a restriction enzyme site were trimmed from the restriction site onward, leaving only the 5′ end. The filtered single-read alignments were then rejoined as paired read alignments. The processed alignments were then used for scaffolding with Salsa2[79], which analyses the normalized frequency of Hi-C interactions between all pairs of contig ends to determine a likely ordering and orientation of each. We used parameters -m yes -i 5 -p yes to allow Salsa2 to break potentially mis-assembled contigs and perform five iterations of scaffolding. After feedback from curation, later versions of Salsa were developed, which more conservatively determine the number of iterations (v2.1) and actively break at mis-assemblies (v2.2), and run for the Canada lynx, Goode's thornscrub tortoise, and two-lined caecilian. The restriction enzyme(s) used to generate each library were specified using parameters -e GATC,GANTC for Arima and -e GATC for Dovetail and Phase Genomics Hi-C data. The resulting Hi-C scaffolded assembly was named s3.

**Consensus polishing.** To polish bases in both haplotypes with minimal alignment bias, we concatenated the alternate haplotig set (c2 in v1.0 or q2 in v1.5–1.6) to the scaffolded primary set (s3) and the assembled mitochondrial genome (mitoVGP in v1.6). We then performed another round of polishing with Arrow (smrtanalysis 5.1.0.26412) using PacBio CLR reads, aligning with pbalign --minAccuracy=0.75 --minLength=50 --minAnchorSize=12 --maxDivergence=30 –concordant --algorithm=blasr --algorithmOptions=--useQuality --maxHits=1 --hitPolicy=random --seed=1 and consensus polishing with variantCaller --skipUnrecognizedContigs haploid -x 5 -q 20 -X120 –v --algorithm=arrow. While this round of polishing resulted in higher

QV for all genomes herein considered, we noticed that it was particularly sensitive to the coverage cutoff parameter (-x). This is because Arrow generates a de novo consensus from the mapped reads without explicitly considering the reference sequence. Later, we found that the second round of Arrow polishing sometimes reduced the QV accuracy for some species. Upon investigation, this issue was traced back to option -x 5, which requires at least 5 reads to call consensus. Such low minimum requirements can lead to uneven polishing in low coverage regions. To avoid this behaviour, we suggest to increase the -x close to the half sequence coverage (for example, 30× when 60× was used for assembly) and check QV before moving forward.

For genomes with a combined assembly size larger than 4 Gb, we used Minimap2[90] with parameters -ax map-pb instead of Blasr[91] to overcome reference index size limitations.

Two more rounds of base-pair polishing were performed with linked reads. The reads were aligned with Longranger align 2.2.2, which incorporates the Lauriat for barcode-aware alignment[83]. From the alignments, homozygous mismatches (variants) were called with FreeBayes[83] v1.2.0 using default options. Consensus was called with bcftools consensus[92] with -i'QUAL>1 && (GT='AA' || GT = 'Aa')' -Hla.

**VGP Trio Pipeline v1.0–v1.6.** The trio pipeline is similarly designed to the standard pipeline, except for the use of parental data (Extended Data Fig. 3b). When parental genomes are available, the child's CLR reads are binned to maternal and paternal haplotypes, and assembled separately as haplotype-specific contigs (haplotigs) using TrioCanu[20]. In brief, parental specific marker *k*-mers were collected using Meryl[23] from the parental Illumina WGS reads of the parents. These markers were filtered and used to bin the child's CLR read. A haplotype was assigned given the markers observed, normalized by the total markers in each haplotype. The subsequent purging, scaffolding, and polishing steps were similarly updated with the use of Purge_Dups[14] (v1.6). We extended binning to linked reads and Hi-C reads, by excluding read pairs that had any parental-specific marker. The binned Hi-C reads were used to scaffold its haplotype assembly, and polished with the binned linked reads from the observation of haplotype switching using the standard polishing approach. During curation, one of the haplotype assemblies with the higher QV and/or contiguity was chosen as the representative haplotype. The heterogametic sex chromosome from the unchosen haplotype was added to the representative assembly. However, while curating several trios, we found that in regions of low divergence between shared parental homogametic sex chromosomes (that is, X or Z), a small fraction of offspring CLR data was mis-assigned to the wrong haplotype. This mis-alignment resulted in a duplicate, low-coverage offspring X or Z assembly in the paternal (for mammals) or maternal (for birds) haplotype, respectively, which required removal during curation. We are working on methods to improve the binning accuracy for resolution of this issue going forward.

For the female zebra finch in particular, contigs were generated before the binning was automated in the Canu assembler as TrioCanu1.7, and therefore a manual binning process was applied as described in the original Trio-binning paper[20] (Supplementary Methods). Contigs were assembled for each haplotype using the binned reads, excluding unclassified reads. The contigs were polished with two rounds of Arrow polishing using the binned reads, and scaffolded following the v1.0 pipeline with no purging. Additional scaffolding rounds with Bionano (s4) and Hi-C were applied. Scaffolds were renamed according to the primary scaffold assembly of the same individual (s5), with sex chromosomes grouped as Z in the paternal assembly and W in the maternal assembly following synteny to the Z chromosome from the curated male zebra finch VGP assembly. Two rounds of SR polishing were applied using linked reads, by mapping on both haplotypes. After haplotype switches were discovered, additional rounds of polishing were applied using binned linked reads (Supplementary Methods).

**Mitochondrial genome assembly.** Similar to other recent methods[93,94], we developed a reference-guided MT assembly pipeline. MT reads in the raw CLR data were identified by mapping the whole read set to an existing reference sequence of the specific species or of closely related species using Blasr. Filtered mtDNA CLRs were assembled into a single contig using Canu v1.8, polished with Arrow using CLR and then FreeBayes v1.0.2 together with bcftools v1.9 using short reads from the 10XG data (Extended Data Fig. 3c). The overlapping sequences at the ends of the contig were trimmed, and the remaining contig sequence circularized. The mitoVGP pipeline is made available at https://github.com/VGP/vgp-assembly/tree/master/mitoVGP. A more detailed protocol description of the assembly pipeline and new discoveries from the MT assemblies are published elsewhere[33].

## Curation

The VGP genome assembly pipeline produces high quality assemblies, yet no automated method to date is free from the production of errors, especially during the scaffolding stages. To minimize the impact of the remaining algorithmic shortcomings, we subjected all assemblies to rigorous manual curation. All data generated for a species in this study and other publicly available data (for example, genetic maps, gene sets and genome assemblies of the same or closely related species) were collated, aligned to the primary assembly and analysed in gEVAL[95] (https://vgp-geval.sanger.ac.uk/index.html), visualizing discordances in a feature browser and issue lists. In parallel, Hi-C data were mapped to the primary assembly and visualized using Juicebox[96] and/or HiGlass[97]. With these data, genome curators identified mis-joins, missed joins and other anomalies, and corrected the primary assembly accordingly. No change was made without unambiguous evidence from available data types; for example, a Hi-C suggested join would not be made unless supported by BioNano maps, long-read data, or gene alignments. When sequencing the heterogametic sex, we identified sex chromosomes based on half coverage, homology alignments to sex chromosomes in other species, and the presence of sex chromosome-specific genes.

**Contamination removal.** A succession of searches was used to identify potential contaminants in the generated assemblies.

1) A megaBLAST[98] search against a database of common contaminants (ftp://ftp.ncbi.nlm.nih.gov/pub/kitts/contam_in_euks.fa.gz) requiring $e \leq 1 \times 10^{-4}$, reporting matches with ≥98% sequence identity and match length 50–99 bp, ≥94% and match length 100–199 bp, or ≥90% and match length 200 bp or above.

2) A vecscreen (https://www.ncbi.nlm.nih.gov/tools/vecscreen/) search against a database of adaptor sequences (ftp://ftp.ncbi.nlm.nih.gov/pub/kitts/adaptors_for_screening_euks.fa)

3) After soft-masking repeats using Windowmasker[75], a megaBLAST search against chromosome-level assemblies from RefSeq requiring $e \leq 1 \times 10^{-4}$, match score ≥100, and sequence identity ≥98%; regions matching highly conserved rDNAs were ignored.

Manual inspection of the results was necessary to differentiate contamination from conservation and/or horizontal gene transfer. Adaptor sequences were masked; other contaminant sequences were removed. Assemblies were also checked for runs of Ns at the ends of scaffolds, created as artefacts of the iterative scaffolding process, and when found they were trimmed.

**Organelle genomes.** These were detected by a megaBLAST search against a database of known organelle genomes requiring $e \leq 1 \times 10^{-4}$, sequence identity ≥90%, and match length ≥500; the databases are available at ftp://ftp.ncbi.nlm.nih.gov/blast/db/FASTA/mito.nt.gz and ftp://ftp.ncbi.nlm.nih.gov/refseq/release/plastid/*genomic.fna.gz. Only scaffolds consisting entirely of organelle sequences were assumed to be organelle genomes, and replaced by the genome from the separate organelle assembly pipeline. Organelle matches embedded in nuclear sequences that were found to be NuMTs were kept.

**False duplication removal.** Retained false duplications were identified using Purge_Haplotigs[13] run either after scaffolding and polishing (Anna's hummingbird, kākāpō, male zebra finch, female zebra finch, platypus, pale spear-nosed bat, and greater horseshoe bat) or on the c1 before scaffolding (two-lined caecilian, flier cichlid, Canada lynx, and Goode's thornscrub tortoise). Subsequent manual curation identified additional haplotypic duplications for the listed assemblies and also those that were not treated with Purge_Haplotigs (Eastern happy, climbing perch, zig-zag eel). The evidence used included read coverage, sequence self-comparison, transcript alignments, Bionano map alignments and Hi-C 2D maps, all confirming the superfluous nature of one allele. The identified haplotype duplications were moved from the primary to the alternate assembly.

**Chromosome assignment.** For a scaffold to be annotated as a chromosome, we used evidence from Hi-C as well as genetic linkage or FISH karyotype mapping when available. For Hi-C evidence, we considered a scaffold as a complete chromosome (albeit with gaps) when there was a clear unbroken diagonal in the Juicebox or HiGlass plots for that scaffold and no other large scaffolds that could be joined to that same scaffold; if present and no unambiguous join was possible, we named it as an unlocalized scaffold for that chromosome. When we could not find evidence of a complete chromosome, we kept the scaffold number for its name. We named all evidence-validated scaffolds as chromosomes down to the smallest Hi-C box unit resolution allowed with these characteristics. When there was an established chromosome terminology for a given species or set of species, we use the established terminology except when our new assemblies revealed errors in the older assembly, such as scaffold/chromosome fusions, fissions, rearrangements, and non-chromosome names. For species without an established chromosome terminology, we named the scaffolds as chromosomes numbers 1, 2, 3 …, in descending order of scaffold size. For the sex chromosomes, we used the letters X and Y for mammals and Z and W for birds.

**Using comparative genomics to assess assembly structure.** In cases where a high-quality chromosome-level genome was available for a closely related species, comparative genome analysis was performed. The polished primary assembly (t3.p) was mapped to the related genome using MashMap2[86] with --pi 75 -s 300000. The number of chromosomal differences was identified using a custom script available at https://github.com/jdamas13/assembly_comparison. This resulted in the identification of ~60 to ~450 regions for each genome assembly flanking putative misassemblies or lineage-specific genome rearrangements. To identify which were real misassemblies, the identified discrepancies were communicated to the curation team for manual verification (see above).

To identify any possible remaining mis-joins, each curated avian and mammalian assembly was compared with the zebra finch (taeGut2) or human (hg38) genomes, respectively. Pairwise alignments between each of the VGP assemblies and the clade reference were generated with LastZ[99] (version 1.04) using the following parameters: $C = 0$ $E = 30$ $H = 2000$ $K = 3000$ $L = 2200$ $O = 400$. The pairwise alignments were converted into the UCSC 'chain' and 'net' formats with axtChain (parameters: -minScore = 1000 -verbose = 0 -linearGap = medium) followed by chainAntiRepeat, chainSort, chainPreNet, chainNet and netSyntenic, all with default parameters[100]. Pairwise synteny blocks were defined using maf2synteny[101] at 100-, 300-, and 500-kb resolutions. Evolutionary breakpoint regions were detected and classified using an ad hoc statistical approach[102]. This analysis identified 2 to 90 genomic regions per assembly that could be flanking misassemblies, lineage-specific chromosome rearrangements, or reference-specific chromosome rearrangements (116 in the human

and 26 in the zebra finch). Determining the underlying cause for each of the flagged regions will need further verification. All alignments are available for visualization at the Evolution Highway comparative chromosome browser (http://eh-demo.ncsa.illinois.edu/vgp/).

### Annotation
NCBI and Ensembl annotation pipeline used in this study are described in the Supplementary Methods.

### Evaluation
Detailed methods for other types of evaluation, including BUSCO runs, mis-join and missed-join identification, reliable blocks, collapsed repeats, telomeres, RNA-seq and ATAC–seq mapping, and false gene duplications are in the Supplementary Methods. No statistical methods were used to predetermine sample size, the experiments were not randomized, and the investigators were not blinded to group during experiments and outcome assessment.

### Reporting summary
Further information on research design is available in the Nature Research Reporting Summary linked to this paper.

## Data availability
All raw data, intermediate and final assemblies are publicly available via GenomeArk (https://vgp.github.io/genomeark), archived on NCBI/EBI BioProject under accession PRJNA489243 with annotations, and browsable on the UCSC Genome Browser (https://hgdownload.soe.ucsc.edu/hubs/VGP/). The final primary assembly from the automated pipeline before curation is browsable on gEVAL (https://vgp-geval.sanger.ac.uk) with all four raw data mappings. The VGP assembly pipeline is available as a stand-alone pipeline (https://github.com/VGP/vgp-assembly) as well as a workflow on DNAnexus (https://platform.dnanexus.com/). A VGP-specific assembly hub portal in the U.C. Santa Cruz browser is available as a gateway to access all VGP genome assemblies and annotations (https://hgdownload.soe.ucsc.edu/hubs/VGP).

## Code availability
All codes used in the VGP Assembly Pipeline and the VGP Trio Pipeline are publicly available at https://github.com/VGP/vgp-assembly/tree/master/pipeline.

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

**Acknowledgements** We thank the following persons for feedback and support: R. Johnson, E. Karlsson, K. Lindblad Toh, W. Jun, I. Korf, W. Haerty, G. Etherington, B. Clavijo, and A. Komissarov for discussions in the early stages of the project; R. Fuller for help with the G10K website maintenance, and H. Segal for help with with VGP website development; M. Linh Pham for help with initial grant writing; L. Shalmiyev for administrative help; D. Church, G. Kol, K. Baruch, O. Barad, I. Liachko, E. Muzychenko, S. Garg, and M. Kolmogorov for preliminary analyses performed on one or more genomes; K. Oliver, C. Corton and J. Skelton for data generation; E. Harry for technical support in scaff10x and Pretext; C. Mazzoni for coordinating students and training at Leibniz Institute for Zoo and Wildlife Research and Berlin Center for Genomics in Biodiversity Research; and M. Driller, C. Caswara, M. Vafadar, N. Hill, D. De Panis, A. Whibley, B. Maloney, C. Mitchell, G. Gallo, J. Gaige, K. Amoako-Boadu, M. Jose Gomez, M. Montero, D. Ratnikov, S. Brown, S. Zylka, S. Marcus, and T. Carrasco for completing training and testing the VGP pipeline by producing ordinal representative genome assemblies not described in this manuscript. We thank our company partners (listed below), NCBI, EBI, and Amazon AWS, including AWS for sponsoring sequence storage. J. Fekecs and D. Leja created the animal images, and J. Kim modified them to silhouettes. We thank them for their permission to publish. A.R., S.K., B.P.W. and A.M.P. were supported by the Intramural Research Program of the NHGRI, NIH (1ZIAHG200398). A.R. was also supported by the Korea Health Technology R&D Project through KHIDI, funded by the Ministry of Health & Welfare, Republic of Korea (HI17C2098). S.A.M., I.B. and R.D. were supported by Wellcome Trust grant WT207492; W.C., M. Smith, Z.N., Y.S., J.C., S. Pelan, J.T., A.T., J.W. and Kerstin Howe by WT206194; L.H., F.M., Kevin Howe and P. Flicek by WT108749/Z/15/Z, WT218328/B/19/Z and the European Molecular Biology Laboratory. O.F. and E.D.J. were supported by Howard Hughes Medical Institute and Rockefeller University start-up funds for this project. J.D. and H.A.L. were supported by the Robert and Rosabel Osborne Endowment. M.U.-S. received funding from the European Union's Horizon 2020 research and innovation programme under the Marie Skłodowska-Curie grant agreement (750747). F.T.-N., J. Hoffman, P. Masterson and K.C. were supported by the Intramural Research Program of the NLM, NIH. C.L., B.J.K., J. Kim and H.K. were supported by the Marine Biotechnology Program of KIMST, funded by the Ministry of Ocean and Fisheries, Republic of Korea (20180430). M.C. was supported by Sloan Research Fellowship (FG-2020-12932). S.C.V. was funded by a Max Planck Research Group award from the Max Planck Society, and a Human Frontiers Science Program (HFSP) Research grant (RGP0058/2016). T.M.L., W.E.J. and the Canada lynx genome were funded by the Maine Department of Inland Fisheries & Wildlife (F11AF01099), including when W.E.J. held a National Research Council Research Associateship Award at the Walter Reed Army Institute of Research (WRAIR). C.B. was supported by the NSF (1457541 and 1456612). D.B. was funded by The University of Queensland (HFSP - RGP0030/2015). D.I. was supported by Science Exchange Inc. (Palo Alto, CA). H.W.D. was supported by NSF grants (OPP-0132032 ICEFISH 2004 Cruise, PLR-1444167 and OPP-1955368) and the Marine Science Center at Northeastern University (416). G.J.P.N. and the thorny skate genome were funded by Lenfest Ocean Program (30884). M.P. was funded by the German Federal Ministry of Education and Research (01IS18026C). M. Malinsky was supported by an EMBO fellowship (ALTF 456-2016). The following authors' contributions were supported by the NIH: S. Selvaraj (R44HG008118); C.V.M., S.R.F., P.V.L. (R21 DC014432/DC/NIDCD); K.D.M.

(R01GM130691); H.C. (5U41HG002371-19); M.D. (U41HG007234); and B.P. (R01HG010485). D.G. was supported by the National Key Research and Development Program of China (2017YFC1201201, 2018YFC0910504 and 2017YFC0907503). F.O.A. was supported by Al-Gannas Qatari Society and The Cultural Village Foundation-Katara, Doha, State of Qatar and Monash University Malaysia. C.T. was supported by The Rockefeller University. M. Hiller was supported by the LOEWE-Centre for Translational Biodiversity Genomics (TBG) funded by the Hessen State Ministry of Higher Education, Research and the Arts (HMWK). H.C. was supported by the NHGRI (5U41HG002371-19). R.H.S.K. was funded by the Max Planck Society with computational resources at the bwUniCluster and BinAC funded by the Ministry of Science, Research and the Arts Baden-Württemberg and the Universities of the State of Baden-Württemberg, Germany (bwHPC-C5). B.V. was supported by the Biomedical Research Council of A*STAR, Singapore. T.M.-B. was funded by the European Research Council under the European Union's Horizon 2020 research and innovation programme (864203), MINECO/FEDER, UE (BFU2017-86471-P), Unidad de Excelencia María de Maeztu, AEI (CEX2018-000792-M), a Howard Hughes International Early Career award, Obra Social "La Caixa" and Secretaria d'Universitats i Recerca and CERCA Programme del Departament d'Economia i Coneixement de la Generalitat de Catalunya (GRC 2017 SGR 880). E.C.T. was supported by the European Research Council (ERC-2012-StG311000) and an Irish Research Council Laureate Award. M.T.P.G. was supported by an ERC Consolidator Award 681396-Extinction Genomics, and a Danish National Research Foundation Center Grant (DNRF143). T.W. was supported by the NSF (1458652). J. M. Graves was supported by the Australian Research Council (CEO561477). E.W.M. was partially supported by the German Federal Ministry of Education and Research (01IS18026C). Complementary sequencing support for the Anna's hummingbird and several genomes was provided by Pacific Biosciences, Bionano Genomics, Dovetail Genomics, Arima Genomics, Phase Genomics, 10X Genomics, NRGene, Oxford Nanopore Technologies, Illumina, and DNAnexus. All other sequencing and assembly were conducted at the Rockefeller University, Sanger Institute, and Max Planck Institute Dresden genome labs. Part of this work used the computational resources of the NIH HPC Biowulf cluster (https://hpc.nih.gov). We acknowledge funding from the Wellcome Trust (108749/Z/15/Z) and the European Molecular Biology Laboratory. We thank Le Comité Scientifique Régional du Patrimoine Naturel and Direction de l'Environnement, de l'Aménagement et du Logement, Guyanne for research approvals and export permits.

**Author contributions** Wrote the paper and co-coordinated the study: A.R., E.D.J., A.M.P., R.D., E.W.M., Kerstin Howe, S.A.M., O.F. Coordination with vendors: J. Korlach, S. Selvaraj, R.E.G., A.H., M. Mooney. Collected samples: M.T.P.G., W.E.J., R.W.M., G.Z., B.V., M.T.B., J. Howard, S.C.V., T.M.L., F.G., W.C.W., D.B., J. M. George, M.T.B., D.I., A.D., D.E., B.R., T.E., M. Wilkinson, G.T., A. Meyer, A.F.K., P. Franchini, H.W.D., H.S., M. Wagner, G.J.P.N., R.D., E.D.J., E.C.T., R.H.S.K. Generated genome data: O.F., I.B., M. Smith, B.H., J.M., S.W., C.B., A. Meyer, A.F.K., P. Franchini, I.G., D.F.C., C.V.M. Generated genome assemblies: A.R., S.A.M., S.K., M.P., S.B.K., R.H., J.G., Z.N., J.L., B.P.W., M. Malinsky. Generated/modified software: S.K., A.R., S.B.K., R.H., Z.K., J. Korlach, I.S., C.D., Z.N., A.H., J.L., J.G., E.G., C.V.M., S.R.F., N.H.P. Pipeline development: A.R., S.A.M., G.F., S.K., M.U.-S., A.F., M. Simbirsky, B.T.H., T.P., M.P., E.W.M., R.D., A.M.P. Generated MT assemblies: G.F., J. Korlach. Curation: Kerstin Howe, W.C., Y.S., J.C., S. Pelan, J.T., A.T., J.W., Y.Z., J.D., H.A.L. Sex chromosomes: Y.Z., R.S.H., K.D.M., P. Medvedev, J. M. Graves. Hummingbird karyotype analyses: M. Houck, A. Misuraca, M.P., E.W.M., E.D.J. Annotation: F.T.-N., L.H., J. Hoffman, P. Masterson, K.C., F.M., Kevin Howe, P. Flicek, D.B. Evaluation analysis: A.R., J.D., M.U.-S., J. Kim, C.L., B.J.K., M.C., G.L.G., L.J.C., F.T.-N., L.H., J. M. George, J.G., R.E.D., D.G., S.E.L., D.F.C., C.V.M., S.R.F., P.V.L., E.O., F.O.A.-A., S. Secomandi, C.T., M. Hiller, H.K., Kerstin Howe, E.W.M., R.D., A.M.P., E.D.J. Biological findings: J.D., J. Kim, C.L., B.J.K., G.L.G., L.J.C., H.A.L., A.R., E.D.J. Data availability: A.R., S.A.M., W.C., A.F., S. Paez, M. Simbirsky, B.T.H., B.P.W., W.K., H.C., M.D., L.N., B.P., A.M.P., E.D.J. G10K council, founders, and coordination of VGP: T.M.-B., A.J.C., F.D.P., R.D., M.T.P.G., E.D.J., K.-P.K., H.A.L., R.W.M., E.W.M., E.C.T., B.V., G.Z., A.M.P., S. Paez, J. M. Graves, O.A.R., D.H., S.J.O., T.W. and B.S. All authors reviewed the manuscript.

**Competing interests** During the contributing period, B.T.H., M. Simbirsky, A.F. and M. Mooney were employees of DNAnexus Inc. S.B.K., R.H., Z.K., J. Korlach, I.S. and C.D. were full-time employees at Pacific Biosciences, a company developing single-molecule long read sequencing technologies. R.E.G., N.H.P., and J.G. were affiliated with Dovetail Genomics, a company developing genome assembly tools, including Hi-C. I.G. was affiliated with Oxford Nanopore Technologies, a company generating long read sequencing technologies. A.H. and J.L were employees of Bionano Genomics, a company developing optical maps for genome assembly. S. Selvaraj was an employee of Arima Genomics, a company developing Hi-C data for genome assemblies. R.D. is a scientific advisory board member of Dovetail Inc. P. Flicek is a member of the Scientific Advisory Boards of Fabric Genomics, Inc., and Eagle Genomics, Ltd. H.C. receives royalties from the sale of UCSC Genome Browser source code, LiftOver, GBiB, and GBiC licenses to commercial entities. S.K. has received travel funds to speak at symposia organized by Oxford Nanopore. M.D. and L.N. receive royalties from licensing of UCSC Genome Browser. For W.E.J., the content here is not to be construed as the views of the DA or DOD. All other authors declare no competing interests.

**Additional information**
**Correspondence and requests for materials** should be addressed to Kerstin Howe., E.W.M., R.D., A.M.P. or E.D.J.

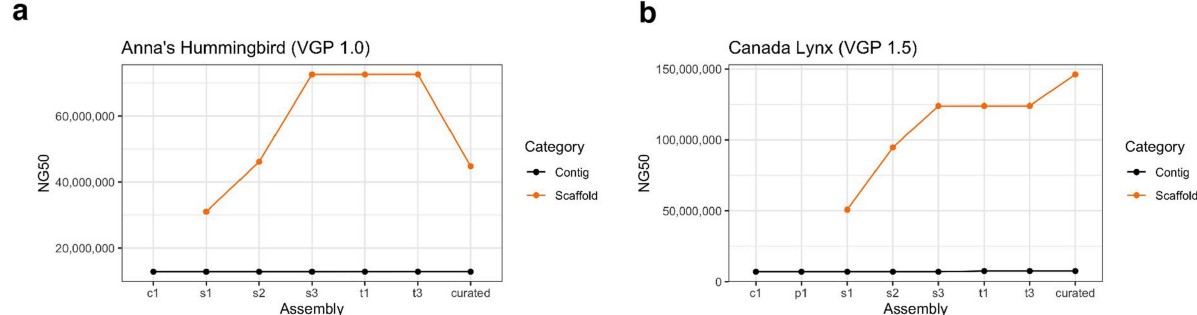

**a** Anna's Hummingbird (VGP 1.0)

**b** Canada Lynx (VGP 1.5)

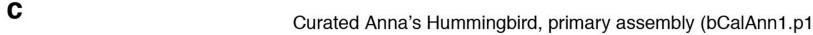

**c** Curated Anna's Hummingbird, primary assembly (bCalAnn1.p1)

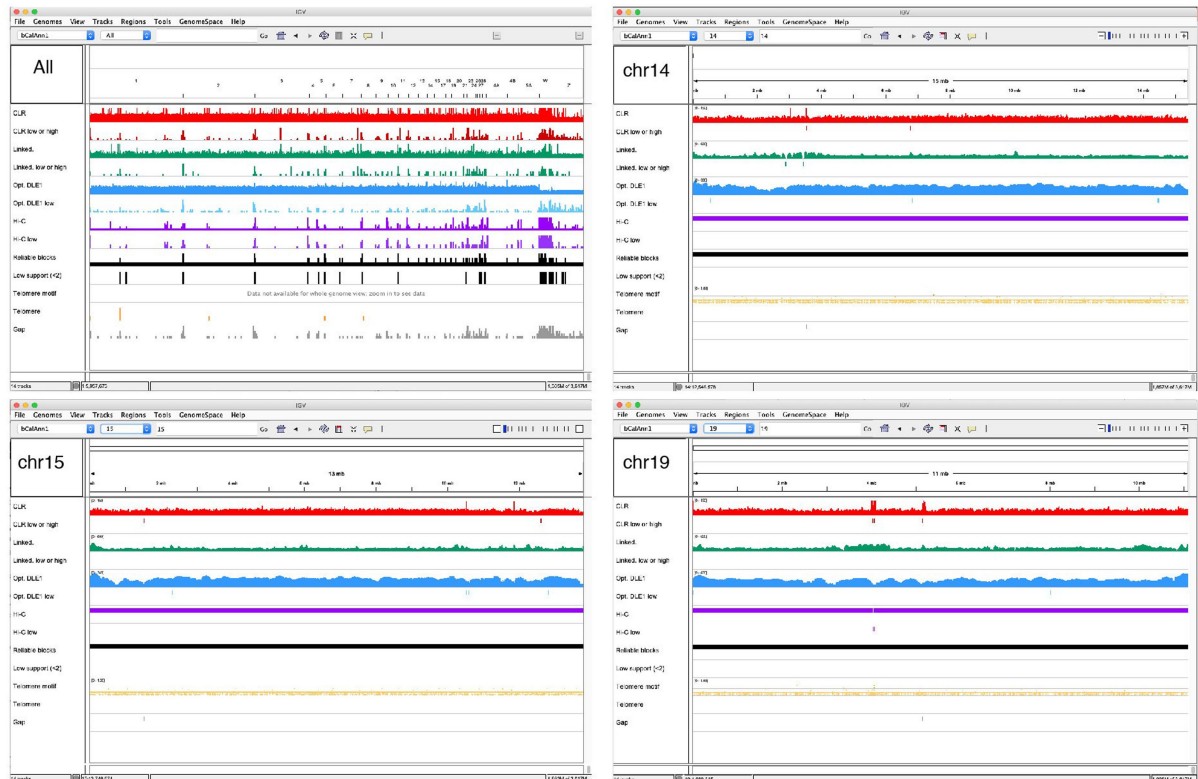

**Extended Data Fig. 1 | Assessment of completeness of the Anna's hummingbird assembly. a**, **b**, Steps and NG50 continuity values of the VGP assembly pipeline that gave the highest quality assembly for Anna's hummingbird (**a**) and Canada lynx (**b**) in this study. The specific steps are outlined further in Extended Data Fig. 2a, and Methods. **c**, Whole-genome alignment of CLR (red), linked reads (green), optical maps (blue), and Hi-C reads (purple) of the Anna's hummingbird, along with telomere motif (TTAGGG and its reverse complement, yellow) and gaps (grey) using Asset software[103]. For each data type, the first row shows the mapped coverage, and the second shows the number of counts of low coverage or signs of collapsed repeats. Larger chromosomal scaffolds (1–19) have fewer gaps and low coverage or collapsed regions compared with the micro chromosomes (20–33). Chromosomes 14, 15 and 19 of the Anna's hummingbird were the most structurally reliable scaffolds, having only one gap each with no low-support regions. We defined reliable blocks as those supported by at least two technologies. Reliable blocks excluded regions with structural assembly errors, such as collapsed repeats or unresolved segmental duplications. Low-support regions are those where the reliable blocks row has a peak.

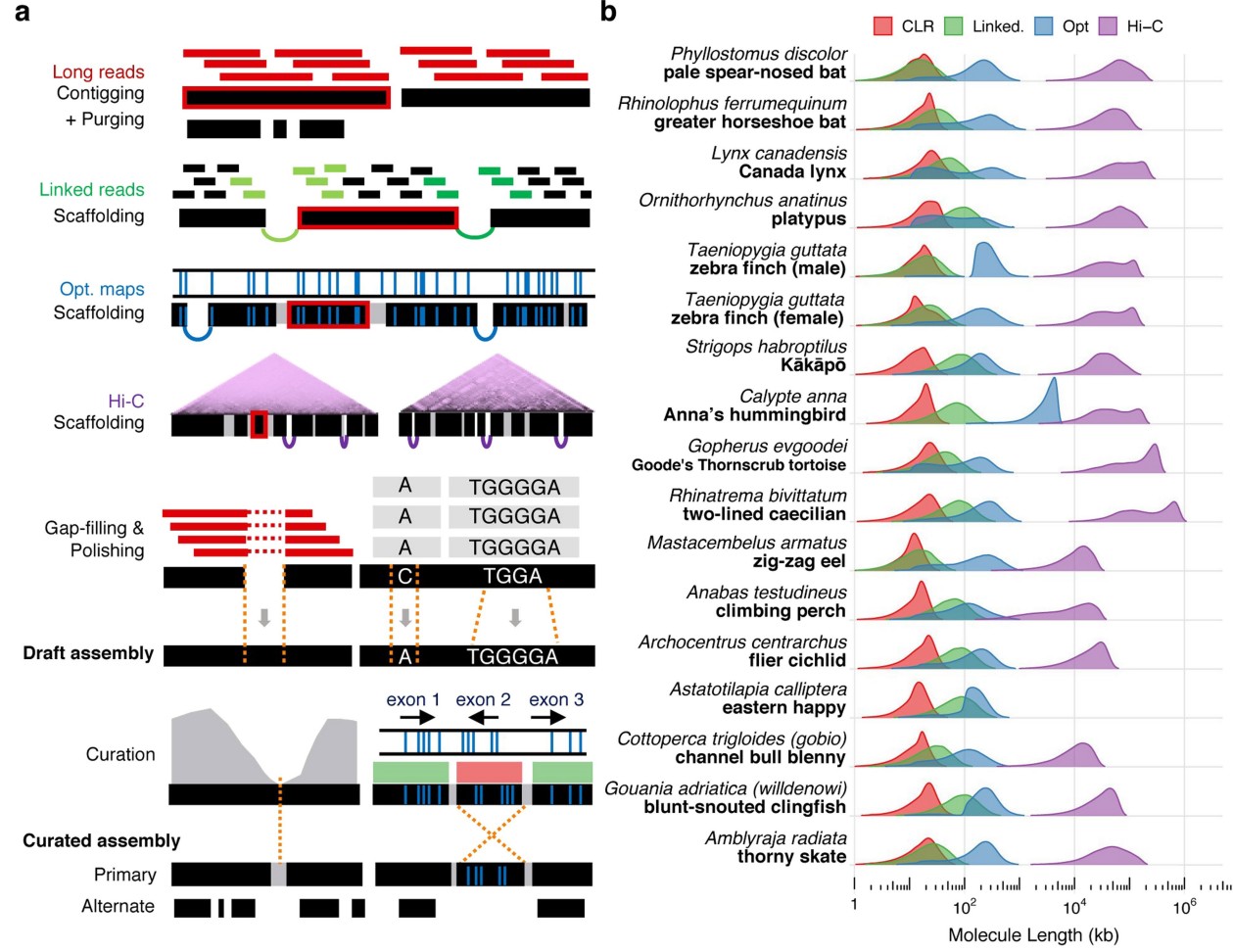

**Extended Data Fig. 2 | VGP assembly pipeline applied across multiple species. a**, Iterative assembly pipeline of sequence data types (coloured as in **b**) with increasing chromosomal distance. Thin bars, sequence reads; thick black bars, assembled contigs; black bars with space and arcing links, scaffolds; grey bars, gaps placed by previous steps; thick red border, tracking of an example contig in the pipeline. The curation step shows an example of a mis-assembly break identified by sequence coverage (grey, left) and an example of an inversion error (right) detected by the optical map. **b**, Intra-molecule length distribution of the four data types used to generate the assemblies of 16 vertebrate species, weighted by the fraction of bases in each length bin (log scaled). Molecule length above 1 kb was measured from read length for CLR, estimated molecule coverage for linked reads, raw molecule length for optical maps, and interaction distance for Hi-C reads. For each species, the fragment length distribution of each data type was similar to those for the Anna's hummingbird, with differences primarily influenced by tissue type, preservation method, and collection or storage conditions (unpublished data).

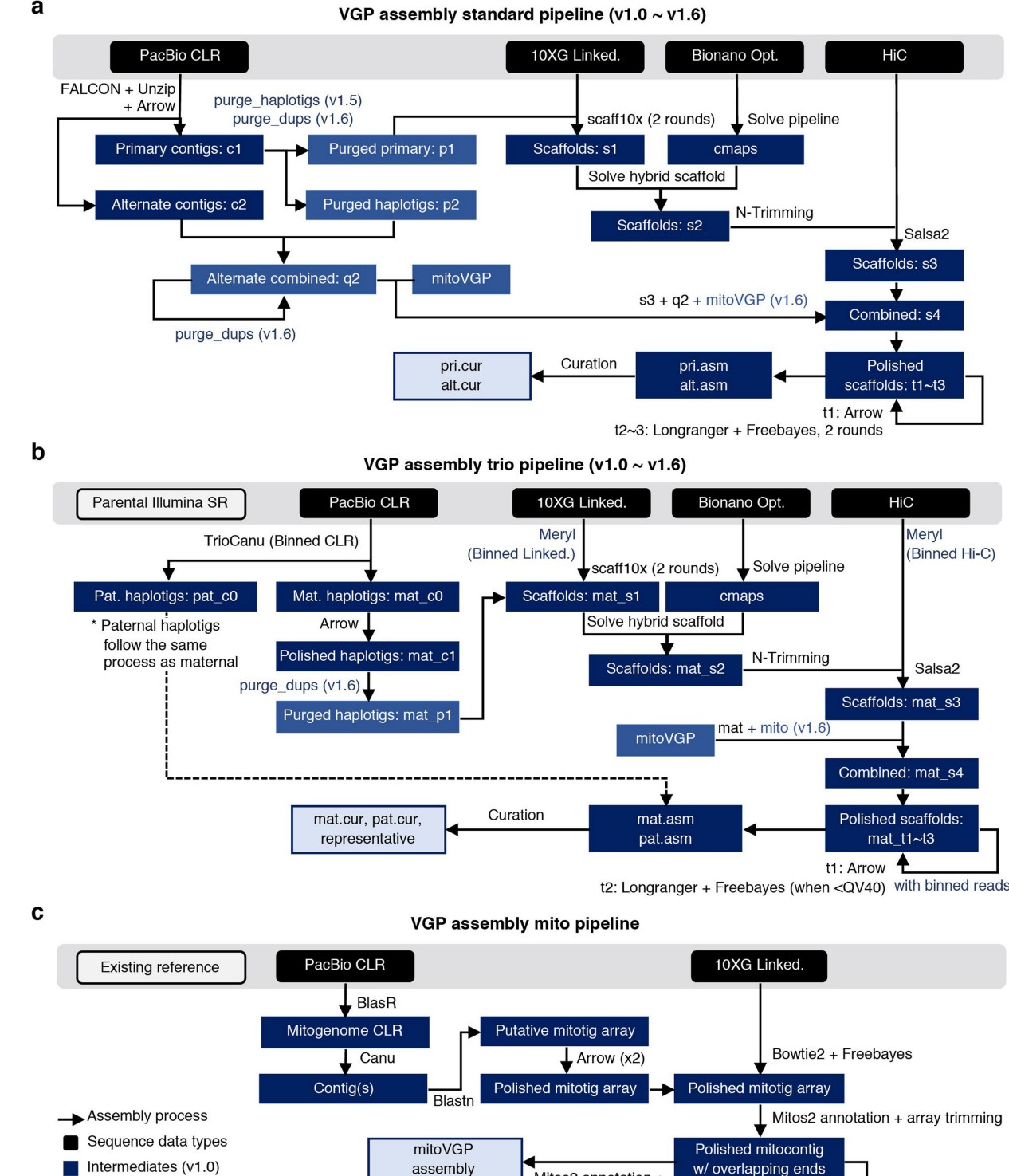

**Extended Data Fig. 3 | Flow charts of assembly pipelines used to generate high-quality assemblies in this study. a**, Standard VGP assembly pipeline when sequencing data of one individual, that generated the highest quality assemblies: generate primary pseudo-haplotype and alternate haplotype contigs with CLR using FALCON-Unzip[17]; generate scaffolds with linked reads using Scaff10x[74]; break mis-joins and further scaffold with optical maps using Solve[87]; generate chromosome-scale scaffolds with Hi-C reads using Salsa2[79]; fill in gaps and polish base-errors with CLR using Arrow (Pacific BioSciences); perform two or more rounds of short-read polishing with linked reads using FreeBayes[85]; and perform expert manual curation to correct potential

assembly errors using gEVAL[25,95] **b**, Standard VGP trio assembly pipeline when DNA is available for a child and parents[20]. Dashed line indicates that the other haplotype went through the same steps before curation. In addition to the curated assemblies of both haplotypes, a representative haplotype with both sex chromosomes is submitted. **c**, Mitochondrial assembly pipeline. Figure key applies to **a**–**c**. Steps newly introduced in v1.5–v1.6 are highlighted in light blue. c, contigs; p, purged false duplications from primary contigs; q, purged alternate contigs; s, scaffolds; t, polished scaffolds. Further details and instructions are available elsewhere[33] and at https://github.com/VGP/vgp-assembly.

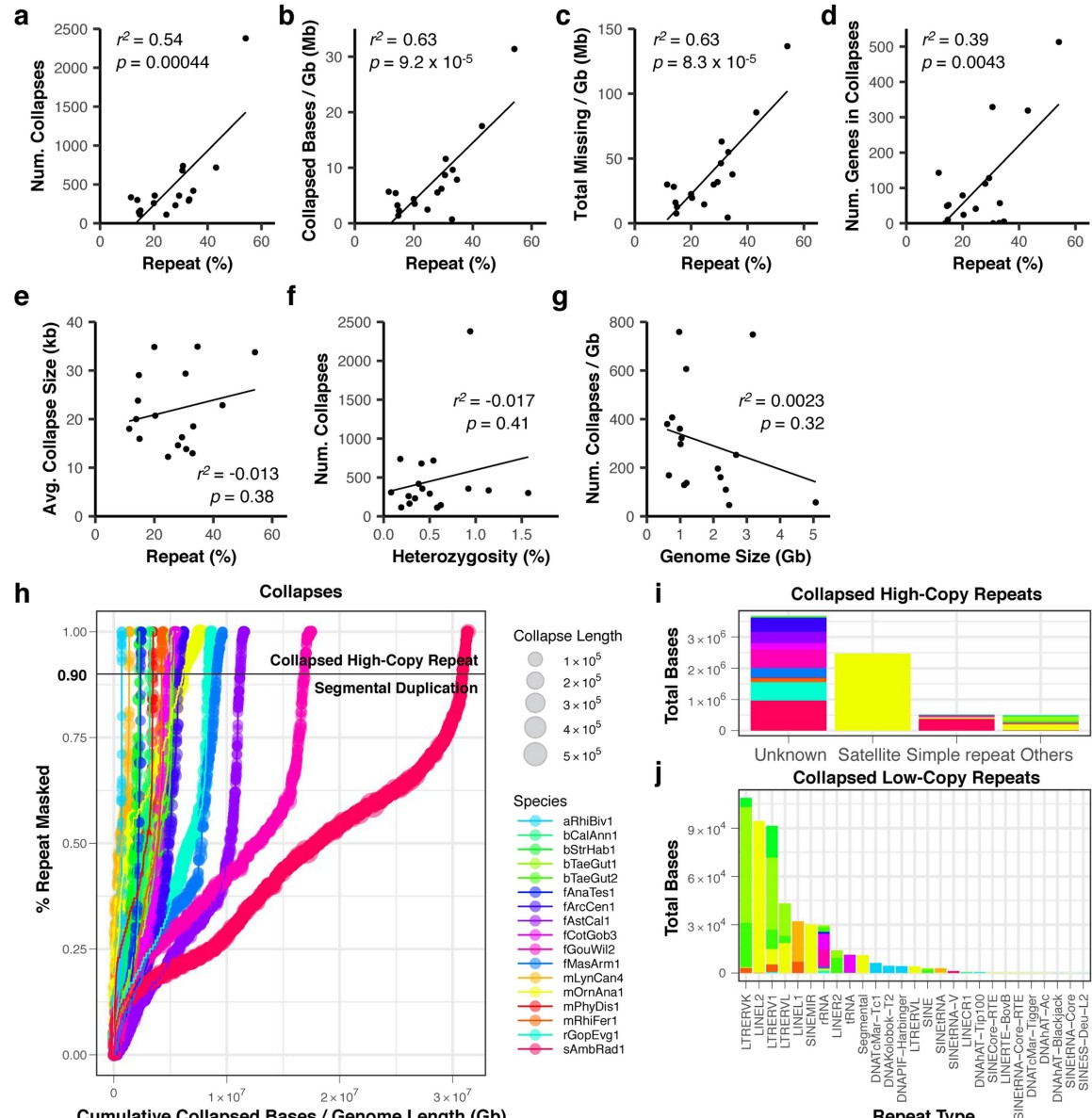

**Extended Data Fig. 4 | Relationship between collapses and genomic characteristics. a**, Correlation between the total number of collapses and percentage repeat content estimated in the submitted curated versions of *n* = 17 genomes from 16 species. **b**, Correlation between total number of bases in collapsed regions per Gb and repeat content. **c**, Correlation between total missing bases collapsed per Gb and repeat content. **d**, Correlation between total number of genes (coding and non-coding) in the collapsed regions and repeat content. **e**, Lack of correlation between the average collapsed size and repeat content. **f**, Lack of correlation between the total number of collapses and percentage heterozygosity. **g**, Lack of correlation between the total number of collapses per Gb and genome size. Genome size, heterozygosity, and repeat content were estimated from 31-mer counts using GenomeScope[71]. Reported are adjusted *r²* and *P* values from *F*-statistics. **h**, Cumulative collapsed bases per Gb in each collapse and percentage repeat masked. Each circle is

coloured by species with its size relative to the length of the collapse as it appears in the assembly. Collapses above the horizontal bar (>90%) are further classified as collapsed high-copy repeats, and those below the horizontal bar are classified as segmental duplications (low-copy repeats). **i**, Major repeat types in collapsed high-copy repeats. Most of the repeats were masked only with WindowMasker[75], with no annotation available by RepeatMasker[104]. **j**, Minor repeat types in collapsed repeats. This is a breakdown of the repeats categorized as 'Others' in **i**, owing to the smaller scale. Bar colours in **i** and **j** are as in **h**. Note smaller scale in **j** compared with **i**. Collapsed satellite arrays were almost exclusively found in the platypus, comprising ~2.5 Mb. Collapsed simple repeats were the major source in the thorny skate (~400 kb). There was a higher proportion of LTRs in birds, LINEs and SINEs in mammals, and DNA repeats in the amphibian. Among the genes in the collapses, many were repetitive short non-coding RNAs. *P* values from *F*-statistics.

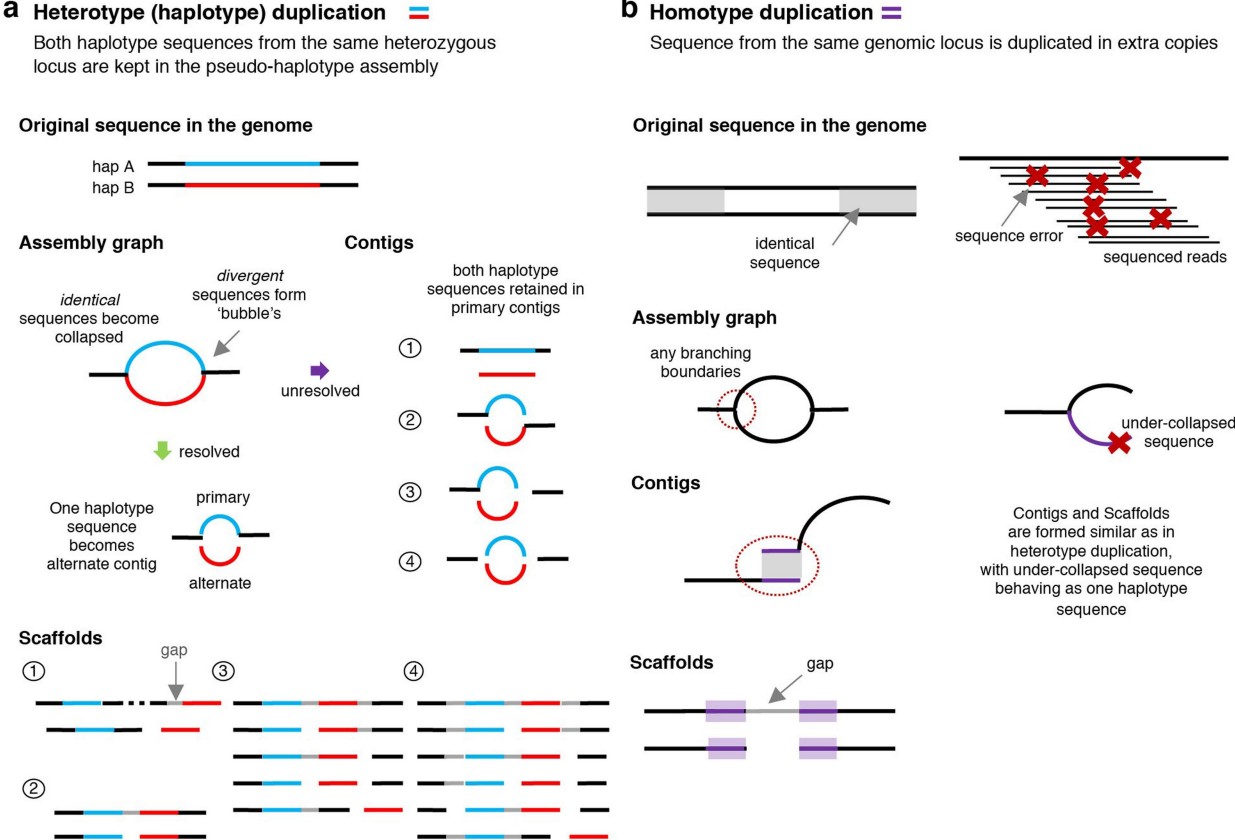

**a** **Heterotype (haplotype) duplication** ▬▬

Both haplotype sequences from the same heterozygous locus are kept in the pseudo-haplotype assembly

**Original sequence in the genome**

hap A
hap B

**Assembly graph**

*identical* sequences become collapsed

*divergent* sequences form 'bubble's

unresolved

resolved

One haplotype sequence becomes alternate contig

primary

alternate

**Contigs**

both haplotype sequences retained in primary contigs

①
②
③
④

**Scaffolds**

① gap ③ ④
②

**b** **Homotype duplication** ▬

Sequence from the same genomic locus is duplicated in extra copies

**Original sequence in the genome**

identical sequence

sequence error

sequenced reads

**Assembly graph**

any branching boundaries

under-collapsed sequence

**Contigs**

Contigs and Scaffolds are formed similar as in heterotype duplication, with under-collapsed sequence behaving as one haplotype sequence

**Scaffolds** gap

**Extended Data Fig. 5 | False duplication mechanisms in genome assembly. a**, False heterotype (haplotype) duplications occurs when more divergent sequence reads from each haplotype A (blue) and B (red) (maternal and paternal) form greater divergent paths in the assembly graph (bubbles), while nearly identical homozygous sequences (black) become collapsed. When the assembly graph is properly formed and correctly resolved (green arrow), one of the haplotype-specific paths (red or blue) is chosen for building a 'primary' pseudo-haplotype assembly and the other is set apart as an 'alternate' assembly. When the graph is not correctly resolved (purple arrow), one of four types of pattern are formed in the contigs and subsequent scaffolds. Depending on the supporting evidence, the scaffolder either keeps these haplotype contigs on separate scaffolds or brings them together on the same scaffold, often separated by gaps: 1. Separate contigs: both contigs are retained in the primary contig set, an error often observed when haplotype-specific sequences are highly diverged. 2. Flanking contigs: the assembly graph is partially formed, connecting the homozygous sequence of the 5′ side to one haplotype (blue) and the 3′ side to the other haplotype (red). 3. Partial flanking contigs: only one haplotype (blue) flanks one side of the homozygous sequence. 4. Failed connecting of contigs: all haplotype sequences fail to properly connect to flanking homozygous sequences. **b**, False

homotype duplications occur where a sequence from the same genomic locus is duplicated, and are of two types: 1. Overlapping sequences at contig boundaries: in current overlap-layout-consensus assemblers, branching sequences in assembly graphs that are not selected as the primary path have a small overlapping sequence (purple), dovetailing to the primary path where it originated a branch. The size of the duplicated sequence is often the length of a corrected read. Subsequent scaffolding results in tandem duplicated sequences with a gap between. 2. Under-collapsed sequences: sequencing errors in reads (red x) randomly or systematically pile up, forming under-collapsed sequences. Subsequent duplication errors in the scaffolding are similar to the heterotype duplications. Purge_haplotigs[13] align sequences to themselves to find a smaller sequence that aligns fully to a larger contig or scaffold, and removes heterotype duplication types 1, 3, and 4. Purge_dups[14] additionally uses coverage information to detect heterotype duplication type 2 and homotype duplications. We distinguished the two types of duplications by: 1) haplotype-specific variants in reads aligning at half coverage to each heterotype duplication; 2) differing consensus quality that resulted from read coverage fluctuations when aligning reads to homotype duplications; and 3) *k*-mer copy number anomalies in which homotype duplications were observed in the assembly with more than the expected number of copies.

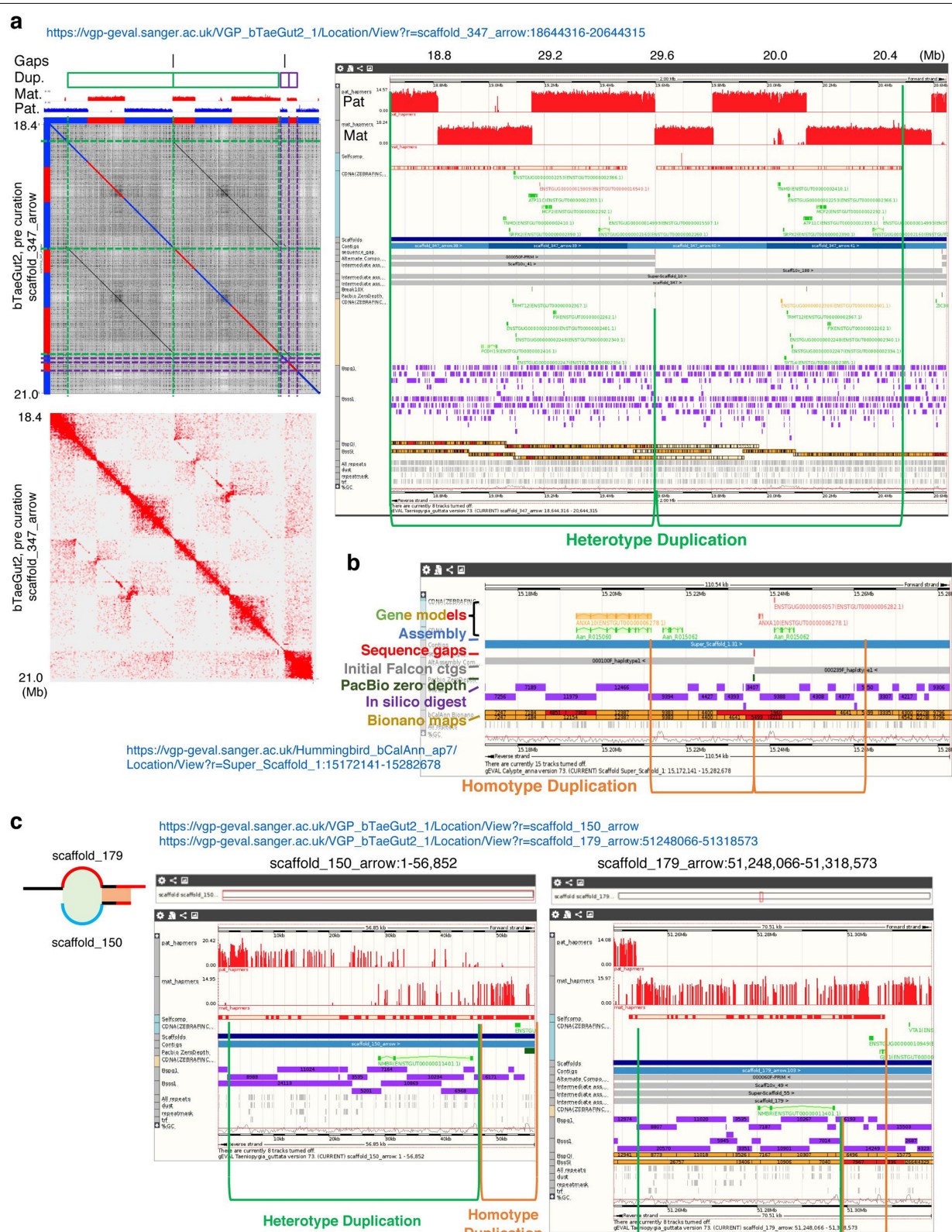

**Extended Data Fig. 6 | False duplication examples fixed during manual curation. a**, An example of a heterotype duplication in the female zebra finch, non-trio assembly. Left, a self-dot plot of this region generated with Gepard[105], with sequences coloured by haplotypes. Gaps, duplicated sequences (green and purple), and haplotype-specific marker densities are indicated at the top. Right, a detailed alignment view of the green haplotype duplication with paternal and maternal markers, self-alignment components, transcripts annotated, contigs, bionano maps, and repeat components displayed in gEVAL[95]. **b**, Example of a homotype duplication found in the hummingbird assembly. These were caused by an algorithm bug in FALCON, which was later fixed. **c**, Example of a combined duplication involving both heterotype (green) and homotype (orange) duplications. Assembly graph structure is shown on the left for clarity, highlighting the overlapping sites at the contig boundary shaded following the duplication type. Assembly errors including the above false duplications were detected and fixed during the curation process.

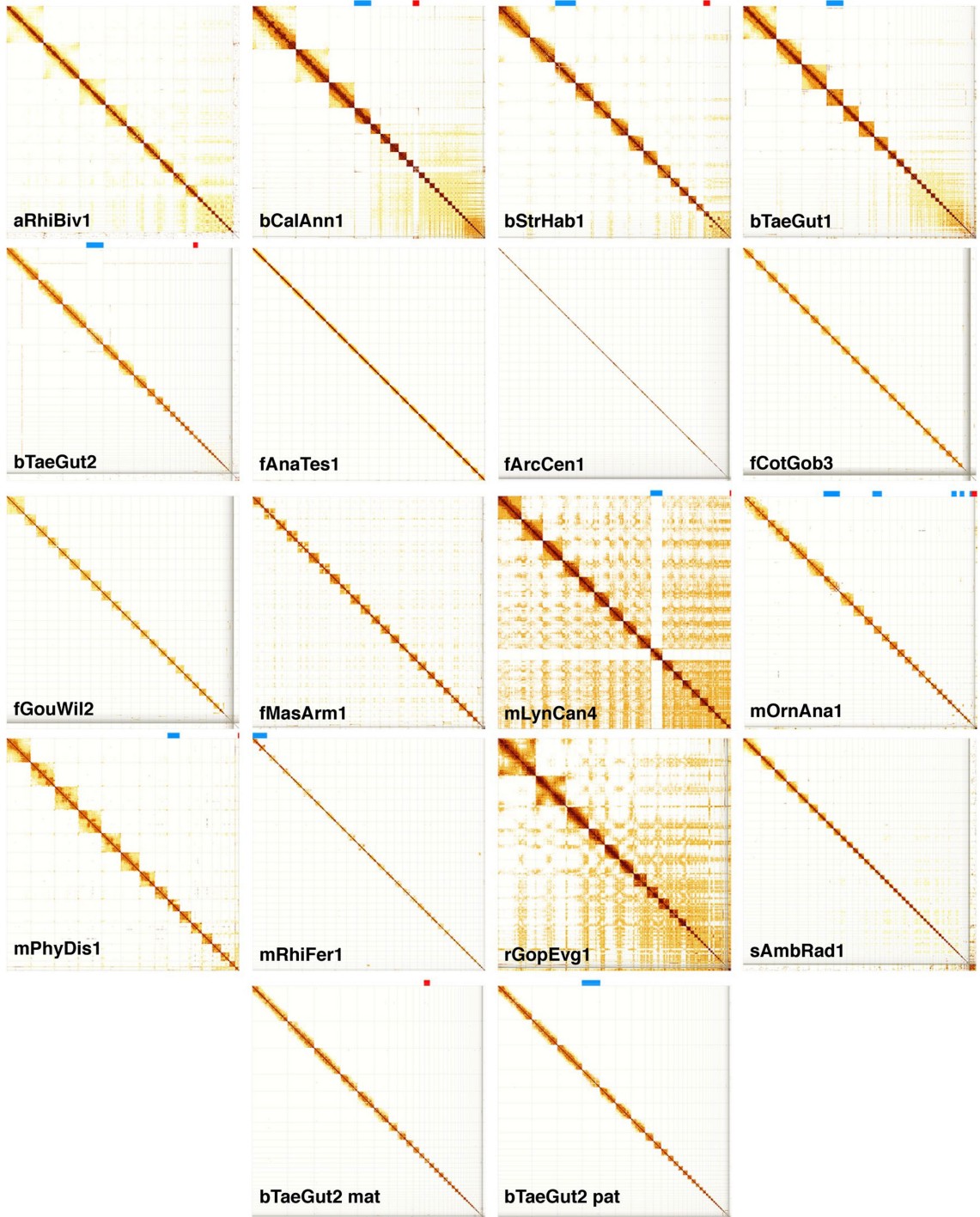

**Extended Data Fig. 7 | Evidence of near-complete chromosome scaffolds in the VGP assemblies.** Shown are Hi-C interaction heat maps for each species after curation, visualized with PretextView[106]. A scaffold is considered a putative arm-to-arm chromosome when all Hi-C read pairs in a row and column map to a square (that is, an assembled chromosome) on the diagonal without any other interactions off the diagonal. Those with remaining off-diagonal matches to smaller scaffolds are not linked because of ambiguous order or orientation, and are instead submitted as 'unlocalized' belonging to the relevant chromosome. Bands at the top of each heat map show scaffolds identified as X, Z (blue) or Y, W (red) sex chromosomes. The Hi-C map of fAstCal1 is not included as we had no remaining tissue left of the animal used to generate Hi-C reads.

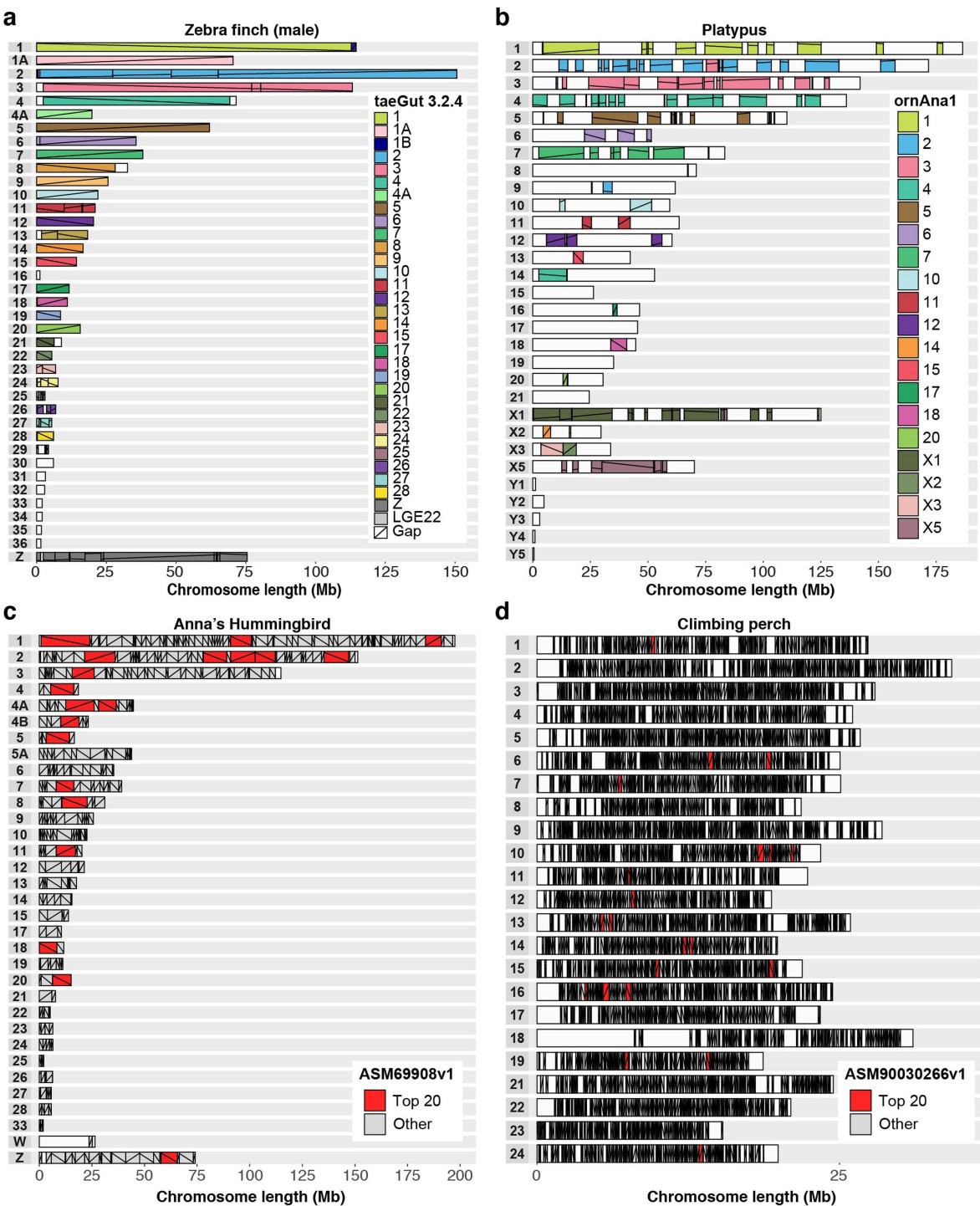

**Extended Data Fig. 8 | Comparison of chromosomal organization between previous and new VGP assemblies. a**, Zebra finch male compared to a previous reference assembly of the same animal. **b**, Platypus male compared with a previous reference female assembly (so the Y chromosomes are absent in the previous reference). **c**, Hummingbird female compared to a previous reference of the same animal. **d**, Climbing perch compared to a previous reference. Each row represents a VGP-generated chromosome for the target species. Colours depict identity with the reference (see key to the right); more than one colour indicates reorganization in the VGP assembly relative to the reference. The lines within each block depict orientation relative to the reference; a positive slope is the same orientation as the reference, whereas a negative slope is the inverse orientation. Gaps are white boxes with no lines, in the reference relative to the VGP assembly. A white box for the entire chromosome means a newly identified chromosome in the VGP assembly. Top 20 is the longest 20 scaffolds of the hummingbird and climbing perch assemblies. Accession numbers of the assemblies compared are listed in Supplementary Table 19.

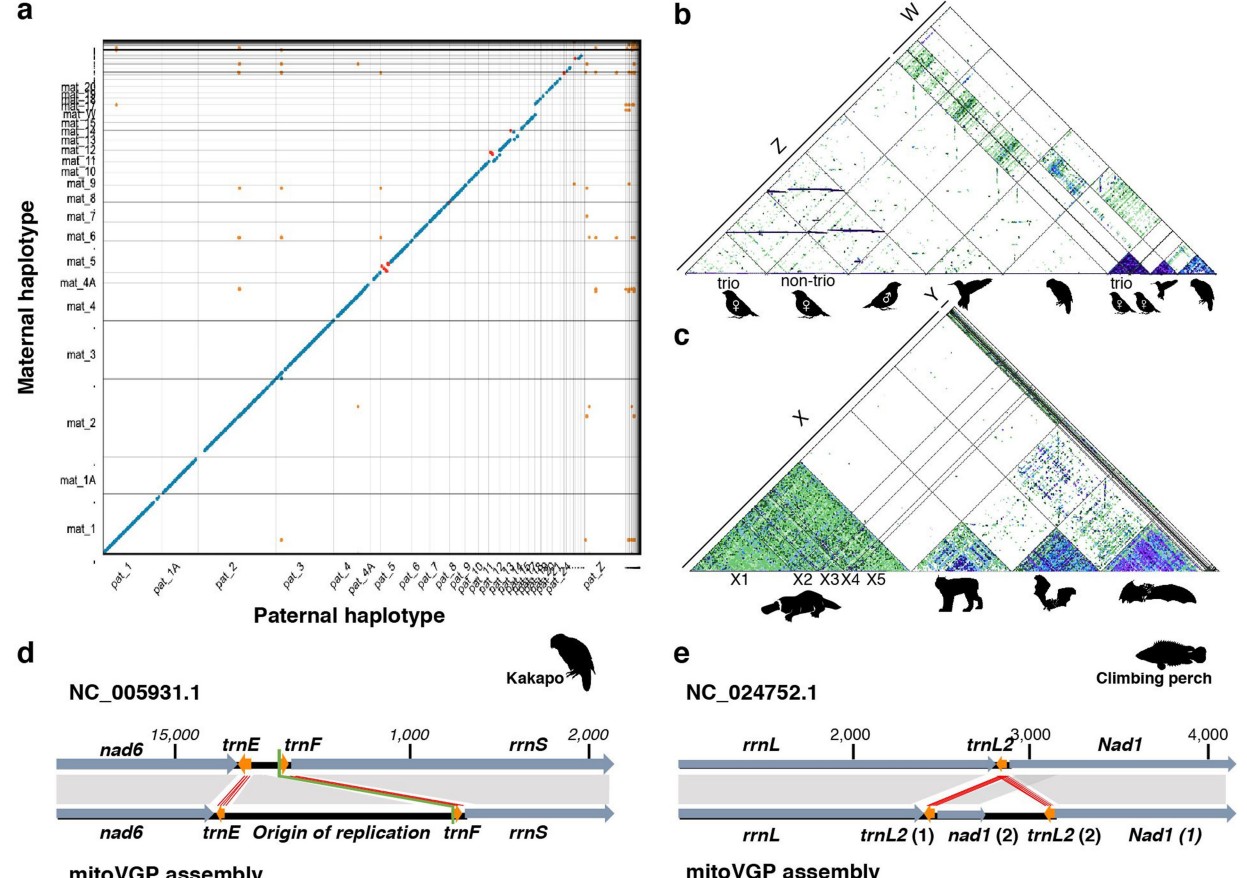

**Extended Data Fig. 9 | Haplotype-resolved sex chromosomes and mitochondrial genomes. a**, Alignment scatterplot, generated with MUMmer NUCmer[107], visualized with dot[108], of maternal and paternal chromosomes from the female zebra finch trio-based assembly. Blue, same orientation; red, inversion; orange, repeats between haplotypes. The paternal Z chromosome is highly divergent from the maternal W, and thus mostly unaligned. **b**, Alignment scatterplot of assembled Z and W chromosomes across the three bird species, approximated with MashMap2[86]. Segments of 300 kb (green), 500 kb (blue), and 1 Mb (purple) are shaded darker with higher sequence identity, with a minimum of 85%. The smaller size and higher repeat content of the W chromosome are clearly visible. **c**, X and Y chromosome segments of the

mammals (platypus, Canada lynx, pale spear-nosed bat, and greater horseshoe bat) showing a higher density of repeats within the mammalian X chromosome than the avian Z chromosome. **d**, VGP kākāpō mitochondrial genome assembly reveals previously missing repetitive sequences (adding 2,232 bp) in the origin of replication region, containing an 83-bp repeat unit. **e**, VGP climbing perch mitochondrial genome assembly showing a duplication of *trnL2* and partial duplication of *Nad1*, which were absent from the prior reference. Orange arrows and red lines, tRNA genes and their alignments; dark grey arrows and grey shading, all other genes and their alignments; black, non-coding regions; green line, conventional starting point of the circular sequence.

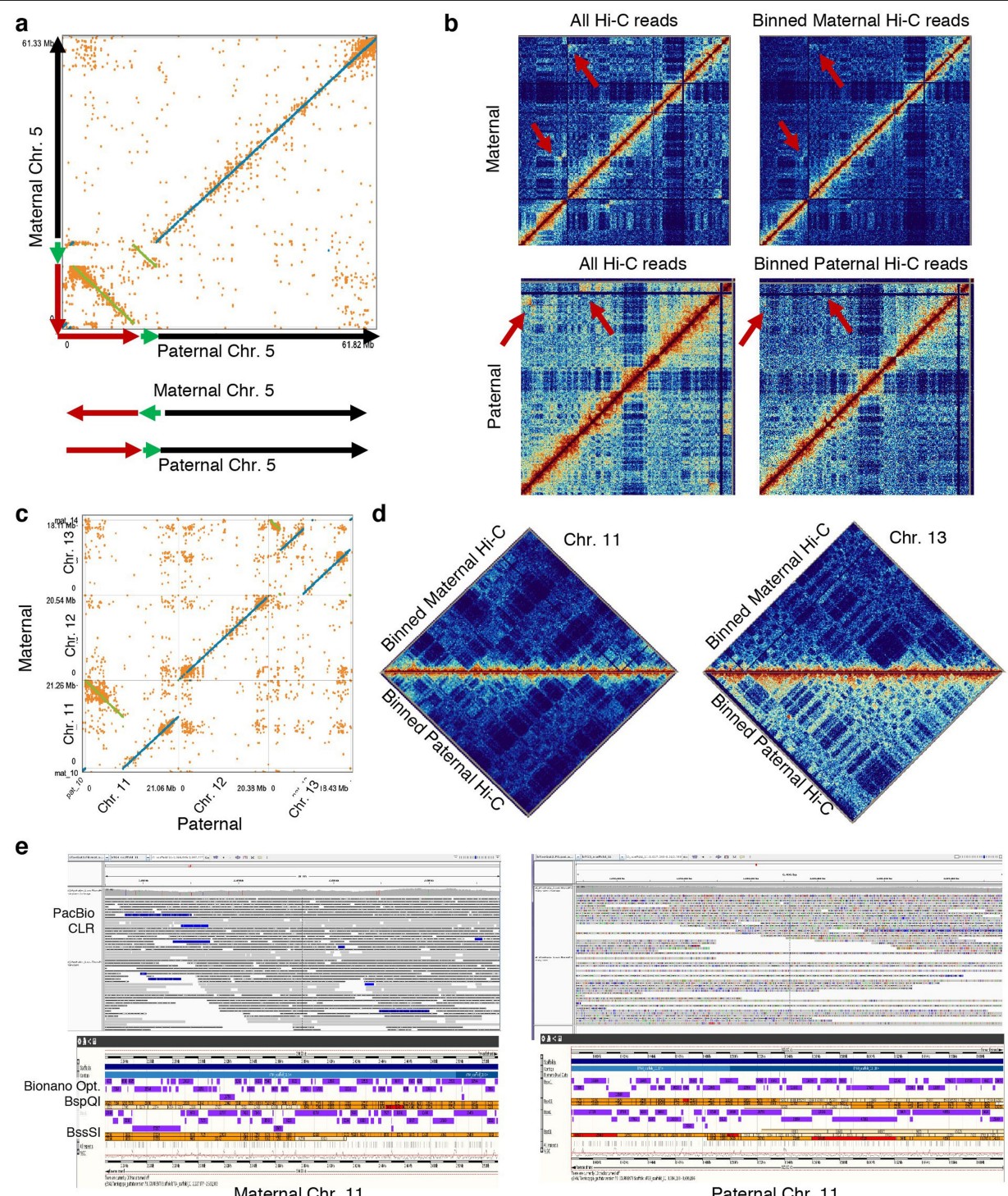

**Extended Data Fig. 10 | Large haplotype inversions with direct evidence in the zebra finch trio assembly. a**, Two inversions (green and red) in chromosome 5 found from the MUMmer NUCmer[107] alignment of the maternal and paternal haplotype assemblies, visualized with dot[108]. **b**, Hi-C interaction plot showing that the trio-binned Hi-C data remove most of the interactions from the other haplotype (red arrows), which could be erroneously classified as a mis-assembly if only one haplotype was used as a reference. **c**, An 8.5-Mb inversion found on chromosome 11 and a complicated 8.1-Mb rearrangement on chromosome 13 between maternal and paternal haplotypes. **d**, No mis-assembly signals were detected from the binned Hi-C interaction plots, indicating that the haplotype-specific inversions are real. **e**, Half the PacBio CLR span and Bionano optical maps agree with the inversion breakpoints in chromosome 11, supporting the haplotype-specific inversion.

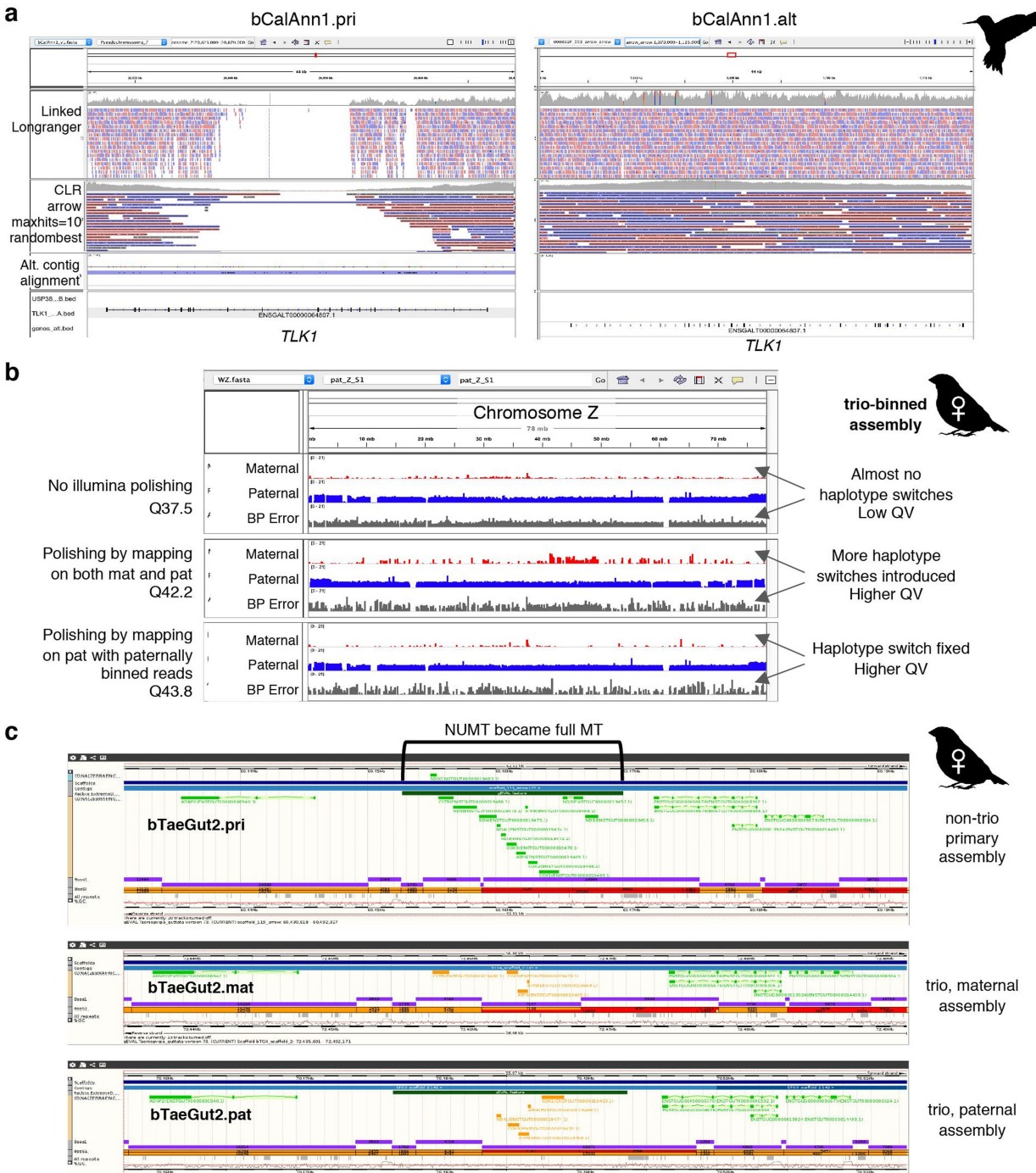

**Extended Data Fig. 11 | Polishing artefacts. a**, An example of uneven mapping coverage in the primary and alternate sequence pair of the Anna's hummingbird assembly. In this example, the alternate (alt) sequence was built at higher quality, attracting all linked-reads for polishing. The matching locus in the primary (pri) assembly was left unpolished, resulting in frameshift errors in the *TLK1* gene. **b**, Haplotype-specific markers (red for maternal, blue for paternal) and error markers found in the assembly on the Z chromosome (inherited from the paternal side) of the trio-binned female zebra finch assembly. Each row shows markers before short-read polishing, mapping all reads to both haplotype assemblies, and polishing by mapping paternally binned reads to the paternal assembly. Polishing improves QV, but introduces haplotype switch errors when using reads from both haplotypes as shown in row 2. This can be avoided when using haplotype binned reads for polishing. **c**, Example of over-polishing. The nuclear mitochondria (NuMT) sequence was transformed as a full mitochondria (MT) sequence during long-read polishing owing to the absence of the MT contig, where the NuMT attracted all long reads from the MT. In comparison, the trio-binned assembly had the MT sequence assembled in place, preventing mis-placing of MT reads during read mapping.

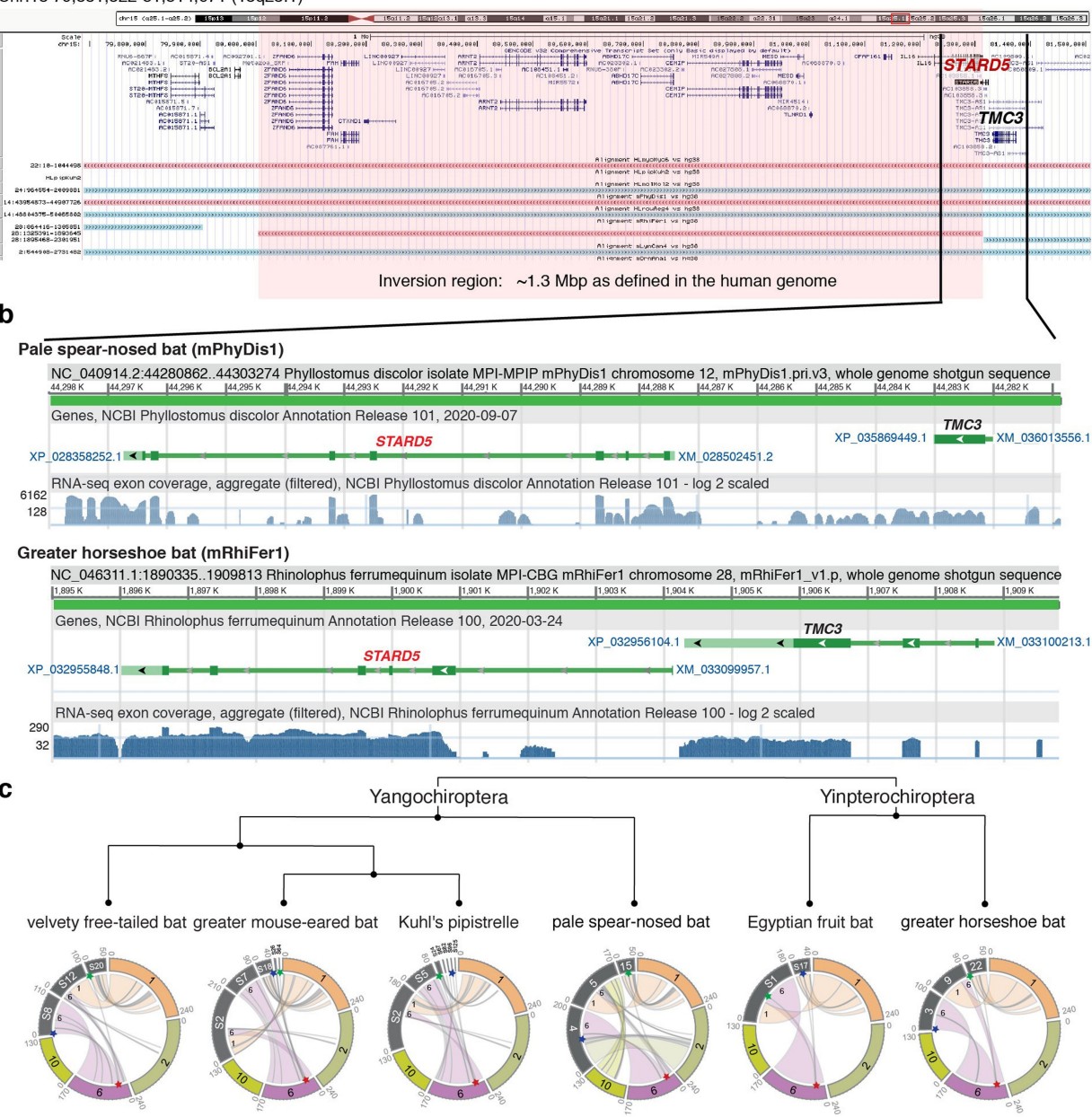

**Extended Data Fig. 12 | Chromosome evolution among the bat species sequenced. a**, Genes surrounding an inversion in the greater horseshoe bat, relative to human chromosome 15 (red highlight). The *STARD5* gene is directly disrupted by this inversion, which separates exons 1–5 from exon 6 in the greater horseshoe bat. **b**, RNA-seq tracks showing the lack of RNA splicing evidence of *STARD5* transcripts in the greater horseshoe bat (bottom) in comparison to the pale spear-nosed bat where the *STARD5* gene is not disrupted (top). **c**, Circos plots of chromosome organization relationships between the each of the analysed bats and segments of the human chromosomes 1, 2, 6 and 10. Red star, breakpoint location in human chromosome 6, depicting the

fission of the boreoeutherian chromosome 5 in the bat ancestor; blue star, the region upstream of the breakpoint in the bats; green star, the region downstream of the breakpoint in the bats. The red starred breakpoint was confirmed as reused, as opposed to assembly errors, in chromosomal rearrangements of the pale spear-nosed bat, Egyptian fruit bat, and greater horseshoe bat. There is no evidence of reuse for the velvety free-tailed bat. We could not confirm breakpoint reuse in the greater mouse-eared bat or Kuhl's pipistrelle at the chromosomal scale because they were on small scaffolds that may not be completely assembled.

# Extended Data Table 1 | Summary metrics of the curated and submitted vertebrate species assemblies

| Class | Order | Species | Common name | Genome | | Continuity | | | | | Struc. Acc. | | Base Acc. | | | Func. Comp. | Chr. Status | |
|---|---|---|---|---|---|---|---|---|---|---|---|---|---|---|---|---|---|---|
| | | | | Het (%) | Rep (%) | Prim. Size (Gb) | Alt. Size (% Prim) | Contig NG50 | Scaffold NG50 | Gaps / Gb | Reliable block NG50 | Collapsed Mb / Gb | Map. QV | k-mer QV | k-mer Comp. (%) | BUSCO Comp. (%) | Assigned to Chr. (%) | Sex Chr. |
| **Mammals** | Chiroptera | *Phyllostomus discolor* | pale spear-nosed bat | 0.9 | 20.3 | 2.1 | 97.3 | 6.4 | 171.7 | 419 | 15.0 | 3.5 | 39.1 | 38.6 | 96.8 | 93.7 | 99.88 | XY |
| | | *Rhinolophus ferrumequinum* | greater horseshoe bat | 0.3 | 20.0 | 2.1 | 73.0 | 25.2 | 84.0 | 76 | 40.2 | 4.4 | 46.9 | 41.3 | 98.0 | 96.4 | 99.28 | X |
| | Carnivora | *Lynx canadensis* | Canada lynx | 0.2 | 14.7 | 2.4 | 58.8 | 7.4 | 147.3 | 350 | 23.5 | 1.4 | 39.5 | 36.8 | 97.6 | 94.6 | 99.95 | XY |
| | Monotremata | *Ornithorhynchus anatinus* | platypus | 0.4 | 34.6 | 1.9 | 84.7 | 12.4 | 70.1 | 277 | 17.4 | 7.9 | 42.3 | 40.4 | 96.6 | 96.1 | 98.23 | X1-5 Y1-5 |
| **Birds** | Passeriformes | *Taeniopygia guttata* | zebra finch (male) | 1.1 | 11.5 | 1.1 | 91.3 | 12.1 | 71.6 | 295 | 21.5 | 5.7 | 43.4 | 41.5 | 97.2 | 98.2 | 99.25 | ZW |
| | | | zebra finch (female) | 1.6 | 13.8 | 1.1 | 85.7 | 4.4 | 73.5 | 713 | 7.2 | 5.4 | 40.8 | 39.4 | 95.4 | 97.8 | 96.08 | ZW |
| | | | zebra finch (female - mat) | 1.6 | 13.8 | 1.0 | NA | 5.4 | 71.7 | 585 | 12.0 | NA | 38.0 | 43.4 | 97.1 | 94.6 | 97.55 | W |
| | | | zebra finch (female - pat) | 1.6 | 13.8 | 1.0 | NA | 4.4 | 71.3 | 796 | 11.7 | NA | 39.1 | 43.7 | 97.1 | 97.9 | 97.16 | Z |
| | Psittaciformes | *Strigops habroptilus* | Kākāpō | 0.3 | 15.0 | 1.1 | 24.9 | 9.1 | 83.2 | 325 | 28.4 | 2.3 | 46.3 | 43.2 | 98.1 | 98.4 | 99.31 | ZW |
| | Apodiformes | *Calypte anna* | Anna's hummingbird | 0.6 | 14.4 | 1.1 | 89.9 | 12.8 | 44.7 | 405 | 33.1 | 3.2 | 40.4 | 38.6 | 94.3 | 96.2 | 99.09 | ZW |
| **Reptiles** | Testudines | *Gopherus evgoodei* | Goode's thornscrub tortoise | 0.4 | 30.6 | 2.3 | 85.3 | 10.5 | 131.6 | 245 | 17.1 | 8.7 | 41.1 | 38.4 | 96.2 | 96.2 | 94.92 | - |
| **Amphibians** | Rhinatrematidae | *Rhinatrema bivittatum* | two-lined caecilian | 0.5 | 33.0 | 5.3 | 86.8 | 3.4 | 486.9 | 672 | 6.1 | 0.7 | 38.7 | 38.2 | 96.7 | 81.9 | 97.36 | - |
| **Teloest Fishes** | Synbranchiforme | *Mastacembelus armatus* | zig-zag eel | 0.1 | 33.2 | 0.6 | 7.4 | 4.8 | 23.3 | 402 | 10.6 | 9.6 | 38.7 | 44.5 | 95.9 | 98.0 | 96.77 | - |
| | Anabantiformes | *Anabas testudineus* | climbing perch | 0.6 | 24.6 | 0.6 | 97.0 | 4.6 | 23.5 | 479 | 13.0 | 2.5 | 24.3 | 37.1 | 92.6 | 98.4 | 99.75 | - |
| | Cichliformes | *Archocentrus centrarchus* | flier cichlid | 0.4 | 29.4 | 0.9 | 110.8 | 2.0 | 35.6 | 797 | 2.6 | 6.2 | 38.4 | 34.7 | 94.5 | 97.3 | 88.79 | - |
| | | *Astatotilapia calliptera* | eastern happy cichlid | 0.2 | 30.8 | 0.9 | No alt | 4.0 | 36.7 | 557 | 2.8 | 11.6 | 34.8 | 37.4 | 93.9 | 97.1 | 96.62 | - |
| | Perciformes | *Cottoperca trigloides (gobio)* | channel bull blenny | 0.3 | 28.7 | 0.6 | 110.2 | 5.9 | 25.2 | 730 | 3.4 | 5.5 | 28.8 | 33.6 | 87.2 | 91.8 | 94.36 | - |
| | Gobiesociformes | *Gouania adriatica (willdenowi)* | blunt-snouted clingfish | 0.5 | 43.1 | 0.9 | 6.7 | 1.2 | 36.8 | 1,238 | 3.9 | 17.5 | 33.9 | 38.6 | 87.9 | 95.4 | 93.54 | - |
| **Cartilaginous fishes** | Rajformes | *Amblyraja radiata* | thorny skate | 0.9 | 54.1 | 2.6 | 57.2 | 0.8 | 49.8 | 1,467 | 2.3 | 31.4 | 38.4 | 39.3 | 87.8 | 88.6 | 95.37 | X |

Colour shading indicates degree of heterozygosity or repeats (red), primary assembly sizes and relative size of alternate haplotypes (orange), continuity measures (green), gaps and collapses (blue), and base call accuracy (purple). A dash indicates that the sex chromosomes were not found or are not known. Accessions are available in Supplementary Table 10.

**Extended Data Table 2 | Annotation summary statistics in previous and newly assembled VGP reference genomes**

| Species | | Hummingbird | | Zebra finch | | | Platypus | |
|---|---|---|---|---|---|---|---|---|
| Assembly | | ASM69908v1 | bCalAnn1_v1.p (VGP) | Taeniopygia_guttata-3.2.4 | bTaeGut1_v1.p (VGP) | bTaeGut2.pat.v3 +W (VGP) | Ornithorhynchus_ anatinus-5.0.1 | mOrnAna1.p.v1 (VGP) |
| Assembly accession | | GCF_000699085.1 | GCF_003957555.1 | GCF_000151805.1 | GCF_003957565.1 | GCF_008822105.2 | GCF_000002275.2 | GCF_004115215.1 |
| NCBI Annotation Release | | N/A | 101 | N/A | 104 | 105 | N/A | 104 |
| **All Genes** | Total | 16,234 | 16,230 | 24,719 | 22,186 | 21,543 | 34,289 | 30,932 |
| | Genes with alternative variants | 3,649 | 5,897 | 6,429 | 9,312 | 9,130 | 4,414 | 9,485 |
| **Coding genes** | Total | 14,714 | 14,711 | 18,482 | 17,438 | 16,197 | 19,883 | 18,200 |
| | With orthologs to human | 12,250 | 12,502 | 11,709 | 12,801 | 12,903 | 10,563 | 14,730 |
| | > 80% cov. by SwissProt protein | 13,330 | 13,538 | 15,569 | 15,154 | 14,090 | 16,165 | 16,486 |
| | > 80% SwissProt cov. by protein | 10,697 | 13,237 | 11,661 | 15,845 | 14,830 | 8,635 | 16,137 |
| **Problematic coding genes** | **Partial** | **1,637** | **110** | **4,109** | **212** | **289** | **5,601** | **228** |
| | with >5% *ab initio* | 1,828 | 890 | 1,557 | 702 | 534 | 4,676 | 1,334 |
| | with corrections | 909 | 1,056 | 2,658 | 739 | 468 | 1,258 | 834 |
| **CDS** | **Total** | **22,448** | **29,231** | **36,951** | **43,977** | **42,385** | **29,130** | **43,203** |
| | **Fully-supported (no *ab initio*)** | **18,280** | **27,549** | **33,879** | **42,799** | **41,466** | **20,630** | **40,677** |
| | Mean length | 1,774 | 1,949 | 1,885 | 2,263 | 2,283 | 1,494 | 2,077 |
| | Median length | 1,332 | 1,452 | 1,334 | 1,620 | 1,623 | 1,047 | 1,512 |

Annotation results of VGP assemblies and previous reference assemblies with the NCBI Eukaryotic Genome Annotation Pipeline, using the same RNA-seq data and nearly identical sets of transcripts and proteins on input. Highlighted rows are plotted in Fig. 5c, d.

# nature research

| | |
|---|---|

# Reporting Summary

Nature Research wishes to improve the reproducibility of the work that we publish. This form provides structure for consistency and transparency in reporting. For further information on Nature Research policies, see our Editorial Policies and the Editorial Policy Checklist.

## Statistics

For all statistical analyses, confirm that the following items are present in the figure legend, table legend, main text, or Methods section.

| n/a | Confirmed | |
|---|---|---|
| ☐ | ☒ | The exact sample size (*n*) for each experimental group/condition, given as a discrete number and unit of measurement |
| ☐ | ☒ | A statement on whether measurements were taken from distinct samples or whether the same sample was measured repeatedly |
| ☐ | ☒ | The statistical test(s) used AND whether they are one- or two-sided *Only common tests should be described solely by name; describe more complex techniques in the Methods section.* |
| ☒ | ☐ | A description of all covariates tested |
| ☒ | ☐ | A description of any assumptions or corrections, such as tests of normality and adjustment for multiple comparisons |
| ☐ | ☒ | A full description of the statistical parameters including central tendency (e.g. means) or other basic estimates (e.g. regression coefficient) AND variation (e.g. standard deviation) or associated estimates of uncertainty (e.g. confidence intervals) |
| ☒ | ☐ | For null hypothesis testing, the test statistic (e.g. $F$, $t$, $r$) with confidence intervals, effect sizes, degrees of freedom and $P$ value noted *Give P values as exact values whenever suitable.* |
| ☒ | ☐ | For Bayesian analysis, information on the choice of priors and Markov chain Monte Carlo settings |
| ☒ | ☐ | For hierarchical and complex designs, identification of the appropriate level for tests and full reporting of outcomes |
| ☒ | ☐ | Estimates of effect sizes (e.g. Cohen's *d*, Pearson's *r*), indicating how they were calculated |

*Our web collection on statistics for biologists contains articles on many of the points above.*

## Software and code

Policy information about availability of computer code

| | |
|---|---|
| Data collection | We collected genome sequence data of various technologies from individuals representing 16 vertebrate species. |
| Data analysis | All software used is at at https://github.com/VGP/vgp-assembly, as of November 2020. The assembly software and versions used are listed in Supplementary Table The specific VGP assembly pipeline used is at https://github.com/VGP/vgp-assembly/tree/master/pipeline. |

For manuscripts utilizing custom algorithms or software that are central to the research but not yet described in published literature, software must be made available to editors and reviewers. We strongly encourage code deposition in a community repository (e.g. GitHub). See the Nature Research guidelines for submitting code & software for further information.

## Data

Policy information about availability of data

All manuscripts must include a data availability statement. This statement should provide the following information, where applicable:
- Accession codes, unique identifiers, or web links for publicly available datasets
- A list of figures that have associated raw data
- A description of any restrictions on data availability

All raw sequence and assembly data are available at Genome Ark https://vgp.github.io/genomeark/ and the NCBI BioProject page PRJNA489243 https://www.ncbi.nlm.nih.gov/bioproject/489243

April 2020

# Field-specific reporting

Please select the one below that is the best fit for your research. If you are not sure, read the appropriate sections before making your selection.

$\boxtimes$ Life sciences      $\square$ Behavioural & social sciences      $\square$ Ecological, evolutionary & environmental sciences

For a reference copy of the document with all sections, see nature.com/documents/nr-reporting-summary-flat.pdf

# Life sciences study design

All studies must disclose on these points even when the disclosure is negative.

| | |
|---|---|
| Sample size | For most analyses, we analyzed data from a sample size of n = 16 vertebrate species, and n = 17 individuals; we had two individuals for one species (the zebra finch). For the chromosomal evolution analyses, we added an additional n = 4 bat species. |
| Data exclusions | Only sequence data that failed quality control were excluded or repeated. |
| Replication | We confirmed the ability to replicate all code using multiple rounds of assembly or analyses. |
| Randomization | No analyses required generating randomize data sets. |
| Blinding | No analyses required being blind to groups |

# Reporting for specific materials, systems and methods

We require information from authors about some types of materials, experimental systems and methods used in many studies. Here, indicate whether each material, system or method listed is relevant to your study. If you are not sure if a list item applies to your research, read the appropriate section before selecting a response.

## Materials & experimental systems

| n/a | Involved in the study |
|---|---|
| $\boxtimes$ | $\square$ Antibodies |
| $\boxtimes$ | $\square$ Eukaryotic cell lines |
| $\boxtimes$ | $\square$ Palaeontology and archaeology |
| $\square$ | $\boxtimes$ Animals and other organisms |
| $\boxtimes$ | $\square$ Human research participants |
| $\boxtimes$ | $\square$ Clinical data |
| $\boxtimes$ | $\square$ Dual use research of concern |

## Methods

| n/a | Involved in the study |
|---|---|
| $\boxtimes$ | $\square$ ChIP-seq |
| $\boxtimes$ | $\square$ Flow cytometry |
| $\boxtimes$ | $\square$ MRI-based neuroimaging |

# Animals and other organisms

Policy information about studies involving animals; ARRIVE guidelines recommended for reporting animal research

| | |
|---|---|
| Laboratory animals | The care and collection of a laboratory female zebra finch sample was done under an approved IACUC protocol at the Rockefeller University. The male laboratory sample was obtained from the approval mention of the previous reference genome in Warren et al 2011 Nature. |
| Wild animals | Samples of the 15 other species were collected from wild animals, with approved permits of the source institutions and local governments involved. These sources, persons with the permits, geographic location, sex and relative age are listed in Supplementary Table 8 and the BioSample submissions in NCBI and ENA. |
| Field-collected samples | No laboratory work was conducted on field-collected animals |
| Ethics oversight | Rockefeller University for the zebra finch species. |

Note that full information on the approval of the study protocol must also be provided in the manuscript.

