## [Peer Review File · Nature]

Peer Review File**Manuscript Title:**

Towards complete and error-free genome assemblies of all vertebrate species

Editorial Notes:**Reviewer Comments & Author Rebuttals****Reviewer Reports on the Initial Version:****Ref #1**

Rhie et al present "Towards complete and error-free genome assemblies of all vertebrate species" that describes the current status and initial data release of the Vertebrate Genome Project (VGP). The paper is expansive describing the VGP sequencing and assembly process, results from 16 new reference genomes, a review of the challenges encountered by these assemblies, and a prospective on future work needed in the field. The paper presents many interesting results, although several major issues need to be addressed.

- The authors assert they have developed "optimal assembly approaches", yet the presented pipeline is not viable for future projects: 10X Genomics no longer offers linked reads sequencing and the long-read data were generated on out of date RSII and Sequel I platforms. More discussion is needed on what are the best options using current technology. The most obvious issue is the current draft includes only passing mentions of ONT ultra long reads and PacBio HiFi data but we have found them extremely valuable, often more so than PacBio CLR reads (as do many of the authors of this paper such as for the HiCanu algorithm or the Telomere-to-Telomere project). The approaches for polishing, for example, will change substantially when using HiFi reads.
- More generally, there is minimal discussion on the remaining challenges: What will be the recommended assembly process in the future? What will be the process for curating >70,000 genomes? How will tens of thousands of genomes be annotated or systematically analyzed together?
- The review aspects of the paper (Figures 1-3) describes many results that are extremely well established in the literature across many species: long-read assemblies are more continuous; repeats affect continuity; false duplications are common assembly errors, trio-binning improves assembly, etc. These sections should be minimized with appropriate citations e.g. (Pendleton et al, 2015 Nature Methods; Gordon et al, 2016, Science; Matthews et al, 2018, Nature; Nowoshilow et al, 2018, Nature; Liu et al, 2020, Cell; etc).
- There are surprisingly few details presented on the bioinformatics approaches involved. The raw data offer very little value without these software; The authors should describe the key algorithms and techniques with citations.
- The VGP team has made very valuable data contributions through the improved reference genomes, although it seems these genomes are all being described in other companion papers. For example, all of the results in the "New biological discoveries" section are "in press" or "in submission". This reduces the potential impact of this paper and the authors should clarify which results, if any, are unique to this paper
- Table 1 is a useful decomposition of assembly quality, but many of the fields lack justification of the thresholds used: why only 1Mbp haplotype phase blocks for a 7.C.Q50 assembly, and why only measure gene content and not other functional elements, etc. The authors admit "required quality

of an assembly depends on the biological questions" yet the examples are presented without supporting data or justification.

- For each metric in Table 1, the authors should describe in detail how it could be computed/estimated from the data and the limitations for measuring them. Relatedly, instead of reporting raw contig NG50 or scaffold NG50 metrics for the X.Y.Q quality metrics, these should values be defined using reliable blocks since contig length and scaffold length are easily inflated. Other metrics need more precision: "the relatively clean Hi-C heatmap" is insufficient.

Ref #2

The authors present an impressive set of assemblies of diverse vertebrate species that are much better than previous assemblies. They use a combination of many technologies and present a lot of evaluations that will be useful for the community. They rely primarily on PacBio CLR for generating contigs, since it performed best of the technologies available when most of this work was performed. They develop a set of useful metrics to summarize different aspects of assembly performance. The QV40 of these assemblies is lower than most current and future assemblies will be, due to the advent of much more accurate PacBio HiFi long reads in the past year, but QV40 is sufficient for many analyses like annotation of many genes and comparisons between species. The authors also present a really interesting discussion about the prevalence of erroneous duplications in assemblies when they are not done carefully. Besides the smaller suggestions below, my primary concern is that there should be more discussion of the likelihood of collapsed segmental duplications in the assemblies, and ideally evaluations that show how prevalent these collapses are. Otherwise, I expect this manuscript will be very useful to the community and represents a large amount of excellent technical work that I expect will continue to lead to new discoveries and future improvements in assemblies.

1. In the Fig 1 caption, this sentence was not clear to me "*Assembly generated with software updates to FALCON29, following CLR + Linked + Opt. + Hi-C combination." This makes it sound like v1.0 is the same as "CLR + Linked + Opt. + Hi-C" even though they are different assemblies. As I understand from later in the text, the difference is only the updated FALCON version?
2. Fig 1H would be easier to see on a log-log scale or with a separate zoomed inset since most of the chromosomes are much smaller than the first few
3. It would be useful to state the reason why short reads have much shorter contig lengths (e.g., due to breaks at tandem repeats, mobile elements, or other repeats?) after stating "These findings indicate that given current sequencing technology and assembly algorithms, it is not possible to achieve high contig continuity with short reads alone."
4. State how Q40 was determined earlier on (short read mapping and/or kmer-based)
5. The question of how well segmental duplications and other difficult regions are represented is one of the biggest holes in the current manuscript. I suspect that they were frequently collapsed or not represented well using the technologies and methods available at the time of this work, which is understandable. I don't expect the authors to improve their assemblies, but it is important that this limitation be highlighted and ideally quantified in some metrics. Can you add a metric related to potential errors in segmental duplications and other repeats to Fig 3a (e.g., identifying regions with higher than expected coverage)?
6. Can you add a clearer definition of "repeat" to the legend of Fig 3 than just referencing GenomeScope, since there are so many types of repetitive genomic elements?
7. On page 16, they state "The total number of genes annotated went down in the VGP assemblies (Extended Data Table 1), which we believe is because the VGP assemblies have fewer false duplications (0.5 to 1.7%) compared with their previous counterparts (5.1 to 12.3%; except for the climbing perch at 0.4 vs 1.2%; Supplemental Table 19)." Wouldn't it be possible to show that the specific genes in the old assemblies and not in the new assemblies are due to duplications?
8. On page 16, could the authors state what the +- values are (e.g., StDev), and do a statistical

test to see if this is a significant change? "Further supporting these results, the VGP assemblies had 6 to 13% higher k-mer completeness (95+-3.5% average; Fig. 3a) compared with the prior assemblies (88+-4.3% average; Supplementary Table 19)."

9. In Fig 5b, could the author's clarify what the horizontal line in each bar means?

10. On pages 19-20, the authors state "We further note here, that base error rate inferred directly from k-mers was more comprehensive and accurate than the widely used mapping and variant calling protocols, which artificially inflated QV values (Supplementary Table 17) because they exclude regions that are difficult to map." While this is true, it would be good for the authors to note that k-mer approaches also can artificially inflate QV values, particularly in collapsed segmental duplications or other repeats where the small k-mers used may be accurate even if the assembly is not. Merqury's k-mer based metric is a good way to get QV for the VGP assemblies, but it's important to note its strengths and limitations.

11. They also state "VGP assemblies exceeding Q40 contained fewer frameshift errors, as predicted⁷⁶, and therefore we recommend targeting a minimum QV of 40 whenever possible." This seems like a somewhat arbitrary target, and mostly based on the fact that it is what the VGP achieved for most assemblies. Given the advent of more accurate PacBio HiFi sequencing, which can result in assembly QV's >50, it seems like an outdated target. However, it is helpful that the authors state in the next paragraph what different QV40 assemblies are useful for.

12. It would be good to have a more descriptive README for the VGP code repository at <https://github.com/VGP/vgp-assembly>

Ref #3

I enjoyed reading the manuscript "Towards complete and error-free genome assemblies of all vertebrate species" by VGP consortium. The study is a tour de force on data generation and assembly methods focused on 16 vertebrate species. Starting from the assembly of a hummingbird genome, the study sets up a pipeline that is then applied to other 15 vertebrates to produce high contiguity assemblies. The starting raw data is highly diverse and comprises all the state-of-the-art sequencing methods available; it will be an amazing resource for the scientific community. The study is well designed and written, and it is mostly flawless at the technical level. Overall, this is a titanic task and I would like to congratulate the authors for their work and contribution.

However, the results lack novelty or new biological insights. As indicated by the headers of most sections, which are almost tautological, all the results are confirmatory and technical: "Long-read assemblies are more continuous", "Repeats affect continuity", "False duplications are common assembly errors", "Curation is important for generating a high-quality reference", "Trios help resolve haplotypes"... those are basic principles of genome assembly and have been known and documented for very long time. When things start to get interesting ("New biological discoveries"!!!) the reader is sent to a list of other papers recently submitted or published by members of the consortium, which makes clear no biology was not to be included in this manuscript.

A methodological issue is a suggestion of using a single pipeline to assemble different vertebrate genomes, a pipeline which has been set up based on a single genome (hummingbird). This goes against the advice from other seasoned initiatives, like Assemblathon (this referee has no affiliation to this project). Those show that for each combination of species and raw genomic data, different combinations of software and approaches are necessary to reach the best assembly possible. The genome assembly of each species is a new and unique bioinformatics experiment, that requires a different set of tools. This goes against the one-size-fits-all approach applied and proposed in this manuscript.

I have some other comments and questions that I hope the authors can kindly address:

- 1) I understand the advantages of NG50, and I agree with its use to assess assembly quality. However, for comparison and legacy reasons, I believe N50 should be still included in tables and similar.
- 2) A big emphasis is made on long contiguity assemblies, which favours tech like PacBio and ONT. I completely agree with this view. However, there is almost no consideration to error rate, only to discard an ONT dataset at one point. Of course, we need to know the "real" sequence to infer error rate, which we do not have. But maybe the different assemblies from one genome could be mapped against the preferred one to calculate how much they do diverge at the nucleotide level.
- 3) Line 278, I'd suggest changing "correct" to "best assembly" or similar.
- 4) Line 409, I do not agree that repeats affecting contiguity in assemblies is a hypothesis, I think it is a well-established fact. Hence the need for long-read technologies, optical mapping, HiC, etc.
- 5) I find the metrics in Table 1 confusing and not that helpful, but this might be just me. Unless the genome is considered "complete", the other columns are hard to compare between species with different genome sizes.

Author Rebuttals to Initial Comments:**Editor's comments:**

Your manuscript entitled "Towards complete and error-free genome assemblies of all vertebrate species" has now been seen by 3 referees, whose comments are attached below. While they find your work of potential interest, as do we, they have raised important concerns that in our view need to be addressed before we can consider publication in Nature.

Specifically, we want to highlight that the reviewers have mentioned that some of the technology for generating assemblies is or soon will be outdated/superseded, that a one-size-fits-all approach for genomes does not appear appropriate and that biological insights from the here reported dataset is largely missing. In light of these comments regarding the technical aspects and novelty of technology, we ask that you please add further biological insights to the study as only referring to other companion papers would not appear sufficient overall.

Response: Thank you for the helpful feedback, and for your patience as we worked hard to address the comments of the reviewers. We focused specifically on adding more biological novelty to the main paper, which we believe significantly improved the relevance of our work. In particular, we have added two additional figures that highlight novel GC-rich promoter sequences that are systematically absent from prior short-read assemblies across all classes of vertebrates (**Fig. 6**), as well as a chromosome breakpoint analysis across all 16 assembled vertebrate genomes (**Fig. 7**). We have also added new text and a supplemental figure (**Extended Data Fig. 10**) to describe these findings.

In analyzing what sequences were improved in the VGP assemblies compared to prior reference genomes, we found that between 20% to 50% of the genes in prior Sanger-based and Illumina-based assemblies were missing parts of the promoter region, 5' UTR, and 5' exon sequences, and this was

positively correlated with GC-content. We also found the majority of protein coding genes have a GC-rich pattern whose differential magnitude is much less in the older assemblies, due to biases against sequencing GC-rich regions. In particular, for protein coding genes, there was a sharp rise of GC-content in CpG islands in the 1.5 kbp before the transcription start site, followed by a decreasing sinusoidal-like fluctuation of high GC-content in exons and introns, with introns showing lower content similar to the genome background. Compared to the other species, the bony-fish sequenced have 5% lower GC-content in the promoter regions, and genome-wide. We found that many of these previously missing GC-rich promoter regions are functional, in that they show brain region specific ATAC-Seq peaks for genes with differential expression between the cortex and basal ganglia. These results are summarized in the revised text as well as **Fig. 6**, and clearly demonstrate the ability of high-quality genome assemblies to uncover novel biology. The process of analyzing the GC-content of each assembly also led us to identify six additional chromosomes in the zebra finch VGP assembly, including over 1000 new genes, which were identified by the high GC-content associated with zebra finch microchromosomes. This is the first clear assembly of the microchromosomes in birds.

With the more complete chromosomal assemblies, we were also able to perform a breakpoint and rearrangement analysis for all 16 genomes, as shown in **Fig. 7**. This confirms and conclusively demonstrates extensive chromosome rearrangements in bony fish (teleost) relative to other vertebrates. The skate, a cartilaginous fish closely related to sharks, shows a chromosome organization more similar to the other tetrapod vertebrates. We further complemented this analysis with the 4 additional bat genomes from our companion Bat1K paper (Jebb et al 2020 Nature), and uncovered an increased rate of chromosomal rearrangement in bats, occurring after the last mass extinction event 66 million years ago. Because of the high continuity and chromosome-scale nature of our assemblies, we were able to analyze the effect of these rearrangements on gene content. As an example, we now present a reused breakpoint in bats, which included a repetitive, contiguous set of immune-related genes on human chromosome 6. In all analyzed bat genomes, this region has been uniquely split and rearranged through interactions with multiple chromosomes resulting in six independent reductions of immune related genes in these bat lineages.

Lastly, with regard to the relevance of the technology and assembly approaches used, we would like to note that since genome sequencing technology continues to advance at such a rapid pace, any large-scale, multi-year genomics project must inherently be based on older technology. In the four years we have been working on this project, several new technologies such as PacBio HiFi and Oxford Nanopore ultra-long sequencing emerged. However, the Vertebrate Genomes Project, in part, has helped drive the development of such technologies. Since the beginning, we have partnered with sequencing technology companies, including PacBio and Oxford Nanopore, and have continually evaluated and tested their latest technologies. This continual improvement process has resulted in multiple advances to both the sequencing technology itself and the algorithms used to assemble the data. So while we agree with reviewers 1 and 3 that some of the technologies used here will soon be superseded, we argue that this is inevitable and these technology improvements are an important side effect of large-scale projects like the VGP. Additionally, the assemblies presented here are of the best quality currently available and we have recently completed over 100 additional vertebrate genomes using these methods, vastly expanding the scope and quality of available vertebrate reference genomes. Moving forward, we are continuing to drive improvements in vertebrate genome sequencing technology, and are currently identifying and resolving new issues such as GA-dinucleotide sequence drop-out in PacBio HiFi reads. Similarly, we are continually evolving and improving our assembly process to make use of the best available data.

The principles of the assembly process we introduce are applicable to these new technologies, and we expect for future technologies as well. What our paper does more so than any other genomic study, is perform a systematic evaluation of technologies and algorithms within and across multiple species, allowing us to make rigorous and statistically supported conclusions. Some of our findings support previous hypotheses and refute other hypotheses that were made with only 1 or 2 species. One of our findings is exactly as Reviewer #3 states, that a one-size-fits-all approach does not always work! For example, after applying the standardized pipeline, we found that we needed to modify several steps for the thorny skate and channel bull blenny assemblies to reach our aspired quality metrics. In our revisions, we have better highlighted that this style of customization can be required for particularly difficult genomes. However, in order to scale our project to thousands of genomes, we must develop a standardized approach that works for *most* genomes. We cannot afford to apply a custom set of tools and parameters to every genome and still achieve the needed scalability. Our strategy is to apply our standard pipeline first and then evaluate the resulting assemblies using our proposed quality metrics. In the event our minimum metrics are not achieved, the pipeline is adapted to the specific genome characteristics. We then take the lessons learned from these troublesome genomes and use that information to further improve our automated pipeline. As noted above, it is this iterative improvement process that has moved the project forward towards our ultimate goal of complete and error-free genome assemblies of all vertebrate species.

General updates

There are several updates we made to the manuscript that are not part of any particular reviewer response, but for which we felt it was important to mention to the editor and reviewers. We made several corrections to Fig. 3a. The 3 mammals (2 bats and platypus), and the male zebra finch had a few minor structural assembly updates with some newly identified chromosomes over the revision, which slightly increased the completeness measure. Some of the platypus Y chromosomes were displayed on NCBI as unlocalized, which is now corrected with updated accessions. All updated accessions and versions used over the revision is in Supplementary Table 10, marked in blue. We added RefSeq annotation accessions for the readers as well. The channel bull blenny and blunt-snouted clingfish had a species name update, which is now applied accordingly. Based on findings from other individual animals assembled of the channel bull blenny species (not included in this paper), we found the repeat % measure from GenomeScope was unreliable for this genome and used an alternative method, with details described in Methods. All repeat based analysis was re-done, with our conclusions unchanged. Figures were updated with the new repeat % accordingly in Fig.3cd and g,i. Alternate assembly size of the blunt-snouted clingfish was mis-calculated, and now properly corrected, which also does not change any of the conclusions of the paper.

Referee #1, expertise: genome assembly:

Rhie et al present “Towards complete and error-free genome assemblies of all vertebrate species” that describes the current status and initial data release of the Vertebrate Genome Project (VGP). The paper is expansive describing the VGP sequencing and assembly process, results from 16 new reference genomes, a review of the challenges encountered by these assemblies, and a prospective on future work needed in the field. The paper presents many interesting results, although several major issues need to be addressed.

- The authors assert they have developed “optimal assembly approaches”, yet the presented pipeline is not viable for future projects: 10X Genomics no longer offers linked reads sequencing and the long-read data were generated on out of date RSII and Sequel I platforms. More discussion is needed on what are the best options using current technology. The most obvious issue is the current draft includes only passing mentions of ONT ultra long reads and PacBio HiFi data but we have found them extremely valuable, often more so than PacBio CLR reads (as do many of the authors of this paper such as for the HiCanu algorithm or the Telomere-to-Telomere project). The approaches for polishing, for example, will change substantially when using HiFi reads.

Response: We are happy to hear that the reviewer finds many interesting results in the paper. We fully agree with the reviewer that the optimal assembly approach will evolve over time. However, the results presented here are meant to serve as a guide of the typical challenges encountered during genome assembly, and after four years of working through these problems for many diverse vertebrate genomes we wanted to share the lessons we learned. Many of our suggestions also apply to emerging technologies. For example, regardless of what specific platforms are used, long-reads will be the basis of contig assembly. Purging needs to be applied before scaffolding. Scaffolding will still be required using Hi-C linked reads, ONT ultra long reads, and/or Bionano optical maps, etc.

Currently, as noted by the reviewer, multiple groups, including among our co-authors, are actively developing assembly methods using HiFi (HiCanu). However, we note that HiFi reads were only introduced at the end of last year (2019), 6 months before our paper was submitted, and the methods for assembling this data type remain under active development. We agree with the reviewer that HiFi will change the need for polishing in the future and have expanded our discussion of this in the paper. Our latest results suggest that HiFi-based assemblies can exceed Q50-Q60 without the need for polishing. However, HiFi also introduces some unique challenges such as coverage drop-outs in GA-rich regions, that we have not seen with the CLR reads. This suggests that even HiFi assemblies may require a polishing step in the future, possibly using ONT reads to fill any remaining coverage gaps.

The abrupt discontinuation of 10X Genomics linked reads was unexpected, but was due to legal reasons, not scientific. The general concept of linked reads is well established and multiple replacement technologies are available: e.g. TELL-Seq WGS (Universal Sequencing), stLFR (BGI), and CPTv2-seq (Illumina). We now mention these alternative linked-read technologies as potential substitutes.

With regard to Oxford Nanopore (ONT) sequencing, we note that we trialed this technology multiple times throughout the project but failed for several genomes, including the hummingbird, chicken, and multiple other fish genomes until very recently. The required amount of DNA was unrealistic for some specimens, and for some species like chicken we experienced high rates of pore blocking for unknown reasons. Nanopore has worked on a fix, but the sequencing is still not as efficient

as with mammalian DNA. Given the similarity between PacBio CLR and ONT sequencing, we believe that our proposed assembly process would be directly applicable by simply substituting with the latest ONT r10.3 data type for CLR during contigging. To reflect the flexibility, we now changed “CLR” in **Fig. 2a** to “Long reads”.

The VGP will continuously evaluate the above and other new methods, including with highly heterozygous and repetitive genomes that remain challenging. In doing so, we will continue to adapt our assembly methods to generate the best assemblies possible. Upon reflection, we realized that our manuscript could be mis-read as proposing a one-size-fits-all pipeline, a concern also raised by reviewer 3. To address this we have clarified our message to indicate that one size does not *always* fit all, and now point to examples where we had to change the parts of the pipeline to account for certain genomes with different characteristics.

Below is the added paragraph in the main text, under “Future efforts”:

“Historically, genomics technology advances quickly. We believe the principles of the pipeline we generated will be applicable to new technologies. For the near term, the 10X Genomics linked read technology we used is no longer available as of mid-2020, but can be substituted by three other technologies: TELL-Seq WGS (Universal Sequencing); stLFR (BGI); and CPTv2-seq (Illumina). The ONT data generated here were not considered for further benchmarking beyond contigs due to practical issues concerning systematic base call errors, consistency, and scalability at the time (early 2017). However, the technology has since improved in these areas, and the latest r10.3 base calling leads to higher-quality long reads, which could potentially be used in the contig step of our pipeline. To sequence the 16 species described here, we used the PacBio RSII and Sequel I platforms, and have since used the Sequel II platform to complete over 120 species of a similar or higher quality to the first 16 (**Box 2**). PacBio has recently introduced their next generation of circular consensus reads (CCS) or HiFi, which delivers both reasonable read length (20 kbp) and excellent read accuracy (99.9%) in a single technology, and may eliminate the need for assembly polishing. The VGP infrastructure has been flexibly designed so that we may continually evaluate and adapt to new technologies in order to reach our ultimate goal of producing error-free, gapless, and complete telomere-to-telomere vertebrate genome assemblies.”

- More generally, there is minimal discussion on the remaining challenges: What will be the recommended assembly process in the future? What will be the process for curating >70,000 genomes? How will tens of thousands of genomes be annotated or systematically analyzed together?

Response: We appreciate the reviewer raising this important question. We added an additional paragraph on proposed scaling-up details in **Box 2**, which is about future plans of the VGP:

“The challenges for scaling up that we are working on include: 1) Blanket sample permits for vertebrates, in different countries; 2) High throughput DNA and library sample preparations for ultra-high molecular weight DNA (>200kb); 3) An automated meta-data tracking system for information flow from samples to their genomic data, transcriptomic data, assemblies, and annotations; 4) Increased

efficiency to perform massively parallel high-quality sequencing; 5) Automated assembly pipeline that allows iterative updates, and more efficient assembly compute for hundreds of assemblies simultaneously; 6) A more automated curation process and many curators to manually check each assembly, make fixes where needed, and provide iterative feedback; 7) A more efficient reference-free genome alignment tool that can handle 10,000s of species; and 8) Rapid annotations of genomes in the hundreds per week. The VGP is working on all eight fronts, with the plan that at each Phase of the project will need more and more advanced tools for increased scaling.”

- The review aspects of the paper (Figures 1-3) describes many results that are extremely well established in the literature across many species: long-read assemblies are more continuous; repeats affect continuity; false duplications are common assembly errors, trio-binning improves assembly, etc. These sections should be minimized with appropriate citations e.g. (Pendleton et al, 2015 Nature Methods; Gordon et al, 2016, Science; Matthews et al, 2018, Nature; Nowoshilow et al, 2018, Nature; Liu et al, 2020, Cell; etc).

Response: We agree that some of the individual findings have been presented in the literature, including in the papers cited above, which include authors on the current submitted paper. However, all of the above papers focus on one species (e.g. Gorilla) or one lineage of closely related species (i.e. Soybeans), but not on multiple divergent species. Our study also differs from these past studies in that for the first time we performed a systematic analysis of sequence data types and algorithms on the same individual animal, and used one pipeline across multiple vertebrate species, allowing us to do more quantitative, statistical hypothesis testing. We have not found in the literature the statistically significant correlations we demonstrate with scaffold NG50, contig NG50, genome size, chromosome size, repeat content, heterozygosity, and other genomic features. Further, although these findings may be obvious to some, there are many researchers who remain unaware of current best practices. For example, many investigators continue to carry out genome projects using short reads, without a full awareness of the errors and omissions involved. Nevertheless, as the reviewer indicates, a cohort of scientists, ourselves included, are now convinced of these conclusions, in part based on this study, findings of which we have publicly shared over the last five years. We also believe that our reiteration of these findings, in a quantitative fashion, remains a useful contribution to the field.

We have now cited the above papers, and indicated where they differ and overlap with results in the current paper.

In the “Breadth of data types” section:

“Although we and others have compared some of these technologies on one or two species over the course of our project, to our knowledge this is the first systematic analysis of many sequence technologies, assembly algorithms, and assembly parameters on the same individual.”

In the “VGP assembly pipeline applied across vertebrate diversity” section:

“Although some of these tools had been tested previously, we were not aware of a systematic analysis of a high-quality, haplotype-phased, genome assembly pipeline applied across species

representing a range of genomic diversity, phylogenetic distance, and unique adaptations as the main variables.”

- There are surprisingly few details presented on the bioinformatics approaches involved. The raw data offer very little value without these software; The authors should describe the key algorithms and techniques with citations.

Response: We expanded our description of the key algorithms in the **Methods** and **Supplementary Methods**, with appropriate citations. We also include links to validation software in our GitHub repositories, alongside the assembly pipeline, with expanded documentation.

- The VGP team has made very valuable data contributions through the improved reference genomes, although it seems these genomes are all being described in other companion papers. For example, all of the results in the “New biological discoveries” section are “in press” or “in submission”. This reduces the potential impact of this paper and the authors should clarify which results, if any, are unique to this paper

Response: Actually, these genomes (i.e. their assemblies and sequence characteristics) are being described in this VGP Flagship paper for the first time. The parallel studies all fundamentally rely on and cite the genomes described in this paper. We distributed the biological findings to these other papers to broadly share credit across the consortium and limit the length of any one article. However, in response to the reviewer’s comment, we have conducted additional studies and added novel biological findings to this paper, which we noted above in the response to the editor. This includes discovery of a vertebrate-wide, fluctuating canonical GC-rich pattern in the 1.5 kbp promoter regions, the UTRs, exons and introns of protein coding genes. In the previous genome assemblies, we find between ~20 to 50% of the genes have missing sequence in the 100 bp upstream of the transcription start site and part or all of the 5’ exon sequences, which we would have not known without our new assemblies. This allowed us to discover novel candidate regulatory regions for genes expressed in different cell types in the brain. We also describe new discoveries of comparative chromosome organization across vertebrates, and chromosome evolution in bats. We hope these new biological discoveries satisfy the reviewer’s concerns. We also note that the true impact of the VGP lies in the 100’s of high-quality vertebrate reference genomes that we are in the process of creating, and we expect countless discoveries will spring from these genomes in the coming years.

- Table 1 is a useful decomposition of assembly quality, but many of the fields lack justification of the thresholds used: why only 1Mbp haplotype phase blocks for a 7.C.Q50 assembly, and why only measure gene content and not other functional elements, etc. The authors admit “required quality of an assembly depends on the biological questions” yet the examples are presented without supporting data or justification.

Response: We have now added justifications of the thresholds selected with more details in **Box 3**. This includes rationale for each of the 15 metrics in **Table 1**. For the specific questions the reviewer asked, phase blocks are dependent on the read length used for assembly as well as on the level of heterozygosity in the genome. We set the target realistically to cover one gene and its regulatory regions (typically 10 to 500 kbp) in one phase block, which falls within 1Mbp NG50. New methods generating phase blocks over 1Mbp without parental data are on the horizon. Measuring other functional elements besides genes, such as epigenetic differences along the genome, will require more than sequence and transcriptome data, such as ATAC-Seq data. These functional elements will differ for different cell types, and thus we think this is the purview of downstream annotation and functional studies.

Below is the updated Haplotype phasing part in **Box3**:

“Haplotype phasing: We propose to use *phase block NG50* as a measure for haplotype consistency. A phase block is expected to match one of the parental haplotype sequences, with no haplotype switches. Haplotype consistency is important for gene annotation, because haplotype switches could mix the true gene structure, creating an artificially mixed gene that does not exist in nature. Currently, the most reliable way to measure phase consistency is by using parental sequences. In this study, we use Merqury to infer haplotype blocks from haplotype specific *k-mers*. Accounting for sequencing errors accidentally corrupting a true haplotype specific *k-mer*, we allow short-range switches to occur up to 0.05% (~100 times within 20kbp) within a phase block. We expect block sizes to be more dependent on genome heterozygosity levels, where less heterozygous genomes will have longer runs of homozygosity (ROH) that prevent linking of heterozygous sites when no parental information is used. Heterozygosity will also vary across segments of a genome, and thus, one value may not be equally applicable across the genomes. Therefore, we set smaller block NG50 requirements to cover one gene and its regulatory regions (typically 10 to 500 kbp) in one phase block, which falls within 1Mbp NG50 in the quality metric (**Table 1**), independent of chromosome sizes except for the “finished” quality.

Methods for cross linking distant heterozygous sites using Hi-C or strand-seq are on the horizon, which will help increase phase block continuity. However, accurate measures of the phase blocks are not as well developed with no parental data, presumably because the importance of phasing to prevent errors has been unappreciated. This measure pertains to not only diploid genomes, but also polyploid genomes, which is found in amphibians and fishes.”

We have also reworked the presentation of **Table 1** to make it clear that each column represents a set of quality targets for specific G10K projects of the past, present, and future (B10K-2014, VGP-2016, VGP-2020, Finished). As before, the final column presents the actual quality metrics achieved by the genomes presented in this paper. This nicely shows how genome projects have continually pushed their quality targets higher as technologies improve, and gives a snapshot of where we are now.

- For each metric in Table 1, the authors should describe in detail how it could be computed/estimated from the data and the limitations for measuring them. Relatedly, instead of reporting raw contig NG50 or scaffold NG50 metrics for the X.Y.Q quality metrics, these should values be defined using reliable

blocks since contig length and scaffold length are easily inflated. Other metrics need more precision: “the relatively clean Hi-C heatmap” is insufficient.

Response: We added the requested additional details on how each metric is computed in **Box 3**. We appreciate the reviewers expertise in genome assembly and understand that reliable blocks is a more meaningful measure over contig or scaffold NG50 for indicating structural accuracy. However, contig and scaffold NG50s (more often N50s) are the most widely used metrics for assembly continuity and so we believe it is reasonable to keep them in the shorthand notation over the reliable blocks; we still kept reliable blocks as one of the 15 metrics in Table 1. Although we do have preliminary methods for measuring reliable block size, they are not foolproof. We hope the introduction of the reliable blocks in **Table 1** will motivate the development of better methods to obtain these blocks and be adopted in the community as a basic metric for describing assembly qualities.

We removed the word ‘clean’ and instead provided more details on how to interpret the quality of an assembly based on Hi-C heatmaps.

Below is the section in **Box3** describing the interpretation of Hi-C maps:

“**Chromosome status:** For defining scaffolds as chromosomes, and therefore the percent of the assembly assigned to chromosomes, we believe the current best tool besides genetic linkage or FISH karyotype mapping is Hi-C mapping. We consider a scaffold as a complete chromosome (albeit with gaps) when there is a diagonal signal in the Hi-C mapping plot for that scaffold with no other scaffolds that can be placed in that same scaffold. The Hi-C maps prove useful for identifying large-scale structural aberrations in the assemblies, including false chromosome fusions. The more uniform the Hi-C signal across the main diagonal, the more likely the assembly structure is correct. High-frequency, off-diagonal Hi-C interactions are a strong sign of mis-assembly, of which some can be corrected with manual curation. Based on these criteria, one can then estimate the percent of the genome that is assigned to chromosomes. The thresholds we set from 75% to 100% chromosome-assigned are based on values we generated in this study using different assembly approaches. See Lewin et al. for an alternate view of naming scaffolds.”

Referee #2, expertise: genome sequencing

The authors present an impressive set of assemblies of diverse vertebrate species that are much better than previous assemblies. They use a combination of many technologies and present a lot of evaluations that will be useful for the community. They rely primarily on PacBio CLR for generating contigs, since it performed best of the technologies available when most of this work was performed. They develop a set of useful metrics to summarize different aspects of assembly performance. The QV40 of these assemblies is lower than most current and future assemblies will be, due to the advent of much more

accurate PacBio HiFi long reads in the past year, but QV40 is sufficient for many analyses like annotation of many genes and comparisons between species. The authors also present a really interesting discussion about the prevalence of erroneous duplications in assemblies when they are not done carefully. Besides the smaller suggestions below, my primary concern is that there should be more discussion of the likelihood of collapsed segmental duplications in the assemblies, and ideally evaluations that show how prevalent these collapses are. Otherwise, I expect this manuscript will be very useful to the community and represents a large amount of excellent technical work that I expect will continue to lead to new discoveries and future improvements in assemblies.

Response: We are glad the reviewer views the paper positively. Following the reviewer's request, we now measured collapsed regions, including collapsed segmental duplications and collapsed high-copy repeats, separate from other assembly errors. We find that most of the assemblies do have collapsed base pairs, and the amount of them correlated directly with repeat content of the genome (newly added **Extended Data Fig. 4**). Most of the collapsed high-copy repeats did not match any known repeat type, while the majority of identified ones consisted of satellite and simple repeats. Others included LTR, Line, and SINE repeats, but not many coding genes. We have added the values of collapsed bases per Gbp in **Fig. 3a** for all our assemblies, which ranged from 1.3 ~ 31.4 Mb/Gbp.

The newly added text is under "Repeats dramatically impact continuity" section:

We further assessed the impact of overall repeat content in the collapsed regions. Unlike the other evaluations, we used the curated assemblies to avoid mapping biases as this analysis heavily relies on mapping of the raw CLR reads. We found that the number of collapsed regions (~100 to 750; 2,379 in the skate), their total size per Gbp (0.7 to 17.5 Mb/Gbp; 31.4 Mb/Gbp in the skate), total estimated missing bases due to collapses per Gbp (~4.5 to 85.6 Mb/Gbp; 136.7 Mb/Gbp in the skate), and the number of genes in the collapses (0 to 513), all correlated with percent repeat content of the genomes (**Extended Data Fig. 4a-d**). Interestingly, the average collapsed length did not correlate with repeat content (**Extended Data Fig. 4e**); instead it was within the range of the average CLR read lengths (10–35 kbp). There were no correlations between the number of collapsed bases and genome heterozygosity or genome size (**Extended Data Fig. 4f,g**). Each collapsed region was further classified as segmental duplications or collapsed high-copy repeats, when its fraction annotated as repeat was below or above 90%, respectively, by WindowMasker or RepeatMasker (**Supplementary Methods**). This analysis revealed 77.4 to 99.2%, depending on species, of the collapsed regions consisted of unresolved segmental duplications (**Extended Data Fig. 4h**). The majority of sequences in collapsed high-copy repeats across species did not match any known repeat and was only found by WindowMasker (**Extended Data Fig. 4i**). The most prevalent type of identifiable collapsed repeat was the satellite arrays, almost exclusively found in the platypus, comprising ~2.5 Mbp. Simple repeats were the major sources of the collapsed repeats in the thorny skate (~400 kbp). All other types of repeats in the collapsed repeats were fewer in overall proportion and varied according to lineage, with proportionally higher types of long terminal repeats (LTR) in birds, and short and long interspersed nuclear elements (LINE and SINE) in mammals, and DNA repeats in the amphibian (**Extended Data Fig. 4j**). Among the genes in the collapses, many were repetitive short non-coding RNAs. All of the above findings quantitatively demonstrate the impact that repeat content has on the ability to produce highly continuous and complete assemblies.

In addition to the “Collapsed Mb per Gb” column, we made a few corrections to **Fig. 3a**. The 3 mammals, the bats and the platypus, and the male zebra finch had a few minor structural updates with some newly identified chromosomes over the revision, which slightly increased the completeness measure. Some of the platypus Y chromosomes were displayed on NCBI as unlocalized, which is now corrected with updated accessions. All updated accessions and versions used over the revision is in **Supplementary Table 10**, marked in blue. We added RefSeq annotation accessions for the readers as well. The channel bull blenny and blunt-snouted clingfish had a species name update which is now applied accordingly. Based on findings from other individuals assembled of the same channel bull blenny species from our co-author (not included in this paper), we found the repeat % measure from GenomeScope was unreliable for this genome and used an alternative method instead, with details described in **Methods**. All repeat based analysis was re-done, with our conclusions unchanged as before. Figures were updated with the new repeat % accordingly in Fig.3cd and g,i. Alternate assembly size of the blunt-snouted clinfish was mis-calculated, and now properly corrected.

1. In the Fig 1 caption, this sentence was not clear to me “*Assembly generated with software updates to FALCON29, following CLR + Linked + Opt. + Hi-C combination.” This makes it sound like v1.0 is the same as “CLR + Linked + Opt. + Hi-C” even though they are different assemblies. As I understand from later in the text, the difference is only the updated FALCON version?

Response: Yes, the main difference is the FALCON version used, as well as the Salsa version used for Hi-C scaffolding. We revised the caption to clarify the differences between v1.0 and CLR + Linked + Opt. + Hi-C. *Contigs generated with an updated version of FALCON (**Supplementary Table 2**). Scaffolding was performed in the order of CLR + Linked + Opt. + Hi-C combination, with a slightly older version (2.0 vs. 2.2) of Hi-C scaffolding which was more aggressive in joining.

2. Fig 1H would be easier to see on a log-log scale or with a separate zoomed inset since most of the chromosomes are much smaller than the first few

Response: We thank the reviewer for this suggestion. We followed the reviewer’s first suggestion and replaced Fig. 1h to the log-log scale version, as we found it spread out the chromosome sizes more evenly, showing the correlation better.

3. It would be useful to state the reason why short reads have much shorter contig lengths (e.g., due to breaks at tandem repeats, mobile elements, or other repeats?) after stating “These findings indicate that given current sequencing technology and assembly algorithms, it is not possible to achieve high contig continuity with short reads alone.”

Response: We modified the sentence to indicate that it is consistent with a paper which shows theoretically proof that it is impossible to resolve repeats with short reads:

“These findings are consistent with theoretical predictions and demonstrate that given current sequencing technology and assembly algorithms, it is not possible to achieve high contig continuity with short reads alone as it is impossible to bridge through repeats longer than the length of a read.”

Moreover, as stated in the newly added GC-content section, we found short reads often were deficient in GC-rich regions, which will result in contig breaks due to coverage drops.

4. State how Q40 was determined earlier on (short read mapping and/or kmer-based)

Response: Earlier in the paper, the short read mapping approach was used to determine Q40 on the hummingbird, which is now mentioned in this section of the paper.

5. The question of how well segmental duplications and other difficult regions are represented is one of the biggest holes in the current manuscript. I suspect that they were frequently collapsed or not represented well using the technologies and methods available at the time of this work, which is understandable. I don't expect the authors to improve their assemblies, but it is important that this limitation be highlighted and ideally quantified in some metrics. Can you add a metric related to potential errors in segmental duplications and other repeats to Fig 3a (e.g., identifying regions with higher than expected coverage)?

Response: We thank the reviewer for this suggestion. As in the above response, we now added collapsed bases per Gbp in **Fig. 3a** with a full spectrum shown in **Extended Data Fig. 4**. As the reviewer suspected, collapsed regions were found for all assemblies, correlating with repeat content.

6. Can you add a clearer definition of “repeat” to the legend of Fig 3 than just referencing GenomeScope, since there are so many types of repetitive genomic elements?

Response: We added the definition of repeat to the figure legend:

“Repeat content in GenomeScope was measured by modeling the *k-mer* multiplicity from sequencing reads directly, and thus was neither restricted to certain repeat families, nor the completeness of an assembly.”

7. On page 16, they state “The total number of genes annotated went down in the VGP assemblies (Extended Data Table 1), which we believe is because the VGP assemblies have fewer false duplications (0.5 to 1.7%) compared with their previous counterparts (5.1 to 12.3%; except for the climbing perch at 0.4 vs 1.2%; Supplemental Table 19).” Wouldn’t it be possible to show that the specific genes in the old assemblies and not in the new assemblies are due to duplications?

Response: Yes, in Figures 5e,f, and g in the next paragraph we show examples exactly of what the reviewer ask for, of specific genes in the older assemblies that are false duplications, and are present in the new VGP assemblies in their correct form. We have revised the wording, such that there is a clear link from the finding of reduced number of genes in the old assembly as being false duplications not present in the VGP assembly.

Below is the updated figure legend for Fig.5e-g:

“e-h, Example assembly errors and associated annotation errors in previous (old) reference assemblies corrected in the new VGP assemblies. Both haplotypes of *SPC25* (e) were erroneously duplicated on two different contigs, annotating one as *SPC25*-like. The 5’ end part of *GABRG2* (f) was erroneously annotated as a separate *GABRG2*-like protein coding gene, due to false duplication of exons 2–5. The *VTG2* gene (g) was annotated on 3 scaffolds as part of 3 separate genes, two *VTG2*-like and an intron of *CANP13*.”

8. On page 16, could the authors state what the +- values are (e.g., StDev), and do a statistical test to see if this is a significant change? “Further supporting these results, the VGP assemblies had 6 to 13% higher k-mer completeness (95+-3.5% average; Fig. 3a) compared with the prior assemblies (88+-4.3% average; Supplementary Table 19).”

Response: We have added S.D. to the text. There were only n = 4 older assemblies to make a pairwise comparison possible; in 3 of them the VGP assembly was more complete and 1 the VGP assembly had similarly high completeness, which we think lends itself to an ambiguous statistical test with n = 4. However, it was possible to perform an unpaired t-test comparison on older versus the 16 new VGP assemblies versus the four older assemblies, and here the difference was significant (p = 0.0047; Means 87.900, 94.512; S.D. 4.262, 3.606; SEM 2.131, 0.875; n = 4, 17).

We have included this information in the revised paper:

“Further supporting these results, the VGP assemblies had 0 to 13% higher k-mer completeness (95 average +3.5% S.D. average; Fig. 3a) compared with the prior assemblies (88+4.3%; Supplementary Table 19; p = 0.0047; n = 4 prior and 17 VGP assemblies; unpaired t-test).”

9. In Fig 5b, could the author’s clarify what the horizontal line in each bar means?

Response: We revised the figure legend accordingly (and panel letters, a through i) of what the “inside bar” is:

“c, Total number of coding sequence (CDS) transcripts (full bar) and portion fully supported (inside bar) in the previous and new VGP Anna’s hummingbird, zebra finch, and platypus assemblies using the same input data (accessions in **Supplementary Table 19**).”

10. On pages 19-20, the authors state “We further note here, that base error rate inferred directly from k-mers was more comprehensive and accurate than the widely used mapping and variant calling protocols, which artificially inflated QV values (Supplementary Table 17) because they exclude regions that are difficult to map.” While this is true, it would be good for the authors to note that k-mer approaches also can artificially inflate QV values, particularly in collapsed segmental duplications or other repeats where the small k-mers used may be accurate even if the assembly is not. Merqury’s k-mer based metric is a good way to get QV for the VGP assemblies, but it’s important to note its strengths and limitations.

Response: In highly repetitive regions, it is possible that an actual error in an assembly overlaps a true k-mer by chance, due to the smaller k-mer space in the repetitive region. This may artificially inflate the QV. However, we believe these regions are not accurately measured by mapping and variant calling protocols either, as these regions are difficult to map reads correctly. Nevertheless, we agree with the reviewer and now mention the strengths and limitations of each method in **Box3**:

“**Base accuracy:** There are multiple ways of measuring base-level accuracy, called base pair QV. One approach is to align (i.e map) highly accurate reads to the assembled genome and call base errors similarly to variant calling. We define “mappable” as all reads that align, excluding low-coverage and excessively high-coverage regions (see **Supplementary Methods** for exact parameters used), where we can rely on base error calls. The other, more reliable, way to measure base accuracy is using *k-mers* found both in the assembly and highly accurate unassembled reads. All *k-mers* found only in an assembly are likely produced from a base pair error. By counting these k-mers and comparing the fraction to all *k-mers* found in an assembly, we can estimate the error rate and calculate the quality value using the k-mer survival rate. We found *k-mer* based methods include unmappable regions and thus avoid over-estimated QVs from the mapping-based approach. We note that both mapping-based and *k-mer*-based approaches have limitations of measuring base accuracy in highly repetitive regions, as the short reads are difficult to map accurately and a *k-mer* with a true error may match by chance with some other true *k-mer* that belongs elsewhere in the genome. This may artificially inflate the QV, especially in those repetitive regions.”

Also, as the reviewer noted above, a QV estimate alone is not sufficient to measure the quality of an assembly. A base (or *k-mer*) in a collapsed region can still be locally correct even if the assembly is collapsed or otherwise incomplete. We specifically include measure collapsed regions (including collapsed segmental duplications) in the category of structural accuracy, as these regions are naturally excluded when obtaining reliable blocks along with regions of poor read support (more in **Box3**).

11. They also state “VGP assemblies exceeding Q40 contained fewer frameshift errors, as predicted⁷⁶, and therefore we recommend targeting a minimum QV of 40 whenever possible.” This seems like a somewhat arbitrary target, and mostly based on the fact that it is what the VGP achieved for most assemblies. Given the advent of more accurate PacBio HiFi sequencing, which can result in assembly QV’s >50, it seems like an outdated target. However, it is helpful that the authors state in the next paragraph what different QV40 assemblies are useful for.

Response: We chose a Q40 target because it was one of the first definitions of a “finished” sequence during the Human Genome Project. However, as illustrated in Figure 1 of reference 76 (Koren, S., Phillippy, A. M., Simpson, J. T., Loman, N. J. & Loose, M. Reply to ‘Errors in long-read assemblies can critically affect protein prediction’. Nat. Biotechnol. 37, 127–128, 2019), Q40 also happens to correspond to the QV for which the majority of coding sequences will remain unaffected by sequence errors. We now added this information earlier in the paper:

“Q40 is the mathematical inflection point between where genes go from usually containing an error to usually not⁴⁶”.

As we noted above in response to reviewer 1, HiFi was not available by the time we initiated the VGP. However, we do agree and are aware that HiFi will let us achieve Q50 or higher in the near future. Thus we list >Q50 in our revised notation (7.c.P6.Q50.C95) in the column now changed to VGP-2020, as a future target in **Table 1**.

12. It would be good to have a more descriptive README for the VGP code repository at <https://github.com/VGP/vgp-assembly>

Response: We have added more detail to the VGP README file.

Referee #3, expertise, evolutionary genomics:

I enjoyed reading the manuscript “Towards complete and error-free genome assemblies of all vertebrate species” by VGP consortium. The study is a tour de force on data generation and assembly methods focused on 16 vertebrate species. Starting from the assembly of a hummingbird genome, the study sets up a pipeline that is then applied to other 15 vertebrates to produce high contiguity assemblies. The starting raw data is highly diverse and comprises all the state-of-the-art sequencing methods available; it will be an amazing resource for the scientific community. The study is well designed and written, and

it is mostly flawless at the technical level. Overall, this is a titanic task and I would like to congratulate the authors for their work and contribution.

Response: We thank the reviewer for the congratulatory note, and appreciate the positive feedback.

However, the results lack novelty or new biological insights. As indicated by the headers of most sections, which are almost tautological, all the results are confirmatory and technical: “Long-read assemblies are more continuous”, “Repeats affect continuity”, “False duplications are common assembly errors”, “Curation is important for generating a high-quality reference”, “Trios help resolve haplotypes”... those are basic principles of genome assembly and have been known and documented for very long time. When things start to get interesting (“New biological discoveries”!!!) the reader is sent to a list of other papers recently submitted or published by members of the consortium, which makes clear no biology was not to be included in this manuscript.

Response: We believe each section has more novelty than our headings indicate, but it is hard fitting novelty in the Nature heading format, which allows only up to 46 characters including spaces. We have added a few more words to some headings to bring out the novelty, and requested an exception from the editor. We appreciate the reviewer’s expertise in genome assembly, and agree that a lot of the findings were documented elsewhere. However, we argue that most of the documented principles are fairly recent, and were built on only one or a few species. We believe the novelty of our results lies on a systematic comparison among 16 very divergent species, which is further explained in our response to reviewer #1. We do admit that most of the biological novelty was contained in our companion papers and our flagship paper contained perhaps too little. In response to this reviewer’s concern and that of reviewer #1, we have now added additional findings in the biological discoveries section. To avoid weakening our other papers, we sought to add completely new results, which took some time. Those new findings are highlighted above in the general response to the editor and reviews.

A methodological issue is a suggestion of using a single pipeline to assemble different vertebrate genomes, a pipeline which has been set up based on a single genome (hummingbird). This goes against the advice from other seasoned initiatives, like Assemblathon (this referee has no affiliation to this project). Those show that for each combination of species and raw genomic data, different combinations of software and approaches are necessary to reach the best assembly possible. The genome assembly of each species is a new and unique bioinformatics experiment, that requires a different set of tools. This goes against the one-size-fits-all approach applied and proposed in this manuscript.

Response: We absolutely agree with the reviewer that a one-size-fits-all approach is not always the “best” approach. However, it becomes unrealistic to apply a specialized approach to every single species, when needing to scale up to 100s to 10,000s of species. The purpose of building a standardized pipeline is to find an automated approach with little loss of quality that works for most species. The question remains if we could predict in advance that an assembly will fail with a standardized pipeline. From the 16 species, we learned that when a genome is more repetitive, the continuity is more likely to

suffer, and that an alternative specialized approach will be required. Learning from this lesson, we now measure the repeat content in advance before running the pipeline. However, this does not mean that the assembly process for a highly repetitive genome is completely changed. Key major steps are applicable and would remain. We modified the text to better clarify these points as detailed below.

In the Repeats ... section:

“When applying the same standard VGP assembly pipeline (**Fig. 2a**) across species, all but two of the 17 assemblies exceeded the aspired continuity metrics before curation (**Supplementary Table 13**). The two species, the thorny skate and channel bull blenny, did not meet the minimum NG50 of 1 Mbp contig size and required manual modifications to the pipeline (**Supplementary Note 2**), indicating that exceptions still exist that challenge the automated approach. We found contig NG50 had a significant exponential decrease with increasing repeat content (**Fig. 3c**) and could be used as a predictive value before deciding to take the automated approach. In particular, the thorny skate, contained the highest repeat content (54.1%), substantially higher than those of the other fishes (>24%), mammals (15–35%), and birds (10–15%; **Fig. 3c** and **Supplementary Table 13**) as measured by k-mer multiplicity from sequencing reads.”

In the “Future efforts” section:

“Future efforts should include development of tools that can automatically estimate parameters needed to assemble a genome accurately with different repeat content, heterozygosity levels, and ploidy. However, key steps in our pipeline will remain applicable, such as contigging with long reads first, purging false duplications before scaffolding, polishing with the mitochondrial genome present, and manual curation including Hi-C evaluation.”

We also argue that the conclusions from Assemblathon studies reflect the immaturity of the field and methods at the time, which did not hold true in the current study. A problem with the Assemblathon 2 study at the time was that we had too many variables that changed simultaneously, including different species, different sequence data types, and different algorithms. This made it difficult to discover the cause of differences between assemblies. Thus, here we purposely avoided this approach, and instead, applied multiple approaches to one species, and one approach across multiple species, trying as much as possible to change one variable at a time. This led to clearer conclusions. This rationale is now further described in the revised manuscript:

“Towards this end, and seeking to improve upon the G10K Assemblathon 2 effort³⁰ with multiple variables changing at once, we designed controlled experiments, and first evaluated multiple genome sequencing and assembly approaches extensively on one species, the Anna’s hummingbird (*Calypte anna*).”

1) I understand the advantages of NG50, and I agree with its use to assess assembly quality. However, for comparison and legacy reasons, I believe N50 should be still included in tables and similar.

Response: We added columns showing the N50 numbers in **Supplementary Table 11**. We are trying to avoid using N50s as the standard measure to compare assemblies, since it heavily depends on the assembly size, which often is not the true genome size. We hope the community adopts the use of NG50s over the legacy N50s.

2) A big emphasis is made on long contiguity assemblies, which favours tech like PacBio and ONT. I completely agree with this view. However, there is almost no consideration to error rate, only to discard an ONT dataset at one point. Of course, we need to know the “real” sequence to infer error rate, which we do not have. But maybe the different assemblies from one genome could be mapped against the preferred one to calculate how much they do diverge at the nucleotide level.

Response: We do have a comparison of the different assemblies of one species, the hummingbird, mapped to the curated version, which shows the number of mis-joins in **Fig. 1d**. However, we did not show the nucleotide level divergence, or identity, as the different assembly approaches contain different haplotypes and may inflate or deflate the actual error rate that the reviewer asks for. The consensus error rate, or the base level QV, is a closer measure to the reviewer’s requested ‘inferred error rate’ and is already compared in **Supplementary Table 7**. This table shows both *k-mer* based and mapping-based QVs of Illumina only, CLR, and CLR + ONT assemblies. Regardless of the QV measurement method, the QVs show that CLR + ONT assembly has a much lower base level quality compared to CLR only, even after polishing with CLR and Illumina reads. In addition, we had to drop ONT not only for the higher error rate, but also for scaling issues as mentioned in the response to Reviewer #1. It is true the short-read only assemblies still have higher base-level accuracy than short-read plus long-read combined as in our study; however, we found more complete contigs were helpful to obtain more complete gene structures. Moreover, we believe with the latest HiFi-based assemblies, we will easily achieve comparable base-level accuracy to that of short-read assemblies.

3) Line 278, I’d suggest changing “correct” to “best assembly” or similar.

Response: We changed the wording from ‘correct’ to ‘more accurate’. We felt that saying ‘best assembly’ was too subjective.

“Considering the final curated version of this assembly as more accurate (**Extended Data Fig. 1a**), we identified many mis-joins (supported by multiple lines of evidence) introduced by the automated contig and scaffolding methods (**Fig. 1d**).”

4) Line 409, I do not agree that repeats affecting contiguity in assemblies is a hypothesis, I think it is a well-established fact. Hence the need for long-read technologies, optical mapping, HiC, etc.

Response: We changed the wording to state: “All of the above findings quantitatively demonstrate the impact that repeat content has on the ability to produce highly continuous and complete assemblies.”

As stated in the Response to the Editor’s comment, compared to the other studies, we believe this study performed a systematic evaluation of technologies and algorithms within and across multiple species, allowing us to make rigorous and statistically supported conclusions.

5) I find the metrics in Table 1 confusing and not that helpful, but this might be just me. Unless the genome is considered “complete”, the other columns are hard to compare between species with different genome sizes.

Response: We disagree with the reviewer that it is hard to compare between species with different genome sizes. Metrics affected by genome size are already normalized by the genome size or chromosome size. For example, in the continuity measures, we used the number of gaps per Gbp, in order to not penalize larger genomes for having more gaps. Also, based on our observation that scaffold NG50 correlates with genome size perhaps due to the larger chromosomes, we set the target goal for VGP-2020 as chromosome NG50, which controls for actual genome size, which N50 does not.

The reviewer’s comment made us realize the reliable block metric in the table was affected by chromosome (scaffold) sizes. We had defined this metric as % scaffold NG50 in our original submission, but we found larger scaffold NG50 (in particular the two-lined caecilian had scaffold NG50 = 486.9 Mbp) penalized this reliable block metric. Thus in the revised submission, we changed the metric to the absolute reliable block NG50.

We do not think genome size is going to be as an important determining factor for the metrics in each column as the reviewer indicates, because we have shown that some metrics are affected more by the repeat content and heterozygosity levels than by genome size. Our findings also indicate that it is not necessary to have a complete and error-free reference genome in order to measure completeness and accuracy of another assembly. Many of the metrics and measures we discuss in Table 1 were designed to be “reference free”.

To help make each column of the table less confusing, we have added column headers to Table 1 that relate these columns directly to quality targets set by specific projects, including the G10K in the past, present, and future. The columns headers are now “Finished”, “VGP-2020”, “VGP-2016”, and “B10K-2014”. The basic assembly metric notations are still presented underneath these headers, for those who find it useful as a more quantitative measure. We also now included five metrics in this shorthand notation, as opposed to the previous three, to capture the minimum metrics we are using for the VGP: contig NG50; scaffold NG50; phase block NG50; QV; and % assigned to chromosomes. We also added more explanation in the main text and **Box 3** with justifications of the thresholds noted in each quality category. We feel this revision to **Table 1** nicely illustrates how improvements in sequencing technology over the past decade have d

Reviewer Reports on the First Revision:

Ref #1

The revised manuscript has satisfied my concerns, especially by clarifying the evolving nature of genome sequencing & assembly technology, more precisely defining & justifying the metrics used, and the additional figures 6 & 7 highlighting the insights that were gained through their improved assemblies. I applaud the VPG consortium for their excellent work.

Ref #2

The authors have comprehensively responded to all of the reviewers' comments, including adding biological insights gained from the new high quality assemblies, analyzing limitations of the current assemblies, and demonstrating how the methods here chart a path forward in a rapidly improving technology landscape. My only further suggestion is to add the appropriate citations for the new linked read technologies now referenced as replacements for 10x:

TELL-seq (<https://doi.org/10.1101/852947>), stLFR (<http://dx.doi.org/10.1101/gr.245126.118>), and CPTv2-seq (<https://doi.org/10.1038/nbt.3897>)

Ref #3

I have read the resubmission of the manuscript "Towards complete and error-free genome assemblies of all vertebrate species" by VGP consortium. I want to reiterate my previous comments on the amazing data generation brute force of this paper, even if some technical aspects are not perfect. However, I feel the concerns of these referee, or others for that matter, have been addressed.

I would like to appreciate that the authors have done some changes to the paper. However, they acknowledge and accept the technical and scientific criticisms raised by this referee but also others. Namely, the lack of novel findings. While they have added some novel GC-rich promoters and the analysis of chromosome breakpoints, these are just a few additional lines. If these were such important findings as to be justify a publication in a high profile journal, these would be in the title; the breakpoints are not even mentioned in the Abstract.

The rest of the rebuttal arguments accept the criticisms of the referees and do very little to fix them. Just to pick a few, the response letter acknowledges that the approaches are outdated, but argues that the VGP has helped these technologies to advance or that another 100 are coming; none of these entitles this paper to be published. Chromosome-level genomes of more difficult to sequence organisms are often published in GigaScience, the fact that this article contains 16 vertebrates does not make it more insightful than those due to the lack of biological novelty. The rebuttal also acknowledges that one of their findings is exactly that a one-size-fits-all approach does not always work, which has been repeatedly demonstrated by the Assemblathon researchers for years; the text fails to highlight what is the novelty is here.

To the criticism of some of the findings being well-established in the field ("better data produce better assembly") the rebuttal says that originally they were proven from one or two species genome papers, while here they are proven with multiple species; this is irrelevant, as far as I understand, genomes are assembled one by one, and these findings were shown in several genomes (just not in a single paper). These are why we have long read technologies now.

Ultimately, I feel that the technical issues are not amended, and the most important criticisms for this type of publication, lack of novelty and biological insights, are not dealt with.

Author Rebuttals to First Revision:

Referee #1:

Reviewer comment: The revised manuscript has satisfied my concerns, especially by clarifying the evolving nature of genome sequencing & assembly technology, more precisely defining & justifying

the metrics used, and the additional figures 6 & 7 highlighting the insights that were gained through their improved assemblies. I applaud the VPG consortium for their excellent work.

Response: We thank the reviewer for the positive feedback. Due to space limitations, we had to move the new description on evolving nature of genome sequencing and assembly technology to the supplement, and Box 3 on defining and justifying the metrics to the supplement. We keep the two new figures (now Figures 4 and 5) in the main text.

Referee #2:

Reviewer comment: The authors have comprehensively responded to all of the reviewers' comments, including adding biological insights gained from the new high quality assemblies, analyzing limitations of the current assemblies, and demonstrating how the methods here chart a path forward in a rapidly improving technology landscape. My only further suggestion is to add the appropriate citations for the new linked read technologies now referenced as replacements for 10x: TELL-seq (<https://doi.org/10.1101/852947>), stLFR (<http://dx.doi.org/10.1101/gr.245126.118>), and CPTv2-seq (<https://doi.org/10.1038/nbt.3897>)

Response: We are glad that the reviewer is satisfied with the revised paper. We added the citations suggested for these new linked read technologies. We thank the reviewer for finding them. We had to move this section to the supplement, due to space limitations. Thus, the citations are in the supplemental document.

Referee #3:

Reviewer comment: I have read the resubmission of the manuscript “Towards complete and error-free genome assemblies of all vertebrate species” by VGP consortium. I want to reiterate my previous comments on the amazing data generation brute force of this paper, even if some technical aspects are not perfect. However, I feel the concerns of these referee, or others for that matter, have been addressed.

I would like to appreciate that the authors have done some changes to the paper. However, they acknowledge and accept the technical and scientific criticisms raised by this referee but also others. Namely, the lack of novel findings. While they have added some novel GC-rich promoters and the analysis of chromosome breakpoints, these are just a few additional lines. If these were such important findings as to be justify a publication in a high profile journal, these would be in the title; the breakpoints are not even mentioned in the Abstract.

The rest of the rebuttal arguments accept the criticisms of the referees and do very little to fix them. Just to pick a few, the response letter acknowledges that the approaches are outdated, but argues that the VGP has helped these technologies to advance or that another 100 are coming; none of these entitles this paper to be published. Chromosome-level genomes of more difficult to sequence organisms are often published in GigaScience, the fact that this article contains 16 vertebrates does not make it more insightful than those due to the lack of biological novelty. The rebuttal also acknowledges that one of their findings is exactly that a one-size-fits-all approach does not always work, which has been repeatedly demonstrated by the Assemblathon researchers for years; the text fails to highlight what is the novelty is here.

To the criticism of some of the findings being well-established in the field (“better data produce better assembly”) the rebuttal says that originally they were proven from one or two species genome papers, while here they are proven with multiple species; this is irrelevant, as far as I understand, genomes are assembled one by one, and these findings were shown in several genomes (just not in a single paper). These are why we have long read technologies now.

Ultimately, I feel that the technical issues are not amended, and the most important criticisms for this type of publication, lack of novelty and biological insights, are not dealt with.

Response: We are happy to hear that the reviewer still believes we have generated “amazing data”. We, however, disagree with the remaining concerns. The reviewer states that we have not satisfied the other reviewers’ comments, but reviewers #1 and #2 have clearly stated that they are satisfied.

We disagree that our methodical and biological findings are not novel. It seems that the reviewer was expecting mainly a ‘biological findings’ paper as in standard genomic studies in Nature, where the main focus of the paper is to generate a genome of one or more species and then report novel biological discoveries from those genomes. However, this is not that type of paper. This is the flagship paper of a multi-year consortium effort, with the main focus being on methods development for high-quality genome assemblies and annotations, examples of novel biological findings enabled by these genomes, and a formal announcement of the consortium and its future plans. Some of the methodological findings have been reported, but not the entire package of this study. More biological findings appear in the approximately 18 other papers that will be published concurrently or earlier, which cite this flagship paper for either the source of the genomes, the methods, or the initial biological findings. All future publications making use of the now 120 genomes that have utilized the VGP pipeline reported in this study will build upon and cite this flagship paper for the methodical details.

Further, we reiterate that these assemblies are still amongst the highest quality in the public databases. Genomics technology is evolving rapidly and it is likely that there will be better assemblies in the future, but the methods used to generate those will have learned from the developments and lessons presented in this paper.

We did not include novel biological findings in the title, as the main focus of the paper is on methods that get us towards generating complete and error-free genomes of all vertebrate species, but we now highlight the chromosomal breakpoint evolution finding in the abstract.

We disagree that we have not learned anything novel by having 16 species, as opposed to 1 or 2 species in prior studies. With 16 species, all assembled with the same approaches, we were able to perform quantitative statistical analyses across species in a manner that has not been done before for vertebrates. We were able to learn about specific levels of heterozygosity and repeat content that influenced assembly metrics, how to identify and remove false duplications, the impact of those false duplications on downstream biological analyses, the first more complete sex chromosomes among mammals and birds, obtaining mitochondrial genomes in single molecule reads, and more generally — how to generate high-quality assemblies across multiple species.